# Growth phase diets diminish histone acetyltransferase Gcn5 function and shorten lifespan of *Drosophila* males

Shoko Mizutani[1], Kanji Furuya[1,2], Ayumi Mure[1], Yuuki Takahashi[1], Akihiro Mori[3,10], Nozomu Sakurai [4,11], Takuto Suito[5,6], Kohjiro Nagao[6,12], Masato Umeda[6,13], Kaori Watanabe [1,14], Yukako Hattori [1,7,8✉] & Tadashi Uemura [1,7,9✉]

## Abstract

**The nutritional environment in early life, referred to as the nutrition history, exerts far-reaching health effects beyond the developmental stage. Here, with *Drosophila melanogaster* as a model, we fed larvae on diets consisting of a variety of yeast mutants and explored the resulting histories that impacted adult lifespan. A larval diet comprised of yeast *nat3* KO shortened the lifespan of male adults; and remarkably, this diet diminished the function of histone acetyltransferase Gcn5 in larvae. Concordantly, perturbation of *Gcn5*-mediated gene regulation in the larval whole body or neurons significantly contributed to the earlier death of adults. The *nat3* KO diet is much more abundant in long-chain fatty acids and branched-chain amino acids (BCAAs) than the control yeast diet. Supplementing the control diet with a combination of oleic acid, valine, and acetic acid recapitulated the effects of the *nat3* KO diet on the larval transcriptome and the lifespan of males. Our findings strongly suggest a causal link between a fatty acids- and BCAA-rich diet in developmental stages and lifespan reduction via the adverse effect on the Gcn5 function.**

**Keywords** BCAA; DOHaD; Fatty Acid; Gcn5; Histone Acetyltransferase
**Subject Categories** Chromatin, Transcription & Genomics; Metabolism

## Introduction

Postembryonic development is characterized by massive and rapid growth of juveniles. This developmental stage, in early life, is heavily influenced by the quality and quantity of nutrients consumed by the juveniles (Shim et al, 2013; Leulier et al, 2017; Bhutta et al, 2017; Tu et al, 2023). Moreover, the nutritional environment in early life, referred to hereafter as nutrition history, affects not only that stage, but it results in long-term health effects beyond the developmental stage, even in late adult stages. Such long-term effects have been documented in human epidemiological studies and rodent models, and support the "developmental origin of health and disease (DOHaD)" hypothesis (Hanson and Gluckman, 2014; Langley-Evans, 2015). For example, malnutrition during pregnancy and/or the lactation period affects body weight of newborns and/or infants, which translates to varying risks of coronary heart disease and obesity during adulthood, as well as impacts on lifespan (Eriksson et al, 1999; Ozanne and Hales, 2004; Barker and Thornburg, 2013; Kramer et al, 2023).

A growing body of epigenetic studies has provided plausible explanations concerning mechanisms underlying such long-term effects. Diets may often contain varying quantities of metabolic intermediates, which influence the deposition and removal of covalent modifications of DNA and histones, alter chromatin structure, and lead to long-term perturbations of gene expression if some of those modifications are retained (Etchegaray and Mostoslavsky, 2016; Reid et al, 2017; Sharma and Rando, 2017; Schvartzman et al, 2018; Cavalli and Heard, 2019; Dai et al, 2020). For example, when female mice during pregnancy are fed on diets supplemented with methyl-donating substances, the coat color of the offspring is affected via increased DNA CpG methylation of an upstream regulatory element of an allele of the *Agouti* gene (Waterland and Jirtle, 2003; Jirtle and Skinner, 2007).

However, current mechanistic descriptions of such far-reaching effects still lack detail, leaving some key gaps that need to be filled: first, which nutrition histories cause such long-term effects, apart from general malnutrition and diets supplemented with methyl-donating substances? How can we isolate and scrutinize critical, effective nutrition histories among the wide variety of nutritional

[1]Graduate School of Biostudies, Kyoto University, Kyoto, Japan. [2]Radiation Biology Center, Kyoto University, Kyoto, Japan. [3]Laboratory for Developmental Dynamics, RIKEN Center for Biosystems Dynamics Research, Saitama, Japan. [4]Bioinformation and DDBJ Center, National Institute of Genetics, Mishima, Japan. [5]National Institute for Physiological Sciences, National Institutes of Natural Sciences, Okazaki, Japan. [6]Graduate School of Engineering, Kyoto University, Kyoto, Japan. [7]Center for Living Systems Information Science (CeLiSIS), Kyoto University, Kyoto, Japan. [8]JST FOREST, Tokyo, Japan. [9]AMED-CREST, Tokyo, Japan. [10]Present address: Rowett Institute, University of Aberdeen, Aberdeen, UK. [11]Present address: Department of Frontier Research and Development, Kazusa DNA Research Institute, Kisarazu, Japan. [12]Present address: Department of Biophysical Chemistry, Kyoto Pharmaceutical University, Kyoto, Japan. [13]Present address: HOLO BIO Co., Ltd, Kyoto, Japan. [14]Present address: Whitehead Institute for Biomedical Research, Cambridge, MA, USA. ✉E-mail: yhattori@lif.kyoto-u.ac.jp; tauemura@lif.kyoto-u.ac.jp

environments in early life? Second, what are the quantifiable responses in the juveniles to such effective diets in terms of genome-wide gene expression and metabolism? Third, is a particular nutrition history, or a particular set of the juveniles' responses, causative of the long-term effects in adult animals, and how can we demonstrate such a hypothetical cause-and-effect relationship? Fourth, which cells or tissues actually retain the history and/or transmit the history to the later life stage?

To tackle the above questions, we set out to address the effects of specific dietary environments during the entire developmental stages on the subsequent adult lifespan using *Drosophila melanogaster* (*D. melanogaster*). Research with *D. melanogaster* has contributed insights into nutrient-responsive mechanisms governing development and aging, which are highly conserved among animal species (Droujinine and Perrimon, 2016; Mattila and Hietakangas, 2017; Miguel-Aliaga et al, 2018; Piper and Partridge, 2018; Texada et al, 2020). Opportunely, budding yeast, *Saccharomyces cerevisiae* (*S. cerevisiae*), is one of the major ingredients of laboratory foods for *D. melanogaster*, supplying larvae with numerous nutrients, including critical ones for development, such as sterols (Bos et al, 1976). It has been reported that a restriction of the yeast in the diet only in the late larval stage is associated with changes in adult reproduction, but not with lifespan (Tu and Tatar, 2003). On the other hand, lifespan is adversely affected by a large amount of autoclaved yeast throughout the larval stages, which increases the production of alkene hydrocarbons, named autotoxins, in the adult (Stefana et al, 2017). In addition, how yeast is treated, either live, dried, or heat-treated, in preparing larval diets affects adult life traits including food preference and mating behavior (Grangeteau et al, 2018). However, key metabolites in yeast were not explored. Other earlier works showed that collections of single-gene knockout (KO) strains of bacteria could serve as nutritionally variable diets for *D. melanogaster* or *C. elegans* during development, and many of those strains have the potential, or have been shown, to differentially affect gene expression, metabolism, and development of the animals (Shin et al, 2011; Watson et al, 2014). The single-gene KO collection of yeast (Winzeler et al, 1999) is also expected to be a source of a variety of diets, because deletion of over a third of protein-coding non-essential genes changes the amino acid compositions (Mülleder et al, 2016).

Here, following the above-mentioned leads, we established an interspecies assay using *D. melanogaster* with *S. cerevisiae* (the "live yeast–fly" assay). We imposed dietary interventions throughout the entire larval stages of *D. melanogaster* using the KO collection of yeast, and then quantified the lifespans of eclosed adults on the common standard food to assess how the larval diets affect the adult health (Fig. 1A). Our screen identified a *nat3* (*nat3Δ*) KO yeast strain that shortened the lifespan of *Drosophila* males compared to the parental strain (the control yeast). In the *nat3Δ*-fed larvae, transcriptomic and epigenomic analyses showed that the molecular function of a histone acetyltransferase, General control non-repressible 5 (Gcn5), was diminished. Whole-body partial knockdown of *Gcn5* selectively in larval stages did not affect the total number of emerging adults, but their lifespan was shortened, strongly suggesting that Gcn5-mediated histone acetylation during larval development impacts adult lifespan. Our further tissue-specific analyses suggest that dysfunction of Gcn5 in larval neurons contribute to the earlier death of adults.

The *nat3Δ* yeast was much more abundant in long-chain fatty acids and branched-chain amino acids (BCAAs) than the control yeast. We explored supplementation of the control yeast diet with nutrients individually or in combination during larval development to identify which ones shortened adult lifespan; we found that the effective combination was oleic acid (a representative LCFA in yeast), valine, and acetic acid. Our data support the proposition that the high intake of these nutrients during larval development shortens adult lifespan via their adverse effect on Gcn5 function.

## Results

### A *nat3Δ* yeast diet in larval stages shortens lifespan of males

To generate adults with a wide range of distinct nutrition histories, we established the live yeast–fly assay (Figs. 1A–H and EV1). In each tube, a single-gene KO yeast strain was cultured on a modified synthetic complete medium (mSCM); next, germ-free embryos were placed on top of the cultured yeast (Fig. 1B). Hatched larvae started eating both the live yeast and mSCM, and developed into pupae. Importantly, no larvae pupariated on mSCM without yeast, so larval development critically depended on yeast-derived nutrients (see details in "Live yeast–fly assay" in "Methods").

Of a total of 5153 yeast KO strains, 46 altered the timing and/or the rate of pupariation compared to the control yeast (Fig. 1C; Dataset EV1). Thus, the larval diets comprised of these 46 strains generated appreciable impacts on larval development, and we anticipated that any of these diets could generate long-lasting nutrition histories, which might affect the adult lifespan. We monitored the development of larvae to adults following larval growth on each of the 46 diets, noting the timing and the rate of adult emergence, and found that 29 of the 46 KO yeast strains allowed larvae to produce sufficient numbers of adults for the lifespan assay (Fig. 1D; Appendix Fig. S1; Dataset EV1). After narrowing down the candidates of interest to 29 yeast strains, all of our subsequent live yeast–fly assays employed *white Dahomey* (*wDah*), a *D. melanogaster* stock that is frequently used in lifespan assays (Grandison et al, 2009). Among the 29 yeast KO larval diets, two yeast strains (*nat3Δ* and *brp1Δ*) affected the lifespan of male adults when they were aged on the standard laboratory food (Fig. 1C; Appendix Fig. S2; Datasets EV1 and EV2), indicating that those 2 nutrition histories must have unique features compared to the remaining 27.

One of them was a *nat3Δ* mutant, which shortened the lifespan of males (Fig. 1E; 8.5% reduction in the median lifespan). In males, this long-term effect of the *nat3Δ* diet was reproducible (Fig. 1F; 16% reduction in the median lifespan; see also Dataset EV2). On the other hand, the same larval diet did not affect female lifespan (Fig. EV1E). Accordingly, we focused on males to investigate the underlying mechanism, although we describe the effects of dietary and genetic interventions on females in Fig. EV6 (described later in "Does the dietary or knockdown intervention impact female lifespan?"). In addition to the shorter lifespan, *nat3Δ*-fed larvae and *nat3Δ*-history adults gained significantly more weight than the control larvae and adults, respectively (Fig. 1G,H). Furthermore, adults that possessed the *nat3Δ*

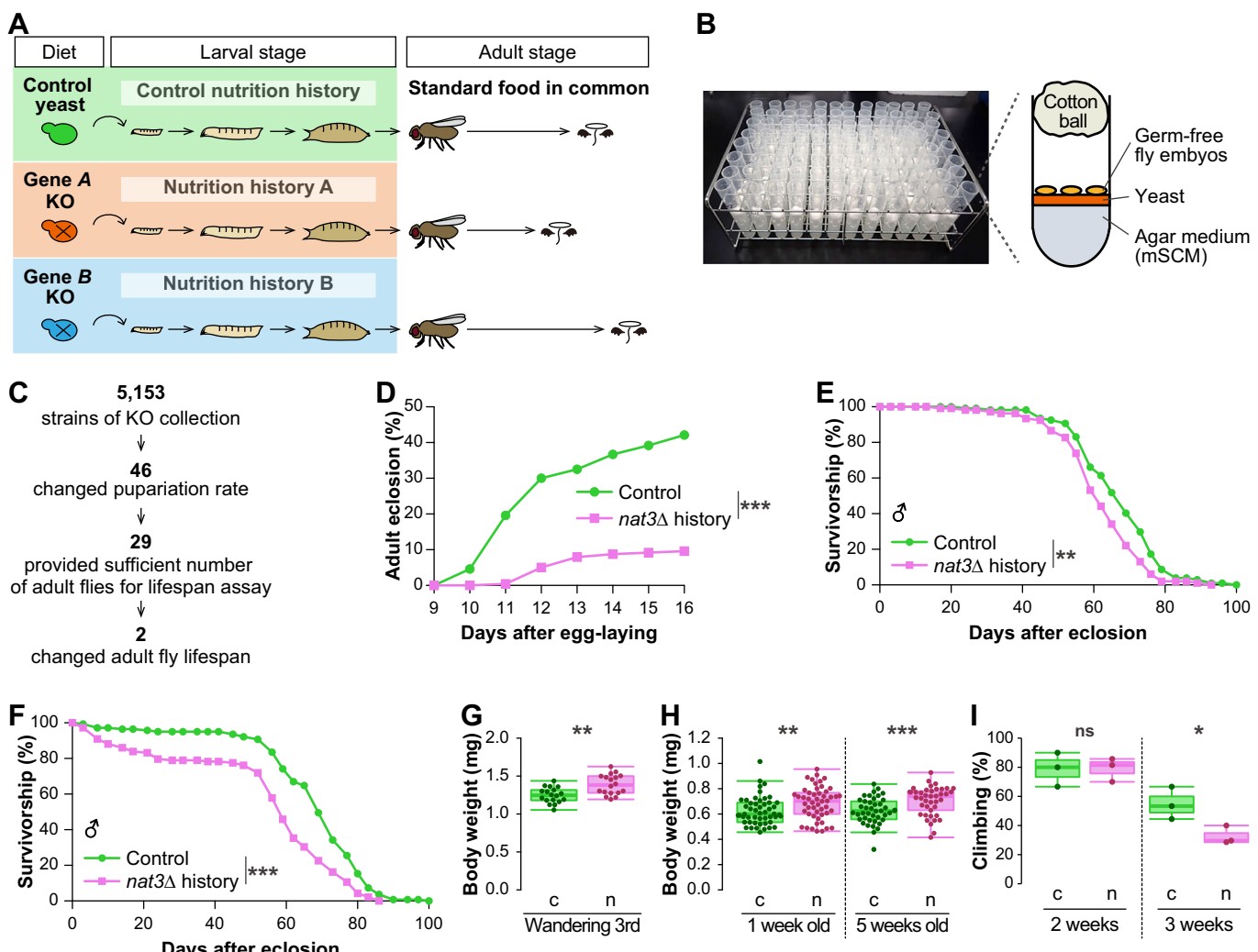

**Figure 1. The live yeast–fly assay and the long-term effect of the *nat3Δ* yeast diet in larval stages on the lifespan of male adults.**

(A) Designs of the live yeast–fly assay and subsequent measurements of lifespan. Larvae were fed on a control yeast strain or individual single-gene knockout yeast strains, such as Gene *A* KO and Gene *B* KO. Eclosed adults with variable nutrition histories were collected and then fed on the common standard laboratory food. Ultimately, the lifespans of those adults were recorded. See some representative characterizations of the live yeast–fly assay in Fig. EV1A–D. (B) A photo of the yeast–fly tubes in a rack. In each tube, one yeast strain was cultured on a modified synthetic complete medium for yeast (mSCM). Then, about thirty germ-free embryos of *white Dahomey* (*wDah*) were placed on top of the cultured yeast and the tube was plugged with an autoclaved cotton ball. (C) Summary of our screen from the start, with 5153 yeast strains, concluding with the isolation of two strains that changed the male adult lifespan. See details of how we isolated the 46 and the 29 yeast strains in "Methods", Appendix Fig. S1 and Dataset EV1. (D–I) Effects of the *nat3Δ* diet on adult eclosion (D) and male lifespan (E, F), body weight (G, H) and adult climbing ability (I). (D) Adult eclosion percentage was calculated from the daily number of eclosed adult flies of both males and females. Throughout this study, both males and females were counted for calculations of adult eclosion percentages (male:female ≈ 1:1). (E, F) Survival curves of males with the control or *nat3Δ* nutrition history. *nat3Δ*-history males died sooner than the control males in all of 10 independent experiments including those shown in (E, F). The panel (E) data is a part of the data of Appendix Fig. S2 Round 1. (G) Body weight of individual control-fed or *nat3Δ*-fed wandering third-instar male larvae ("c" or "n"). Each data point represents the average body weight of 13–15 larvae. (H) Body weight of control-history or *nat3Δ*-history male adults of two ages ("c" or "n"). Each data point represents the weight of a single adult. (I) The total number of control male adults (green, "c") was 30 (3 vials) and those of *nat3Δ* history (pink, "n") was 26 (3 vials) at each age. Boxplots are depicted as in "Statistical analysis" in "Methods". *P < 0.05, **P < 0.01, ***P < 0.001. The exact P values, sample sizes (N), and statistical tests employed are listed in Dataset EV12. Source data are available online for this figure.

nutrition history exhibited an earlier decline in motor performance compared to the control adults when measured at timepoints of 100% survival (Fig. 1I).

The yeast *nat3* gene encodes the catalytic subunit of protein N-terminal acetyltransferase B (NatB; Polevoda et al, 1999). Molecular functions of N-terminal acetylation of substrate proteins are diverse, including regulation of degradation, complex formation and subcellular localization (Aksnes et al, 2016). Almost 200 confirmed or putative target proteins of yeast NatB have been reported (Caesar and Blomberg, 2004; Caesar et al, 2006; Helbig et al, 2010; Van Damme et al, 2012; Croft et al, 2018) and *nat3Δ* yeast exhibit pleiotropic phenotypes, including altered regulation of nicotinamide adenine dinucleotide metabolism (Wilson et al, 2002; Caesar et al, 2006; Croft et al, 2018; Sugaya et al, 2023).

## Gene expression profiles are strikingly similar between nat3Δ-fed larvae and larvae with mutations in the histone acetyltransferase gene Gcn5

We assumed that the nat3Δ diet evoked some specific responses in larvae, which could not be restored on the standard food, even much later in adult life. To unveil such hypothetical responses and identify the key nutrients in the diet, we conducted metabolomic analysis of the yeast strains and whole-body RNA sequencing (RNA-seq) and metabolomic analyses of the following Drosophila male samples (Fig. 2A; Datasets EV3–5): wandering 3rd-instar larvae that were raised on the nat3Δ diet or on the control diet (designated as the nat3Δ-fed larvae or the control larvae hereafter), and young (1 week old) or midlife adults (5 weeks old) that had been subjected to the nat3Δ nutrition history or to the control history in their larval stages (designated as nat3Δ-history adults or control adults hereafter).

Our RNA-seq datasets of male larvae showed that the number of differentially expressed genes between the nat3Δ-fed larvae and the control larvae was 1120, which decreased to about 200 in the young adults and the midlife adults (Fig. 2A; Dataset EV4). We discovered two key features of the larval RNA-seq data. One of them was that upregulated or downregulated genes between the nat3Δ-fed larvae and the control larvae exhibited a striking similarity to those between Gcn5 or Ada2a complete loss-of-function mutant larvae and their control genotypes (Fig. 2B–E; see also Appendix Fig. S3A and details in its legend regarding the rationale for analyzing the Gcn5 data; Carré et al, 2008). Gcn5 is a founding member of the histone acetyltransferase family, and acetylation of histone tails decondenses chromatin, thereby allowing access to transcription factors and co-activators (Brownell et al, 1996; Kuo et al, 1996; Verdin and Ott, 2015). In particular, histone H3 lysine 9 acetylation (H3K9ac) is commonly associated with active transcription (Verdin and Ott, 2015), and H3K9 is a known target of Gcn5 (Carré et al, 2005; Feller et al, 2015). One of the Gcn5-containing complexes is termed "ATAC", which is conserved from yeast to human, and its formation is nucleated by Ada2a protein (Torres-Zelada and Weake, 2021; Dent, 2024). Clustering heatmaps showed that the genes significantly upregulated and down-regulated ("Up" genes and "Down" genes) in the nat3Δ-fed larvae tended to be up and down, respectively, in Gcn5 or Ada2a mutant larvae (Fig. 2B,D). Likewise, Venn diagrams indicated that the Up and Down genes in the nat3Δ-fed larvae overlapped with the Up and Down genes in the Gcn5 or Ada2a mutants, respectively, in a highly correlated manner (Fig. 2C,E).

Since we found the above striking similarity of gene expression profiles between the nat3Δ-fed larvae and the Gcn5 or Ada2a mutant larvae, we asked whether this similarity is unique to the Gcn5 or Ada2a mutant. For this comparative purpose, we sought larval transcriptome data of mutants deficient in other histone modifying enzyme genes. We found microarray data for double mutant larvae affecting the Kdm4A and Kdm4B genes encoding histone lysine demethylase 4 (Tsurumi et al, 2019; designated as Kdm4 mutant hereafter) and compared the gene expression of the Kdm4 mutant larvae with that of the nat3Δ-fed larvae (Fig. 2F,G). Both the heat map and the Venn diagram indicated that there were only small or uncorrelated overlaps between Up or Down genes in the Kdm4 mutant larvae and the Up or Down genes in the nat3Δ-fed larvae. To measure the degree of agreement between the diet effect and the genotype effect on gene expression, we employed Cohen's kappa (Cohen, 1960; Landis and Koch, 1977). The

coefficient, κ, was 0.92 and 0.87 for Gcn5 and Ada2a, respectively, indicating strong agreement; in sharp contrast, κ, was 0.015 for Kdm4. This analysis implies that the nat3Δ diet may affect the function of a group of select proteins, including Gcn5, in larval cells.

All of the above observations prompted us to test whether the nat3Δ diet impaired larval Gcn5 function. Between the larvae raised on the two distinct yeast diets, there were no significant differences in transcript quantities of any subunit genes including Gcn5 and Ada2a (Fig. EV2A; see also its legend). Therefore, we hypothesized that Gcn5 and other subunit proteins were present in similar amounts in larval cells irrespective of the diets, and that the nat3Δ diet impaired the function of the ATAC complex containing Gcn5.

## Genome-wide profiling of H3K9 acetylation strongly suggests that the nat3Δ diet diminishes Gcn5 function

To test the hypothetical connection between the nat3Δ diet and the diminished Gcn5 function, we performed genome-wide profiling of H3K9ac in whole-body male larvae using CUT&RUN (Fig. 3; Dataset EV6; Skene and Henikoff, 2017). We acquired CUT&RUN H3K9ac data from two distinct classes of larval samples (Fig. 3A). First, to examine the differential effect between the two yeast diets on the H3K9ac profile (the dietary effect; Fig. 3A, left), we reared wDah larvae on the control yeast diet or on the nat3Δ diet (designated together as "the yeast-fed samples"). Second, because we anticipated that the nat3Δ-fed larvae partially lost Gcn5 function, we examined the effect of a Gcn5 weak knockdown on the H3K9ac profile (Gcn5 KD effect; Fig. 3A, right). Gcn5 was broadly knocked down in tissues, using a GeneSwitch (GS) driver stock, daughterless GS (DaGS), which allows drug-induced ubiquitous expression of a hairpin RNA (Osterwalder et al, 2001; Roman et al, 2001; Tricoire et al, 2009; Yamashita et al, 2021). We reared larvae with Gcn5 KD genotypes on the standard food without or with the RNAi-inducing drug, RU486 (referred to as "Gcn5 KD−" or "Gcn5 KD + " hereafter, and designated together as "the Gcn5 KD samples", as depicted in Fig. 3A, right). Our RNAi induction condition yielded a weak knockdown of Gcn5, as evidenced by a reduction of the amount of Gcn5 transcripts to 78% of control levels (described later in Fig. 4B).

We found a total of 12,102 H3K9ac peaks and a total of 8419 peaks in the yeast-fed samples and in the Gcn5 KD samples, respectively. We then viewed differential peaks between the yeast-fed samples and between the Gcn5 KD samples (shaded regions in Fig. 3B). Among the 12,102 H3K9ac peaks in the yeast-fed samples, 798 (6.6%) were altered in the nat3Δ-fed larvae, and 786 of those peaks were lower than in the control (blue in Fig. 3C). With the Gcn5 KD, 1233 out of the 8419 peaks (15%) were altered, and the vast majority of those peaks were reduced relative to the control (1203 out of 1233; blue in Fig. 3D). Comparing the dietary effect and the Gcn5 KD effect, we quantified the overlap of the overall length of the Down peaks as well as the overlap of Down peak-containing genes. Out of 890 kb of the nat3Δ diet-induced Down peak intervals, 97 kb (11%) were also reduced by the Gcn5 KD (Fig. 3E). Likewise, out of the 796 genes with reduced H3K9ac occupancy with the nat3Δ diet, 220 genes (28%) overlapped the Gcn5-KD-induced Down peak-containing genes (Fig. 3F). There-fore, both the nat3Δ diet and the weak Gcn5 KD share the feature of

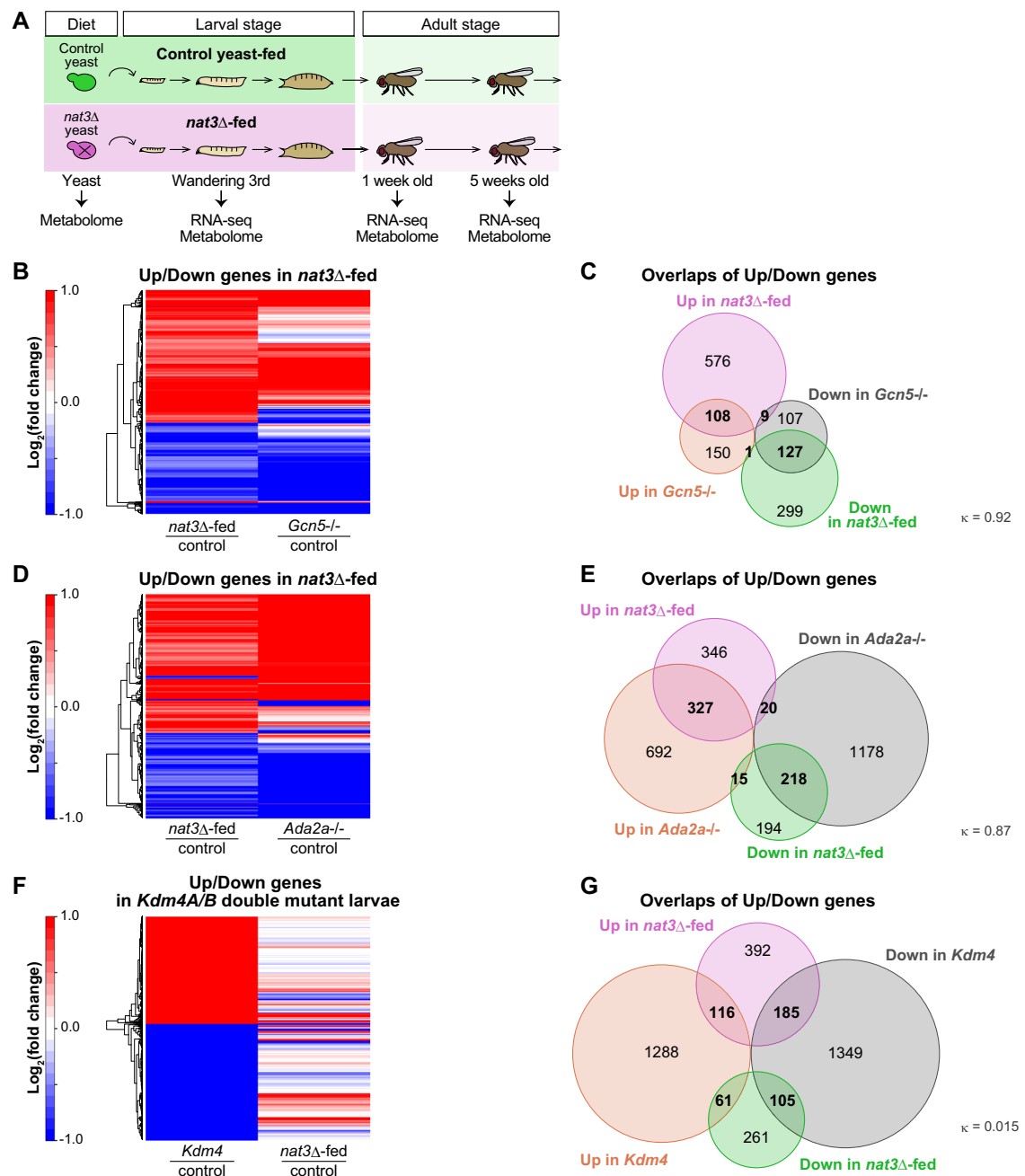

**A** Diet / Larval stage / Adult stage

Control yeast-fed — Control yeast
nat3Δ-fed — nat3Δ yeast

Yeast → Metabolome
Wandering 3rd → RNA-seq, Metabolome
1 week old → RNA-seq, Metabolome
5 weeks old → RNA-seq, Metabolome

**B** Up/Down genes in nat3Δ-fed

**C** Overlaps of Up/Down genes
Up in nat3Δ-fed: 576
108
9
Down in Gcn5-/-: 107
Up in Gcn5-/-: 150
1
127
Down in nat3Δ-fed: 299
κ = 0.92

**D** Up/Down genes in nat3Δ-fed

**E** Overlaps of Up/Down genes
Up in nat3Δ-fed: 346
327
20
Down in Ada2a-/-: 1178
Up in Ada2a-/-: 692
15
218
Down in nat3Δ-fed: 194
κ = 0.87

**F** Up/Down genes in Kdm4A/B double mutant larvae

**G** Overlaps of Up/Down genes
Up in nat3Δ-fed: 392
116
185
Down in Kdm4: 1349
Up in Kdm4: 1288
61
105
Down in nat3Δ-fed: 261
κ = 0.015

partially reducing H3K9ac peaks, suggesting that the *nat3Δ* diet diminishes Gcn5 function in larvae.

In a *D. melanogaster* cell line, a strong enrichment of H3K9ac is seen from the transcription initiation site (TSS) up to 500 bp downstream of expressed genes (Kharchenko et al, 2011), whereas it is located throughout the bodies of transcribed genes in adults (Yin et al, 2011). Consistent with these reports indicating that H3K9ac is generally associated with active transcription, 7930 (86%) of the total 9199 peak-containing genes of the yeast-fed samples are in fact expressed in larvae (Fig. EV2B). When we focused on the Down peaks in the *nat3Δ*-fed larvae and those in the *Gcn5* KD larvae, both groups of peaks shared a feature that the

H3K9ac marks were localized around the TSS, with a stronger enrichment up to 500 bp downstream (Fig. EV2C).

## Whole-body partial knockdown of *Gcn5* only in larval stages shortens adult lifespan

Since the larval *nat3Δ* diet-induced reduction in adult lifespan implicated Gcn5 function as a key determinant, we asked whether a partial knockdown of *Gcn5* during larval development would similarly shorten the lifespan of adults. Thus, we sought to measure the lifespan of males with a *Gcn5* KD history. As described in the previous section, this approach was realized by using the *GS* stocks

**Figure 2. Striking similarity of gene expression profiles between the *nat3Δ*-fed male larvae and larvae with a *Gcn5* histone acetyltransferase gene mutation.**

(A) Experimental designs for omics of the two yeast strains (the control strain and *nat3Δ*, colored green and pink, respectively), the two respective yeast-fed wandering third-instar larvae, and adults of the indicated ages that were endowed with the respective nutrition histories. The larvae and adults were reared as in Fig. 1A,B. The number of differentially expressed genes between the control yeast-fed larvae and the *nat3Δ*-fed male larvae was 1120 (693 up and 427 down in *nat3Δ*-fed), which decreased to 223 in the young male adults (121 up and 102 down in *nat3Δ* history) and 165 in the midlife male adults (39 up and 126 down in *nat3Δ* history). (B–G) Comparison of our RNA-seq data from the yeast-fed male larvae, the microarray data from *Gcn5* or *Ada2a* mutant larvae (Carré et al, 2008) and the microarray data of double mutant larvae of *Kdm4A* and *Kdm4B* (Tsurumi et al, 2019; designated as *Kdm4* mutant). These microarray data were collected presumably from larvae of both the sexes. *Kdm4* gene encodes histone lysine demethylase 4, and Kdm4A and Kdm4B reverse tri-methylation of H3K9 and H3K36 (Lloret-llinares et al, 2008). (B, D, F) Heatmaps showing whether each of significantly upregulated (Up) and downregulated (Down) genes in the *nat3Δ*-fed male larvae (red and blue, respectively, in "*nat3Δ*-fed|control") tends to be up or down in the *Gcn5* or *Ada2a* mutant larvae (red or blue in "*Gcn5-/-*|control" in (B) and in "*Ada2a-/-*|control" in (D)). Because the available microarray data of the *Kdm4* mutant larvae contains only Up and Down genes in the mutant, panel F shows whether each of the Up and Down genes in the mutant (red and blue, respectively in "*Kdm4*|control") tends to be up or down in the *nat3Δ*-fed male larvae (red or blue in "*nat3Δ*-fed|control"). (C, E, G) Venn diagrams showing overlaps between the Up or Down genes in the *nat3Δ*-fed male larvae and the Up or Down genes in the *Gcn5* mutant larvae (C), *Ada2a* mutant larvae (E) or *Kdm4* mutant larvae (G). The numbers of genes in the individual categories are indicated. Changes in gene expression were highly correlated between the *nat3Δ*-fed male larvae and the *Gcn5* mutant larvae, and also between *nat3Δ*-fed male larvae and the *Ada2a* mutant larvae [$\kappa_{Gcn5} = 0.92$ (95% CI: 0.87–0.97) and $\kappa_{Ada2a} = 0.87$ (95% CI: 0.83–0.92); Cohen's kappa coefficient]. However, the correlation was low between the *nat3Δ*-fed male larvae and the *Kdm4* mutant larvae [$\kappa_{Kdm4} = 0.015$ (95% CI: −0.063–0.094)]. We prepared triplicates for the respective conditions of this RNA-seq analysis (Dataset EV4). Source data are available online for this figure.

for drug-induced RNAi (Osterwalder et al, 2001; Roman et al, 2001). We fed larvae on the standard food with the drug (*Gcn5* KD + ) or on the food without the drug (*Gcn5* KD-) (Yamashita et al, 2021), collected adult flies that had different histories of RNAi, and measured their lifespans on the no-drug food (Fig. 4A).

Under our experimental condition, the amount of *Gcn5* transcript was reduced to 78% in the *Gcn5* KD+ larvae compared to *Gcn5* KD− larvae, whereas *Gcn5* transcript levels in the adults were not significantly different whether *Gcn5* was knocked down in the larval stages or not (Fig. 4B,C). Our *Gcn5* KD condition in larvae resulted in a 1-day delay in the timing of the adult emergence, but it did not reduce the total number of emerging adults (Fig. 4D,E). It has been reported that the development of *Gcn5* null mutant larvae takes twice as long as that of the wild-type, and it is arrested at the larval–pupal transition (Carré et al, 2005). Considering these null mutant phenotypes, our *Gcn5* KD condition genetically mimics a partial loss of function. Adults with the *Gcn5* KD+ history died notably earlier (32% reduction in the median lifespan) compared to the adults of the same genotype with the *Gcn5* KD− history (Fig. 4F,G). Our overall findings showed that reduced Gcn5 function during larval development resulted in a shorter adult lifespan. As described in the previous section, we compared the *nat3Δ* dietary effect and the *Gcn5* KD effect on genome-wide H3K9ac profiling (Fig. 3C–F). The median lifespan was shortened to a greater degree by the KD than the *nat3Δ* diet compared to individual controls (compare Fig. 4G with Fig. 1E,F), and this differential effect may reflect the broader impact of the *Gcn5* KD on H3K9 acetylation in the whole genome (Fig. 3C,D).

## Defects in *Gcn5*-modulated gene regulation in larval neurons impact adult lifespan and gene expression in the adult brain

*Gcn5* is broadly expressed in larval tissues including the central nervous system and the intestine (Brown et al, 2014; Leader et al, 2018). We wondered whether a *Gcn5* KD in a specific larval cell type or tissue would shorten the adult lifespan. To explore several key cell types or tissues, we conducted larval stage-specific *Gcn5* KDs in pan-neurons, the intestine plus adipose tissue, and the Malpighian tubule that is functionally related to the kidney, using available *GS* stocks (*ElavGS*, *TIGS*, and *UroGS*, respectively;

Fig. 4H–N; see Appendix Fig. S3B,C for confirmation of the *ElavGS* driver and the *Gcn5* partial knockdown, respectively; Osterwalder et al, 2001; Yamashita et al, 2021). Of the three stocks tested, *Gcn5* KD in pan-neurons dramatically shortened the lifespan of males (Fig. 4L; 39% reduction in the median lifespan), similar to the effect of the whole-body *Gcn5* KD (Fig. 4G). Furthermore, 2-week-old adults with the *Gcn5* KD history in larval neurons exhibited a severe decline in motor performance (Fig. 4K), similar to the *nat3Δ*-history adults (Fig. 1I). The *Gcn5* KD in larval neurons did not lead to a decrease in the number of emerging adults, but the timing was slower (Fig. 4H), similar to the whole-body KD (Fig. 4E). Although our analysis was limited in terms of the diversity of the cell types tested, the developmental and aging effects caused by the pan-neuronal *Gcn5* KD phenocopied the whole-body *Gcn5* KD.

Given that Gcn5 function in larval neurons is critical for adult lifespan and many larval neurons persist in the adult brain, we inferred that the *nat3Δ* diet in larval stages might exert long-term effects on gene expression in the adult brain. To test this hypothesis, we examined gene expression in the brain of the *nat3Δ*-history adult at 3 weeks of age, before the survivorship was decreased. We found that the *nat3Δ* diet had far-reaching consequences, as described below (Fig. 4O). In total, 8941 genes were expressed in the brains of the control adult and the *nat3Δ*-history adult. In this gene set, the expression of 8309 genes was unaffected by the larval diet ("Not changed," the left top) and the vast majority of these unaffected genes were already expressed in the larval CNS (7728/8309 "Already expressed in larval CNS," the right-top bar). In the adult brain, the expression of the remaining 632 genes was affected by the nutrition history, being either lower or higher in the *nat3Δ*-history adults (the green or pink bar in the left). Interestingly, more than 50% of those 632 genes comprised genes that were not expressed in the larval CNS; however, they were induced at some time after pupariation (329/632; "Not expressed in larval CNS" in the right-bottom bar). Such a gene enrichment indicates that the impact of the *nat3Δ* diet on neural gene expression extended well beyond the stage of that food intake (the larval stage) and had a profound effect on the onset of gene expression in the later pupal and adult stages. Finally, we hypothesized that the *nat3Δ* diet-induced reduction in Gcn5 function led to abnormal expression patterns of a group (or

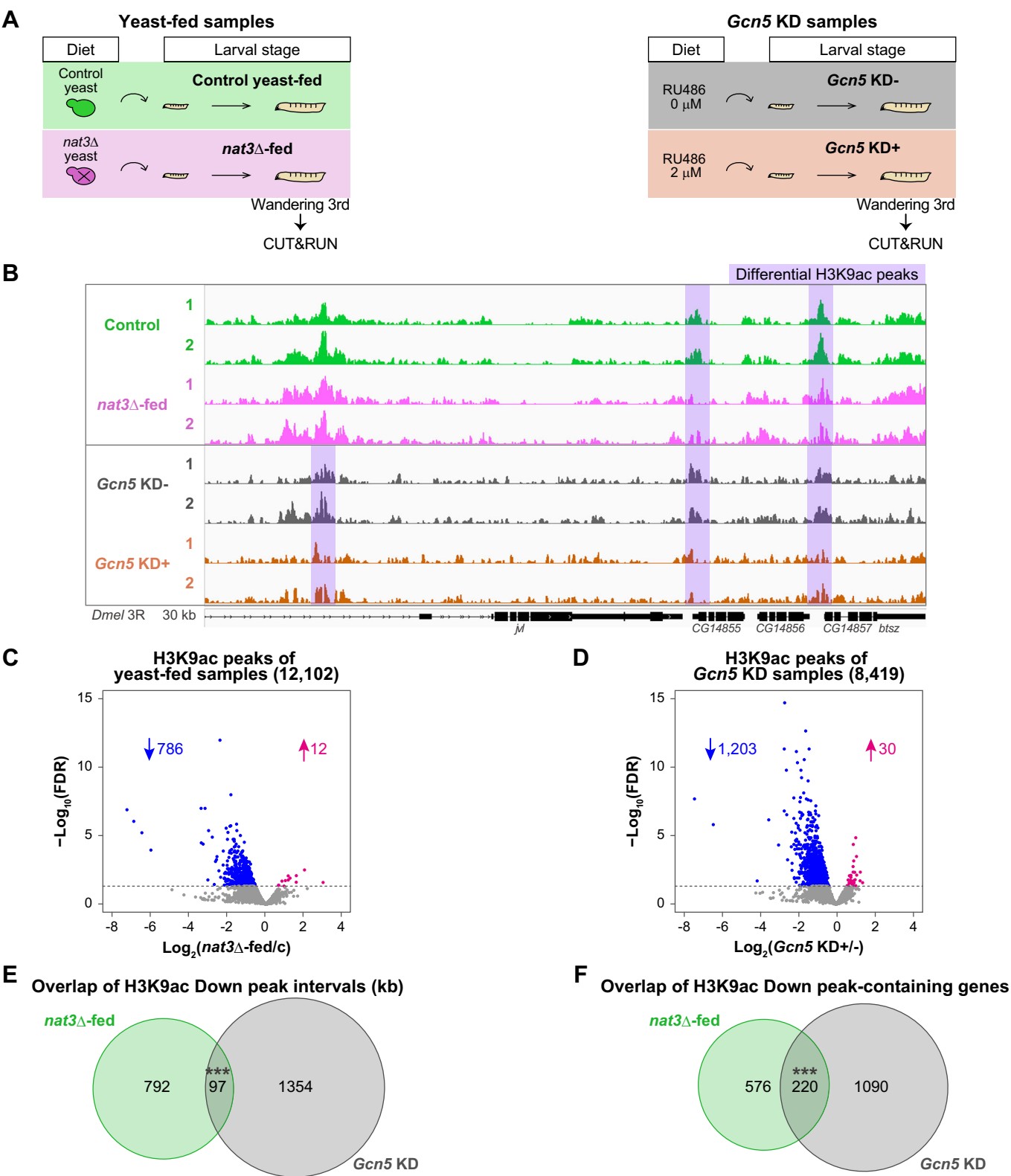

**A**

**Yeast-fed samples**

| Diet | Larval stage |
| --- | --- |
| Control yeast | **Control yeast-fed** |
| *nat3∆* yeast | ***nat3∆*-fed** |

↓ Wandering 3rd
↓ CUT&RUN

***Gcn5* KD samples**

| Diet | Larval stage |
| --- | --- |
| RU486 0 µM | ***Gcn5* KD-** |
| RU486 2 µM | ***Gcn5* KD+** |

↓ Wandering 3rd
↓ CUT&RUN

**B**

Differential H3K9ac peaks

Control 1, 2
*nat3∆*-fed 1, 2
*Gcn5* KD- 1, 2
*Gcn5* KD+ 1, 2

*Dmel* 3R    30 kb

*jM*    CG14855    CG14856    CG14857  *btsz*

**C**    H3K9ac peaks of yeast-fed samples (12,102)

−Log$_{10}$(FDR)

↓786    ↑12

Log$_2$(*nat3∆*-fed/c)

**D**    H3K9ac peaks of *Gcn5* KD samples (8,419)

−Log$_{10}$(FDR)

↓1,203    ↑30

Log$_2$(*Gcn5* KD+/-)

**E**    Overlap of H3K9ac Down peak intervals (kb)

*nat3∆*-fed    *Gcn5* KD

792    97*** 1354

**F**    Overlap of H3K9ac Down peak-containing genes

*nat3∆*-fed    *Gcn5* KD

576    220*** 1090

**Figure 3.   Genome-wide profile of H3K9 acetylation in the *nat3Δ*-fed male larvae and its similarity to that of *Gcn5* KD male larvae.**

(A) Experimental designs for preparing wandering 3rd-instar male larvae of "Yeast-fed samples" and "*Gcn5* KD samples" for CUT&RUN analysis. (Left) The larvae of "Yeast-fed samples" (Control and *nat3Δ*-fed) were reared as in Fig. 1B to examine the dietary effect. (Right) "*Gcn5* KD samples" ("*Gcn5* KD−" and "*Gcn5* KD +") were prepared to examine the effect of knocking down *Gcn5* broadly in larvae. Larvae with a *DaGS* driver and a short-hairpin RNA construct were reared on the standard food without or with an RNAi-inducing drug, RU486. See details in "Methods". (B) Local views illustrating acetylation of histone H3K9 (H3K9ac) in a 30 kb genomic region. The four conditions are labeled as in (A), and two replicates per condition are shown. Shaded regions show differential peaks between the dietary conditions (Control vs. *nat3Δ*-fed) and/or between the *Gcn5* KD conditions (*Gcn5* KD− vs. *Gcn5* KD +). (C, D) Volcano plots showing how H3K9ac peaks are altered between the two dietary conditions (C) and between the *Gcn5* KD conditions (D). A total of 12,102 H3K9ac peaks were identified in the yeast-fed samples (C), whereas 8419 peaks were identified in the *Gcn5* KD samples (D). Reduced and increased peaks are colored blue and magenta, respectively. The numbers of peaks for individual categories are indicated. (E) A Venn diagram showing the overlap of the H3K9ac Down peaks between the dietary conditions and those between the *Gcn5* KD conditions. The cumulative peak intervals (kb) for individual categories are indicated. The overlap (97 kb of peak intervals) was significant. See details in Dataset EV6. (F) A Venn diagram showing the overlap of the Down peak-containing genes between the dietary conditions and those between the *Gcn5* KD conditions. The numbers of peaks for individual categories are indicated. The overlap was significant (220 genes: 28% of the *nat3Δ*-fed down peak-containing genes). We prepared H3K9ac duplicates and an IgG replicate for the respective conditions of this CUT&RUN analysis as explained in "Methods". *$P < 0.05$, **$P < 0.01$, ***$P < 0.001$. The exact $P$ values, sample sizes and statistical tests employed are listed in Dataset EV12. Source data are available online for this figure.

groups) of genes in larval neurons, which may be an indirect cause of the lifespan shortening, and we attempted to verify this hypothesis (Appendix Fig. S3D–K).

## The *nat3Δ* yeast diet is much more abundant in long-chain fatty acids (LCFAs) and branched-chain amino acids (BCAAs) than the control yeast diet

We have so far highlighted the *Gcn5* mutant-like transcriptome profile of the *nat3Δ*-fed male larvae. In addition, we found another key feature of the larval RNA-seq dataset, which hints at tailored metabolic responses to the higher intake of two classes of nutrients in the *nat3Δ* diet. First, we became interested in fatty acids, because enriched GO terms in the upregulated genes in the *nat3Δ*-fed larvae included "Fatty acid degradation" and "Fatty-acyl-CoA binding" (Appendix Fig. S4). Notable representatives of these upregulated genes encode major enzymes of fatty acid β-oxidation (Fig. EV3A–E). All these respective increases in gene expression in the *nat3Δ*-fed larvae led us to speculate that this response could be adaptive to a much higher intake of fatty acids and commensurate increases in the amounts of fatty acyl-CoA species, compared to the control larvae (described later in Fig. 7G).

To verify the aforementioned possibility, we compared the abundance of fatty acids (both esterified and non-esterified ones) between the two yeast strains. Indeed, the *nat3Δ* yeast contained dramatically larger amounts of C16 and C18 fatty acids than the control yeast (Fig. 5A, top; Dataset EV8A). These are classified as long-chain fatty acids (LCFAs; $13 \leq C \leq 19$ or $13 \leq C \leq 21$) and account for more than 95% of total fatty acids in yeast (Kaneko and Ito, 1971). Three major species of the C16 and C18 fatty acids in yeast are palmitic acid (16:0), oleic acid (18:1n-9), and stearic acid (18:0), all of which were increased by more than 5-fold. Moreover, some very long-chain fatty acids ($C \geq 20$ or $C \geq 22$) were also increased in the *nat3Δ* yeast by more than tenfold (Fig. 5A, bottom; Dataset EV8A). Taken together, the *nat3Δ* diet is much more abundant in these fatty acids.

Second, we noticed that the *nat3Δ*-fed larvae had increased expression of genes encoding enzymes of branched-chain amino acid (BCAA) catabolism, which partially overlap with enzymes of fatty acid β-oxidation in that they yield CoA-related metabolites (Fig. EV3F,G; Adeva-Andany et al, 2017; Mann et al, 2021). The upregulated genes unique to the BCAA catabolism included the gene encoding branched-chain aminotransferase (BCAT). Indeed,

BCAAs tended to be more abundant in the *nat3Δ* yeast; and among them, the increase in the amount of valine was the most significant (Fig. 5B; Dataset EV3). Our findings are reminiscent of a recent report that feeding *Drosophila* female adults on a diet with a higher BCAA content increases *Bcat* expression in the head (Weaver et al, 2023).

## The control diet for larvae when supplemented with 3 nutrients (OVA) shortens adult lifespan, just as the *nat3Δ* diet does

Following our observation of elevated LCFAs and BCAAs in the *nat3Δ* yeast, we addressed whether the abundant LCFAs and BCAAs in the *nat3Δ* yeast were the main cause of the long-term effect. For this purpose, we supplemented the control yeast diet with oleic acid, valine, and isoleucine separately or in combination, and monitored whether any of the supplemented diets affected both the development (the timing and the rate of adult eclosion) and adult lifespan, similar to the *nat3Δ* yeast diet. In the course of cooking these supplemented diets, we contrived recipes to uniformly mix oleic acid with the other ingredients and determined appropriate working concentrations of the individual additives (Fig. EV4A–D; see details in the legends and "Live yeast–fly assay" in "Methods"). Under the supplementation conditions employed here, neither oleic acid alone, BCAAs alone, nor their combination shortened adult lifespan significantly, although oleic acid alone and the combination were potent enough to delay the timing and reduce the eclosion rate (Figs. 5C–F and EV4C–J).

We suspected that increased or decreased amounts of other nutrients would be further necessary to affect adult lifespan. In our RNA-seq data, the expression of *Acetyl-coenzyme A synthase* (*AcCoAS*) was increased in the *nat3Δ*-fed larvae relative to the control larvae and AcCoAS produces acetyl-CoA from acetic acid in the cytoplasm (Fig. 5G,H). It has been reported that *Saccharomyces pastorinous* upregulates expression of its *acetyl-CoA synthetase* genes, *ACS1* and *ACS2*, in response to increasing concentrations of acetic acid in the medium (Xu et al, 2021), which led us to speculate that the increased *AcCoAS* expression in the *nat3Δ*-fed larvae was possibly due to a greater supply of acetic acid from the *nat3Δ* yeast than the control yeast. To the control yeast diet, we added acetic acid alone or in combination with oleic acid and/or valine (Figs. 5I–K and EV4K–O). Although the addition of acetic acid by itself slightly delayed the timing of adult eclosion,

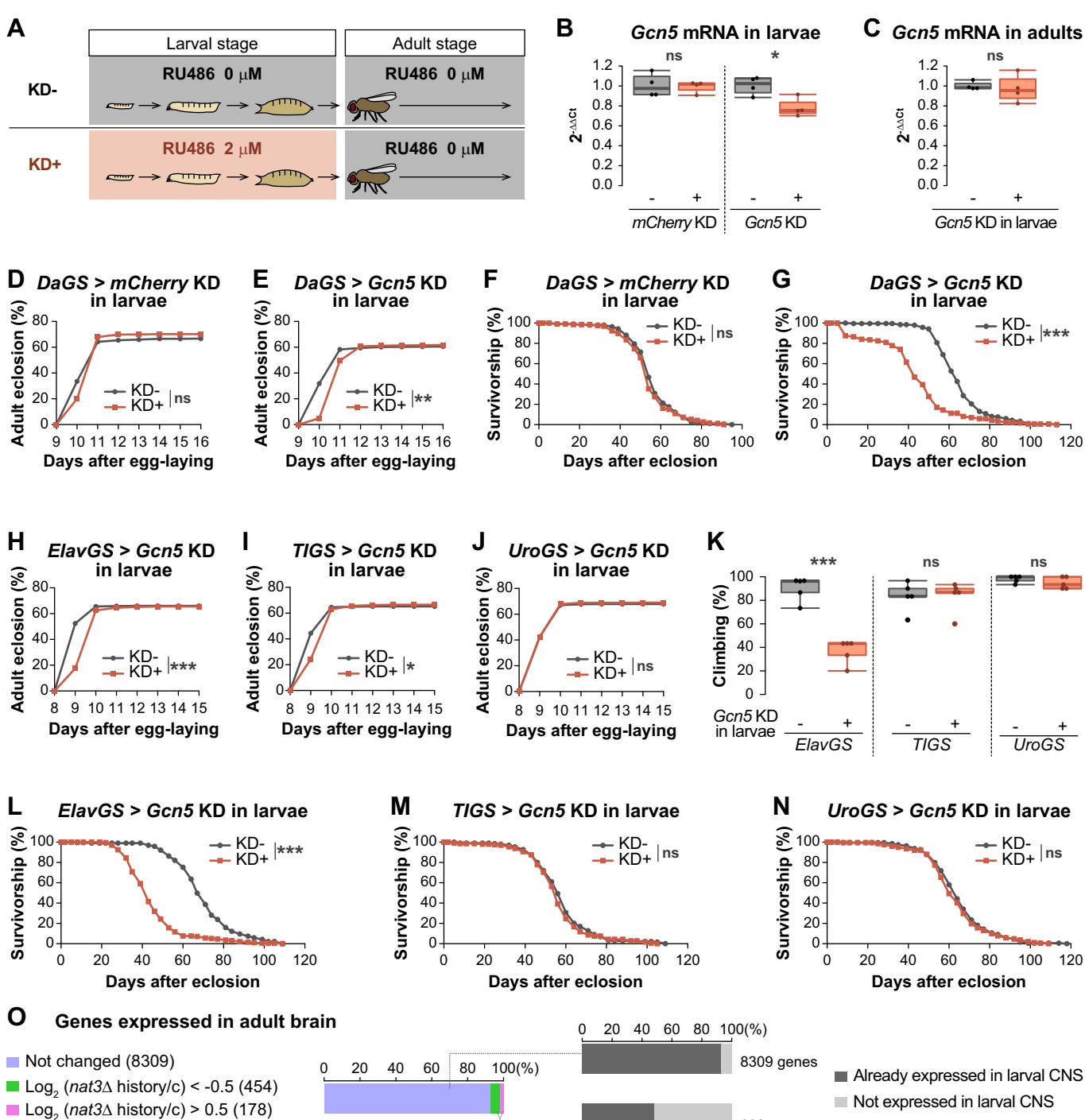

**O** Genes expressed in adult brain

- Not changed (8309)
- $\text{Log}_2$ (*nat3Δ* history/c) < -0.5 (454)
- $\text{Log}_2$ (*nat3Δ* history/c) > 0.5 (178)

8309 genes
632 genes

- Already expressed in larval CNS
- Not expressed in larval CNS

neither did it reduce the eclosion rate nor did it shorten the lifespan (Figs. 5I and EV4K,L). Among the combinations tested, only the supplementation of all 3 nutrients together negatively affected the development and shortened adult lifespan (Figs. 5J,K and EV4M–O; 10% and 6.1% reduction in the median lifespan in Figs. 5K and EV4O, respectively), mimicking the effects of the *nat3Δ* diet (Fig. 1D–F). These results raise the possibility that the concurrent higher intake of the three nutrients (oleic acid, valine, and acetic acid) in larval stages contributes to the shorter lifespan of the

*nat3Δ*-history adults. Hereafter, we refer to the supplemented diet with the 3 nutrients as the OVA diet, and larvae which were reared on the OVA diet as OVA-fed larvae.

Given the elevated level of *AcCoAS* transcripts in the *nat3Δ*-fed larvae, we predicted that *nat3Δ* yeast would produce more acetic acid compared to the control yeast. We measured the amounts of short-chain fatty acids including acetic acid both in the culture media and in the yeast cells, because it has been reported that a substantial amount of acetic acid is secreted by *Saccharomyces*

**Figure 4. Effects on male adult lifespan of larval-stage whole-body or tissue-specific *Gcn5* knockdowns.**

(A) Experimental designs for collecting adults that either experienced a partial gene knockdown (KD) in larval stages (KD + ; RU486, 2 μM) or did not (KD-; RU486, 0 μM), with subsequent monitoring of adult lifespan with full gene expression restored. Larval stage-selective gene KD was conducted essentially as depicted in the righthand side of Fig. 3A. Here, *Gcn5* or other genes were knocked down either broadly using *DaGS* (B–G) or in pan-neurons, the gut plus fat body, and the Malpighian tubule using *ElavGS*, *TIGS*, and *UroGS*, respectively (H–N). (B, C) RT-qPCR to quantify *Gcn5* transcript levels in male larvae (B) and in male adults (C). (B) Results in larvae with the respective genotypes for whole-body knockdowns using shRNAs targeting *mCherry* or *Gcn5*. "−" (gray) indicates no induction of short hairpin RNA expression, whereas "+" (orange) indicates the induction of shRNA by RU486. (C) The *Gcn5* transcript level in adults was not altered whether *Gcn5* had been knocked down in larval stages or not. (D–G) Effects of larval-stage whole-body KD of the indicated genes on adult eclosion (D, E) and male lifespan (F, G). (D, E) Adult eclosion percentage was calculated from the daily number of eclosed male and female adults. (F, G) Survival curves of male adults with or without the respective KD history in larval stages. The data of (B–G) were obtained in a set of experiments. The negative effects of larval-stage *Gcn5* KD on adult eclosion and male lifespan were reproduced in two more independent experiments. (H–N) Effects of the tissue-specific *Gcn5* KD in larval stages on adult eclosion (H–J), climbing ability of two-week-old male adults (K) and male lifespan (L–N). (H–J) Adult eclosion percentage was calculated from the daily number of eclosed male and female adults. (K) The data for 4H-4K were obtained in a set of experiments. See details in "Methods". None of the *Gcn5* KD conditions decreased the number of emerging adults, whereas KD in larval neurons caused a 1-day delay in the timing of the adult emergence and a severe decline in motor performance of the adults. (L–N) Survival curves of male adults with *Gcn5* KD history in larval neurons (L), in the larval intestine plus the adipose tissue (M), and in larval Malpighian tubules (N). The data of L-N were obtained in a set of experiments. Shorter lifespan phenotype of male flies with the neuronal *Gcn5* KD history was reproduced in one more independent experiment. (O) RNA-seq analysis of the adult brain of males reveals a long-term effect of the *nat3Δ* diet in larval stages on gene expression in the adult brain. (Left) Out of a total of 8941 genes expressed in the brain, the expression of 8309 genes was unaffected whether the larval diet was the *nat3Δ* yeast or the control yeast ("Not changed," light purple). Colored green or pink are genes whose $\log_2$ fold change between the control history adults and the *nat3Δ* history adults [$\log_2$(*nat3Δ* history/c)] was smaller than −0.5 (green) or larger than 0.5 (pink). (Right) We prepared triplicates for the respective conditions of this RNA-seq analysis (Dataset EV7). Boxplots are depicted as in "Statistical analysis" in "Methods". *$P < 0.05$, **$P < 0.01$, ***$P < 0.001$. The exact $P$ values, sample sizes and statistical tests employed are listed in Dataset EV12. Source data are available online for this figure.

*cerevisiae* (Blank et al, 2005; Kajihata et al, 2015; Zhang et al, 2022; Yatabe et al, 2023). Because the concentration of acetic acid in the culture medium is in the mM range (Zhang et al, 2022; Yatabe et al, 2023), we suspected that the extracellular pool might be more abundant and more effective to larvae than the intracellular pool in the ingested yeasts. Contrary to our assumption, the amount of acetic acid in *nat3Δ* yeast culture was not higher in the control culture, and actually lower in the cells compared to the respective amounts for the control yeast (Figs. 5L and EV5A; Dataset EV8B,C). Thus, our results did not support the hypothesis of an increased amount of acetic acid in the *nat3Δ* diet, and the characterization of other key compounds that critically influence adult lifespan in combination with oleic acid and valine is regrettably inconclusive. Nonetheless, our results indicate that the control yeast diet supplemented with the 3 nutrients (OVA) mimicked the effects of the *nat3Δ* diet on adult lifespan, as described above, and larval responses, described next.

## The OVA diet recapitulated the effects of the *nat3Δ* diet on larval gene expression and profiling of H3K9 acetylation

We addressed whether the OVA diet elicited similar effects on larval gene expression as the *nat3Δ* diet, and particularly whether the OVA diet was associated with signs of the diminished Gcn5 function. For this purpose, we acquired RNA-seq data and CUT&RUN H3K9ac data of the OVA-fed larvae (Figs. 6 and 7A–C; Appendix Fig. S5). The gene expression profile of the OVA-fed larvae closely resembled that of the *nat3Δ*-fed larvae regarding the overall upregulated and downregulated genes (Fig. 6A,B; Dataset EV9); moreover, the upregulated genes in the OVA-fed larvae included the fatty-acid and the BCAA degradation pathway genes and *AcCoAS*, as those in the *nat3Δ*-fed larvae did (Appendix Fig. S5; compare with Figs. EV3B–E,G and 5H). Most strikingly, the gene expression profile of the OVA-fed larvae was *Gcn5* or *Ada2a* mutant-like, recapitulating the profile of the *nat3Δ*-fed larvae (Fig. 6C–F; compare with Fig. 2B–E; also compare Figs. 6G,H with 2F,G, respectively).

Our CUT&RUN analysis found 10,789 H3K9ac peaks in the control and the OVA-fed samples. Of these peaks, 188 (1.7%) were altered in the OVA-fed larvae, and 187 of those were lower than in the control (blue in Fig. 7A), which is reminiscent of the data for the *nat3Δ*-fed larvae (Fig. 3C) and supports the notion that the OVA diet also diminished Gcn5 function in larvae. In contrast to the OVA diet, the oleic acid plus valine (OV) diet hardly affected H3K9ac peaks (Fig. 7B), indicating a vital contribution of acetic acid in the OVA diet. Comparing the effect of the OVA diet and that of the *nat3Δ* diet, we found significant overlaps with respect to the genes containing these lowered peaks; 26% (58/224) and 21% (46/224) of the OVA Down peak-containing genes overlapped with the respective *nat3Δ* and *Gcn5* KD genes (Fig. 7C). Therefore, both the OVA diet and the *nat3Δ* diet share the feature of partially reducing H3K9ac peaks. Together, the RNA-seq and CUT&RUN datasets support the idea that the supplementation with oleic acid, valine, and acetic acid adversely affect Gcn5 function in larvae.

We speculate that the *nat3Δ* diet and the OVA diet cause the similar metabolite changes through enhanced metabolic pathways, such as fatty-acid degradation, valine degradation and fatty-acid synthesis (described next), which lead to Gcn5 malfunctions in larvae. This is because expression of those metabolic genes were largely unaffected in the *Gcn5* mutant larvae (Carré et al, 2008), making it less likely that the elevated expression of those genes was a consequence of the reduced Gcn5 function.

## In the *nat3Δ*-fed larvae, the quantities of CoA-related metabolites were significantly changed, which could adversely affect Gcn5 function

Finally, we explored how the *nat3Δ* diet could diminish Gcn5 function in the larvae. In parallel to the RNA-seq analyses, we had already conducted metabolomic analyses of the wandering 3rd-instar larvae (Fig. 2A). We therefore looked for alterations in the abundance of any metabolites, which might affect Gcn5 function, in the list of the detected metabolites (Dataset EV5). One metabolite of interest was pantothenate (vitamin $B_5$), a precursor of coenzyme A (CoA), which was about 4-fold more abundant in the *nat3Δ*-fed

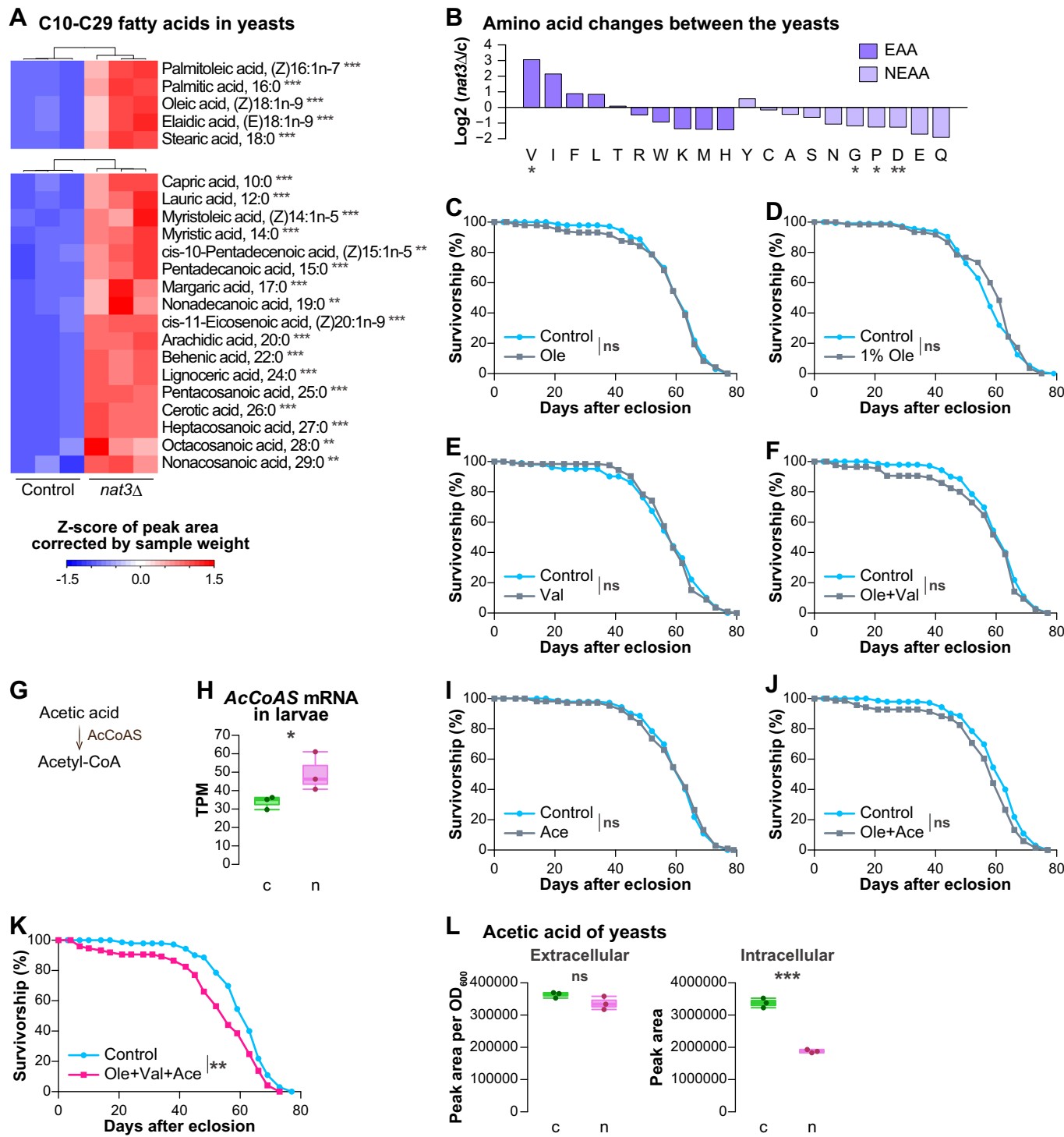

**A** C10-C29 fatty acids in yeasts

Palmitoleic acid, (Z)16:1n-7 ***
Palmitic acid, 16:0 ***
Oleic acid, (Z)18:1n-9 ***
Elaidic acid, (E)18:1n-9 ***
Stearic acid, 18:0 ***

Capric acid, 10:0 ***
Lauric acid, 12:0 ***
Myristoleic acid, (Z)14:1n-5 ***
Myristic acid, 14:0 ***
cis-10-Pentadecenoic acid, (Z)15:1n-5 **
Pentadecanoic acid, 15:0 ***
Margaric acid, 17:0 ***
Nonadecanoic acid, 19:0 **
cis-11-Eicosenoic acid, (Z)20:1n-9 ***
Arachidic acid, 20:0 ***
Behenic acid, 22:0 ***
Lignoceric acid, 24:0 ***
Pentacosanoic acid, 25:0 ***
Cerotic acid, 26:0 ***
Heptacosanoic acid, 27:0 ***
Octacosanoic acid, 28:0 **
Nonacosanoic acid, 29:0 **

Control    nat3Δ

**Z-score of peak area corrected by sample weight**
-1.5    0.0    1.5

**B** Amino acid changes between the yeasts

EAA
NEAA

V I F L T R W K M H Y C A S N G P D E Q

**C** Control / Ole (ns)

**D** Control / 1% Ole (ns)

**E** Control / Val (ns)

**F** Control / Ole+Val (ns)

**G**
Acetic acid
↓ AcCoAS
Acetyl-CoA

**H** *AcCoAS* mRNA in larvae

**I** Control / Ace (ns)

**J** Control / Ole+Ace (ns)

**K** Control / Ole+Val+Ace (**)

**L** Acetic acid of yeasts
Extracellular (ns)
Intracellular (***)

larvae than in the control (Fig. EV5B). Because *Drosophila* cannot synthesize pantothenate and the content of pantothenate within the yeast cells was not higher in the *nat3Δ* yeast compared to the control strain (Fig. EV5C), this increase of pantothenate in the *nat3Δ*-fed larvae was suggestive of altered pantothenate metabolism in larval cells. In the biosynthetic pathway for CoA biosynthesis, pantothenate is first phosphorylated by pantothenate kinase (PANK), and PANK is feedback-regulated by the end product,

CoA, and also by acetyl-CoA and acyl-CoA species (Fig. EV5D; Leonardi et al, 2005). Therefore, we suspected that the prominent increase in pantothenate in the *nat3Δ*-fed larvae implied concomitant increases in CoA and/or acetyl- or acyl-CoA species.

Our measurements of acetyl-CoA and CoA in the *nat3Δ*-fed larvae revealed that there was no significant change in the acetyl-CoA amount, whereas CoA was more abundant in the *nat3Δ*-fed larvae (Fig. 7D,E; Dataset EV10A), which resulted in a significant

◄ **Figure 5. Abundance of fatty acids and amino acids in the *nat3Δ* diet and effects of supplementations of oleic acid, valine and acetic acid to larval diets on male lifespan.**

(A) Heatmaps showing relative abundance of C10-C29 fatty acids in the control yeast and the *nat3Δ* strain. (Top) Abundances of five long-chain fatty acids (C16 and C18) were determined using gas chromatography-mass spectrometry (GC-MS) split mode. (Bottom) Other fatty acids, including very-long-chain fatty acids were measured by GC-MS, splitless mode. The amounts of all detected fatty acids were significantly increased in the *nat3Δ* yeast. (B) $Log_2$ fold changes in amino acid amounts between the control yeast and *nat3Δ* yeast that were obtained on MetaboAnalyst analysis of the liquid chromatography-tandem mass spectrometry (LC-MS/MS) data. The purple bars indicate changes in essential amino acids (EAA) for *Drosophila*, whereas the light purple ones indicate those of non-essential amino acids (NEAA). One BCAA, valine, was the most elevated in the *nat3Δ* yeast. (C–F) Effects of nutrition histories, when supplemented with 0.5% (C) or 1% (D) oleic acid ("Ole") alone, with valine ("Val") alone (E), or with both together (F), on male lifespan. Survival curves of the males with the respective nutrition histories. Control larvae were fed on the live control yeast and the agar medium (mSCM) containing Tween 80 (the control diet) and developed to adults (sky blue). The data of the experimental groups are colored in gray. (G) The enzymatic reaction of Acetyl-coenzyme A synthase (AcCoAS): acetic acid is consumed to produce acetyl-CoA in the cytoplasm. (H) Expression values (transcripts per million; TPM) of *AcCoAS* in the control yeast-fed larvae (green, "c") and the *nat3Δ*-fed larvae (pink, "n"). (I–K) Effects of adding acetic acid ("Ace") alone (I), or acetic acid and oleic acid (J), or the three nutrients, oleic acid, valine, and acetic acid, together (K) to the control nutrition history on male lifespan. Control larvae were fed on the same control diet as in (C–F) (sky blue). (L) Relative amounts of acetic acid in the liquid culture media ("Extracellular") and in the yeast cells ("Intracellular") of the control or *nat3Δ* yeast strain ("c" or "n"). The vertical axes show corrected peak areas by $OD_{600}$ values or raw values from liquid chromatography-mass spectrometry (LC-MS). Boxplots are depicted as in "Statistical analysis" in "Methods". *$P < 0.05$, **$P < 0.01$, ***$P < 0.001$. The exact *P* values, sample sizes and statistical tests employed are listed in Dataset EV12. Source data are available online for this figure.

decrease in the acetyl-CoA|CoA ratio (Fig. 7F). Previous studies indicated that the abundance of acetyl-CoA itself is not solely critical for the activity of yeast Gcn5 in a cell-free system and for bulk histone acetylation or the acetylation of several residues in human or mouse cells, but it is the acetyl-CoA|CoA ratio that matters (Tanner et al, 2000; Albaugh et al, 2011; Pietrocola et al, 2015; Kinnaird et al, 2016; Lee et al, 2014; Cluntun et al, 2015; Carrer et al, 2017). Although *D. melanogaster* Gcn5 activity may not have been characterized in such assays, we speculate that the decreased acetyl-CoA|CoA ratio in the *nat3Δ*-fed larvae is a potential condition that dampens Gcn5 function. Considering the larger intake of LCFAs and upregulation of the fatty-acid degradation pathway genes in the *nat3Δ*-fed larvae, it might be puzzling that the acetyl-CoA amount was not increased, since the fatty-acid degradation pathway produces acetyl-CoA in mitochondria. The unaltered acetyl-CoA amount in the *nat3Δ*-fed larvae could be due to a concurrent increased expression of the gene encoding Fatty acid synthase 1 (FASN1; Fig. EV5E), which consumes acetyl-CoA and produces long-chain fatty acids and CoA in the cytoplasm (Fig. EV5F).

Additionally, we measured the amounts of acyl-CoA species in the *nat3Δ*-fed larvae (Dataset EV10B). It has been reported that long-chain fatty acyl-CoA species, such as palmitoyl-CoA (C16:0), are potent competitive inhibitors of human Gcn5-catalyzed acetylation (Montgomery et al, 2015; Kulkarni et al, 2019). Amounts of four out of six long-chain fatty acyl-CoA species were significantly higher in the *nat3Δ*-fed larvae (Fig. 7G). These long-chain fatty acyl-CoA species can be produced by degradation of diet-derived LCFAs (Fig. 5A) and can also be synthesized in cells (Fig. EV5E,F). Although our measurements of all of the above endogenous inhibitory or competing metabolites cannot address their quantities at subcellular levels simultaneously (Trefely et al, 2020; Trefely et al, 2022), the overall findings raise the possibility that the altered metabolism in the *nat3Δ*-fed larvae adversely affects Gcn5 function (Fig. 7H).

## Does the dietary or knockdown intervention impact female lifespan?

We have so far described how the diets or the *Gcn5* knockdown in larval stages affect gene expression, metabolism, and lifespan

of males. There are substantial differences in gene expression and metabolism in response to the amount of dietary sugar between male and female larvae (Millington et al, 2022). Therefore, we wondered whether any of the dietary or genetic interventions in this study impact the lifespan of female adults, and we assessed the extent to which the gene expression patterns differ between the sexes in larval stages. It is known that female lifespan of *Drosophila melanogaster* strains depends on mating status and fecundity (Austad and Fischer, 2016). Therefore, we measured the lifespans of both mated and virgin females (Fig. EV6A–D). The ubiquitous *Gcn5* knockdown in larval stages strongly shortened the lifespans of both mated females and virgin females (Fig. EV6A,B), just as it shortened the male lifespan (Fig. 4G); thus, the *Gcn5* knockdown in larvae effectively shortened the lifespan irrespective of the sex. On the other hand, it was difficult to draw a definite conclusion concerning the dietary effects on female lifespan, due to the following observations: first, the *nat3Δ* diet for larvae did not affect the lifespan of mated females (Fig. EV1E). Second, the OVA diet marginally reduced the lifespan of mated females, whereas it did not significantly affect that of virgin females (Fig. EV6C,D).

RNA-seq datasets for both sexes were collected from the *Gcn5* KD ± larvae and compared to each other. Downregulated genes in females largely matched those in males, implying that their mutual downregulation contributes to the common effect of the knockdown on the lifespan in both the sexes. Feeding larvae on the OVA diet also produced large overlaps of differentially expressed genes between the sexes (Fig. EV6F; Dataset EV9), and those overlaps include the fatty-acid and the BCAA degradation pathway genes and *AcCoAS* (Appendix Fig. S5). Furthermore, the gene expression profile of the OVA-fed female larvae showed a characteristic similarity to that of the *Gcn5* mutant, much the same as the respective profiles for OVA-fed vs. *Gcn5* mutant male larvae (compared Appendix Fig. S6A,B with Fig. 6C,D, respectively). These parallels suggest that Gcn5 function is impaired in the OVA-fed larvae of both sexes. Nonetheless, the lifespan shortening of the OVA-history female adults was less conclusive compared to that of the males with the same nutrition history, as described above.

To conclude, our above results did not resolve the molecular basis, with respect to sex, of the discordant effects of the dietary interventions and the *Gcn5* knockdown on lifespan. It should be

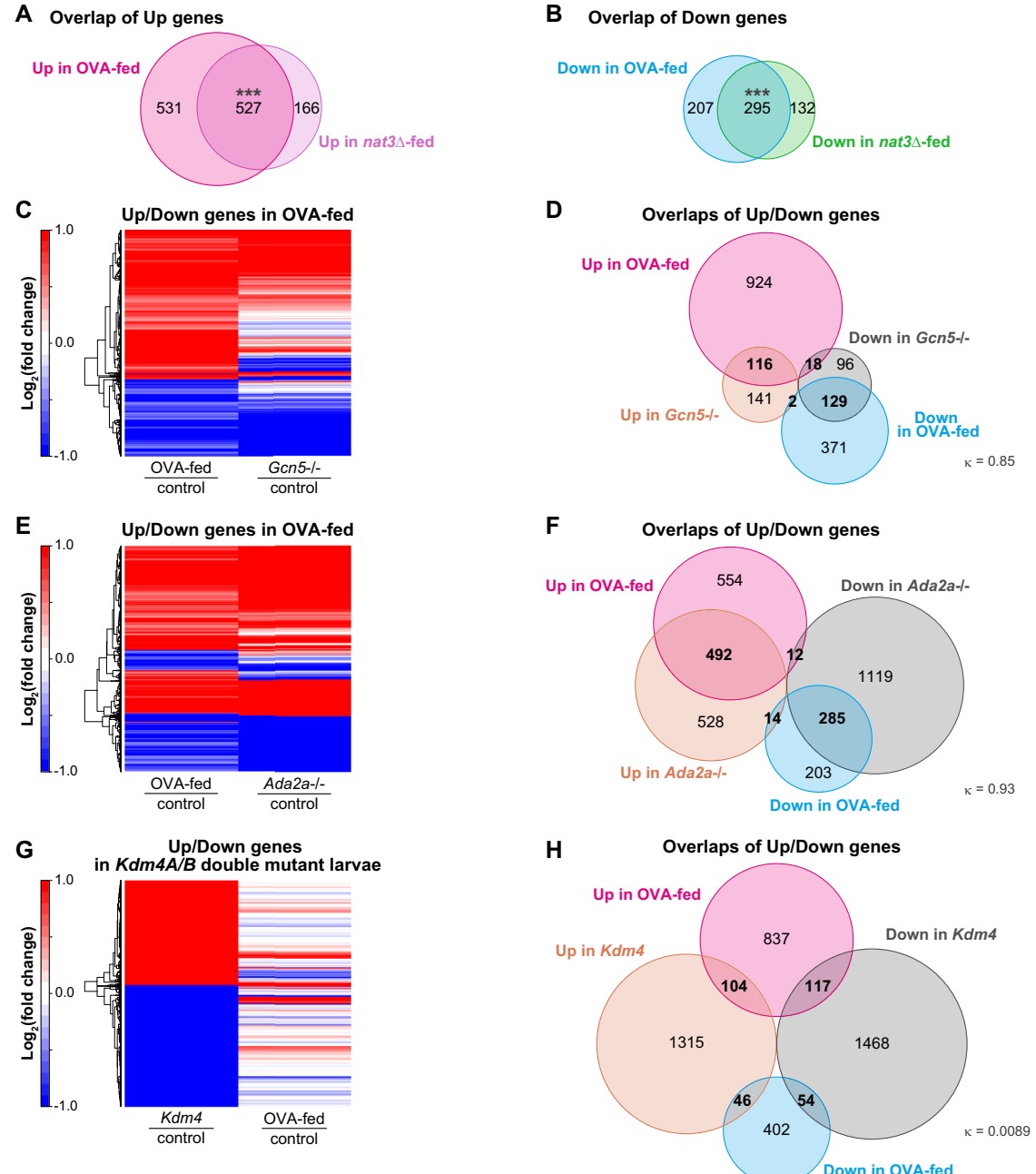

**A** Overlap of Up genes

**B** Overlap of Down genes

**C** Up/Down genes in OVA-fed

**D** Overlaps of Up/Down genes

**E** Up/Down genes in OVA-fed

**F** Overlaps of Up/Down genes

**G** Up/Down genes in *Kdm4A/B* double mutant larvae

**H** Overlaps of Up/Down genes

noted that females, whether they were mated or virgin, lived exceedingly longer than males under every condition tested: the *nat3Δ*-history datasets (compare Fig. 1E,F with Fig. EV1E); the OVA-history datasets (compare Figs. 5K and EV4O with Fig. EV6C,D); and the *Gcn5* KD±-history datasets (compare Fig. 4G with Fig. EV6A,B). We speculate that the OVA-history females, and possibly the *nat3Δ*-history females as well, may have better chances on the standard food to recover from the dietary effect in larval stages than their counterpart males, due to their extended longevity relative to males. Our results indicate that the effects of the *Gcn5* knockdown were more severe than those of the dietary interventions on lifespan irrespective of the sex (compare Fig. 1E,F with 4G, and Fig. EV6A,B with EV6C,D, respectively).

The knockdown effects might be longer-lasting even in female adults compared to the dietary effects.

## Discussion

Using a single-gene KO collection of budding yeast as a diverse source of larval diets, we uncovered an effective nutrition history, comprised of live *nat3Δ*-yeast cells, which exerts far-reaching effects beyond larval development, effectively shortening adult lifespan. The prominent responses in the *nat3Δ*-fed larvae point towards the diminished function of histone acetyltransferase Gcn5. By systematically testing nutrients added to the control yeast diet,

**Figure 6. Strong similarities of gene expression profiles between the OVA-fed male larvae and the *nat3Δ*-fed male larvae or the *Gcn5* mutant larvae.**

(A) Venn diagrams showing overlaps between the Up genes in the OVA-fed male larvae and the Up genes in the *nat3Δ*-fed (A) or the respective Down genes (B). Both the overlaps were significantly large (527 genes: 76% of the *nat3Δ*-fed Up genes and 295 genes in 69% the *nat3Δ*-fed Down genes). (C–H) Comparison of our RNA-seq data from the OVA-fed male larvae, the microarray data from *Gcn5* or *Ada2a* mutant larvae (Carré et al, 2008), and the microarray data of *Kdm4A* and *Kdm4B* double-mutant larvae (Tsurumi et al, 2019; designated as *Kdm4* mutant). These microarray data were collected presumably from larvae of both the sexes. (C, E, G) Heatmaps showing whether each of the significantly upregulated (Up) and downregulated (Down) genes in the OVA-fed male larvae (red and blue, respectively, in "OVA-fed|control") tend to be up or down in the *Gcn5* or *Ada2a* mutant larvae (red or blue in "*Gcn5*-/-|control" in panel C and "*Ada2a*-/-|control" in (E)). Because the available microarray data of the *Kdm4* mutant larvae contains only Up and Down genes in the mutant, panel G shows whether each of the Up and Down genes in the mutant (red and blue, respectively in "*Kdm4*|control") tends to be up or down in the OVA-fed male larvae (red or blue in "OVA-fed|control"). (D, F, H) Venn diagrams showing overlaps between the Up or Down genes in the OVA-fed male larvae and the Up or Down genes in the *Gcn5* mutant larvae (D), *Ada2a* mutant larvae (F), or *Kdm4* mutant larvae (H). The numbers of genes in the individual categories are indicated. Changes in gene expression were highly correlated between the OVA-fed male larvae and the *Gcn5* mutant larvae, and also between OVA-fed larvae and the *Ada2a* mutant larvae [$\kappa_{Gcn5} = 0.85$ (95% CI: 0.79–0.91) and $\kappa_{Ada2a} = 0.93$ (95% CI: 0.9–0.96); Cohen's kappa coefficient]. By contrast, the correlation was low between the OVA-fed male larvae and the *Kdm4* mutant larvae [$\kappa_{Kdm4} = 0.0089$ (95% CI: −0.09–0.11)]. Because we show that both the transcriptomic data of the *nat3Δ*-fed larvae and that of the OVA-fed larvae show significantly similarities to that of *Gcn5* mutant (Figs. 2B–E and 6C–F, respectively), we listed the overlapped Up/Down genes among the three datasets in Dataset EV9C. We prepared triplicates for the respective RNA-seq conditions (Dataset EV9). The exact *P* values, sample sizes and statistical tests employed are listed in Dataset EV12. Source data are available online for this figure.

we found that a combination of three nutrients (2 fatty acids, oleic and acetic acids, and a BCAA, valine) added to the control yeast diet (the OVA diet) recapitulated both the transcriptional and the H3K9ac features of the *nat3Δ*-fed larvae and the long-term effect of the *nat3Δ* diet on lifespan. In turn, these closely resembled the transcriptional features of the *Gcn5* mutant and the shorter lifespan of the flies in which Gcn5 expression was reduced to 78% of normal levels only during larval stages. Most likely, this *nat3Δ* diet or the OVA-diet-induced temporal reduction of Gcn5 function is the root cause of the shorter lifespan (Fig. 7H). Furthermore, larval neurons contribute to the transmission of that Gcn5-related history to later adulthood.

We explored how the intake of the *nat3Δ* diet dampened Gcn5 function and found that the acetyl-CoA|CoA ratio was decreased in the *nat3Δ*-fed larvae. This reduction should affect a broad spectrum of lysine acetyltransferases in cells, not just Gcn5 (Albaugh et al, 2011; Lee et al, 2014; Montgomery et al, 2016). In addition to the decreased acetyl-CoA|CoA ratio, Gcn5 function in the *nat3Δ*-fed larvae would be diminished by the increased amounts of long-chain fatty acyl-CoA species, which have been shown to be potent endogenous inhibitors of human Gcn5 (Montgomery et al, 2015). One such long-chain fatty acyl-CoA, palmitoyl-CoA, is an order-of-magnitude less effective on CBP/p300, which belongs to the same subfamily of histone acetyltransferases (HATs) as Gcn5. If this differential efficacy in a cell-free system (Montgomery et al, 2015) is also the case in *Drosophila* larval cells, it is likely that functions of CBP/p300 and possibly less closely related HATs in other subfamilies are less affected by the *nat3Δ* diet compared to the Gcn5 function.

In addition to the long-chain fatty acyl-CoA species, propionyl-CoA, one of the products of valine catabolism, would contribute to the reduction in the H3K9ac peaks in the *nat3Δ*-fed larvae, if its amount increases. This assumption is based on previous studies showing that propionyl-CoA is an additional competing substrate of Gcn5, and it reduces histone acetylation while increasing histone propionylation in a reciprocal manner in isolated nuclei (Montgomery et al, 2015; Simithy et al, 2017; Trefely et al, 2020). Collectively, our results suggest that the measured or speculative decreases in the acetyl-CoA level relative to other CoA-related metabolites above have a profound role in diminishing or altering Gcn5 function in the *nat3Δ*-fed and the OVA-fed larvae (Fig. 7H). Among the three nutrients of the OVA diet, the role of acetic acid

in the context of reducing histone acetylation remains the most enigmatic except for a finding that acetic acid supplementation to a *Drosophila* larval food reduces the bulk H3K9ac level in hemocyte progenitors in the wild-type background (Tiwari et al, 2020). It should also be noted that our analysis did not provide evidence for the increase of acetic acid in the *nat3Δ* diet itself, where presumably (an) unidentified key nutrient(s), not acetic acid, may contribute to altering the Gcn5 function.

How well have we verified the hypothesis that the *nat3Δ* diet in larval stages is the cause of the shorter lifespan? We have provided evidence that the *nat3Δ* diet reduces Gcn5 function, and also showed that adults with a *Gcn5* knockdown (KD) history in larval stages go on to exhibit a shorter adult lifespan. It should be noted that the profiles of H3K9ac reduction were significantly similar between the *nat3Δ* diet and the *Gcn5* KD, but not identical. A direct test of the hypothesis would require a rescue experiment, which addresses whether the short-lived phenotype of the *nat3Δ*-history adults would be restored if the *nat3Δ* diet-induced reduction in H3K9ac were restored either pharmacologically or genetically, and in fact we made trials along this line. In a previous study, mammalian cell lines were engineered to reduce the bulk histone acetylation by decreasing the amount of nucleocytoplasmic acetyl-CoA, and supplementation of the culture medium with acetic acid restored the acetylation, presumably by replenishing nucleocytoplasmic acetyl-CoA (Wellen et al, 2009). In contrast to this previous study using cell cultures, acetic acid supplementation to the diet of *Drosophila* larvae produces an adverse effect on H3K9 acetylation, as cited above (Tiwari et al, 2020), and is required for shortening adult lifespan in our study. Regarding the genetic rescue, we attempted to address whether *Gcn5* over-expression in the *nat3Δ*-fed or OVA-fed larvae affected lifespan. For this purpose, we performed a pilot experiment by feeding larvae harboring the *DaGS* driver construct; however, we found that they could not grow on live yeast diets in the presence of the chemical activator of the GS protein. Thus, other approaches are necessary.

Are the diet-induced Gcn5 malfunctions directly associated with the alterations in gene regulation? We examined how much the reduction in H3K9ac peaks by the *nat3Δ*-diet or the OVA-diet was correlated with the alterations of gene expression in the individual yeast diet-fed larvae. We found minor and unbiased overlaps between the genes showing diet-induced H3K9ac reductions and

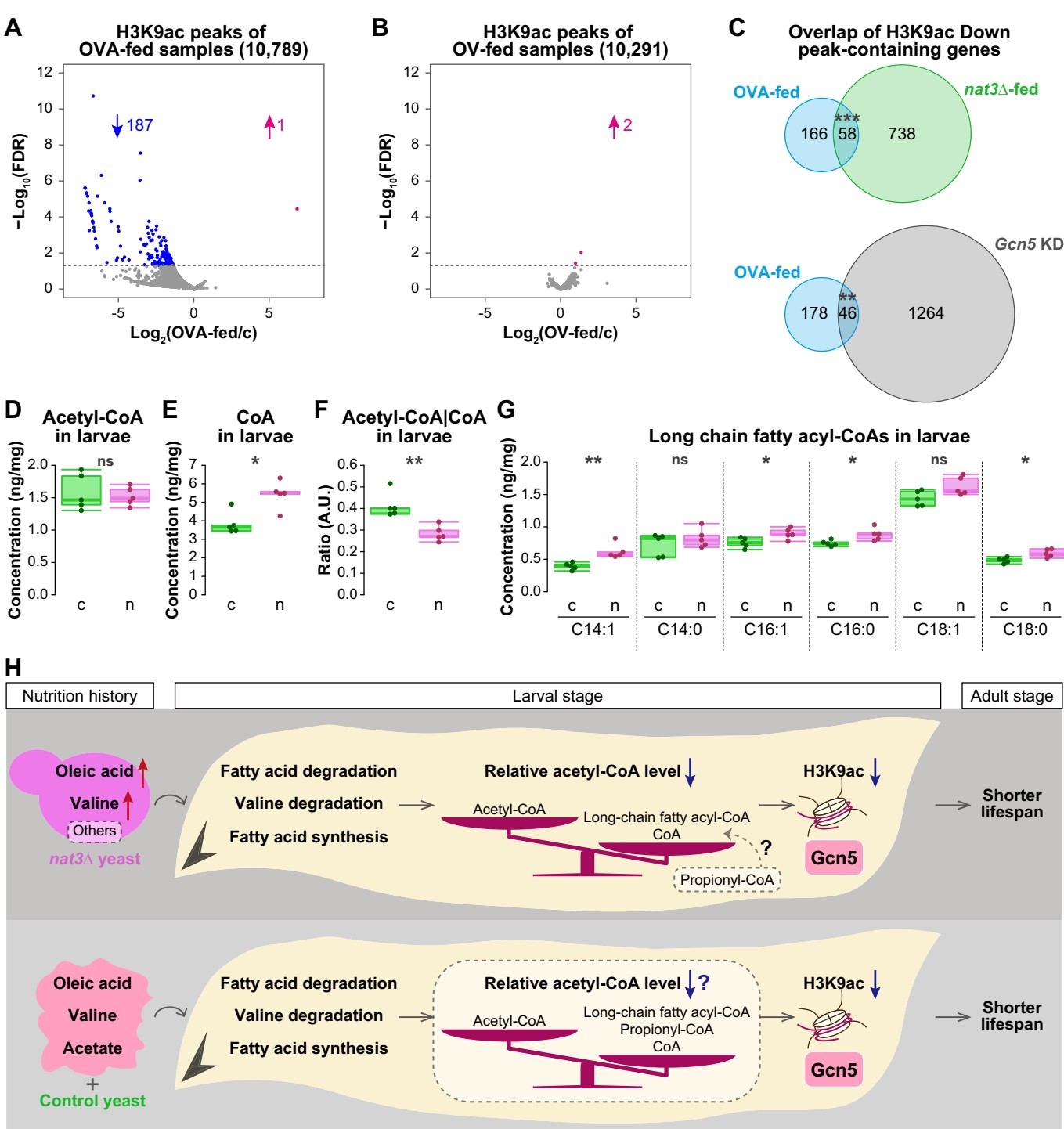

the Up or Down genes both in the *nat3Δ*-fed larvae and the OVA-fed larvae (Appendix Fig. S6C,D). Similarly, overlaps between the Down H3K9ac peak-containing genes and the Up or Down genes in the *Gcn5* KD larvae were minor (Appendix Fig. S6E). These results may reflect secondary and further indirect consequences of changes in transcription that were induced by the reduced H3K9ac peaks. The connections between the H3K9ac levels and the gene expression changes in larvae and adults remain to be investigated.

Our tissue-specific approaches have shown that the diminished *Gcn5* function in larval neurons affects adult lifespan and that the *nat3Δ* diet in larval stages causes a detrimental effect on neuronal gene expression in the adult stage. Although we did not analyze the relevant metabolites or H3K9ac in the larval CNS, a likely scenario would be that *nat3Δ*-derived fatty acids and BCAAs are carried by the *Drosophila* lipoprotein (lipophorin) and in free forms, respectively, cross the blood-brain barrier, and are transported to

**Figure 7. Genome-wide profile of H3K9 acetylation in the OVA- or OV-fed male larvae and quantities of CoA-related metabolites in the *nat3Δ*-fed male larvae.**

(A–C) CUT&RUN analyses of male larvae fed on the control diet, a control diet supplemented with oleic acid, valine and acetic acid (OVA), or a control diet supplemented with oleic acid and valine (OV). (A, B) Volcano plots showing the relative effects of either diet on H3K9ac peaks. Reduced and increased peaks are colored blue and magenta, respectively. (C) Venn diagrams showing the overlaps of the Down peak-containing genes between the OVA-diet condition and either the *nat3Δ*-diet condition or the *Gcn5* KD condition. The numbers of peaks for individual categories are indicated. We prepared H3K9ac duplicates and an IgG replicate for the respective conditions of these CUT&RUN analyses as explained in "Methods". (D–G) Concentrations (ng/mg larval weight) of acetyl-CoA (D), CoA (E), and long chain fatty acyl-CoA species (G), and calculations of acetyl-CoA|CoA ratios for individual replicates (F) in the control or *nat3Δ*-fed larvae ("c" or "n"). Boxplots are depicted as in "Statistical analysis" in "Methods". *$P < 0.05$, **$P < 0.01$, ***$P < 0.001$. The exact $P$ values, sample sizes and statistical tests employed are listed in Dataset EV12. (H) A model of the long-term effect of the *nat3Δ* diet or the OVA diet on adult lifespan. The *nat3Δ* yeast is rich in long-chain fatty acids including oleic acid, a branched-chain amino acid valine; and (an) unidentified additional key nutrient(s). In the *nat3Δ*-fed larvae, gene expression suggests that fatty acid degradation, valine degradation and fatty acid synthesis are enhanced simultaneously, which causes a reduction in the nucleocytoplasmic acetyl-CoA level relative to CoA-related compounds such as CoA, propionyl-CoA (not measured) and long-chain acyl-CoA species. In the OVA-fed larvae, similar changes in the metabolism are assumed by the gene expression while we did not measure their amounts or ratios. This altered metabolism in either set of larvae diminishes the histone acetyltransferase Gcn5 function, which is proposed to result in a shorter adult lifespan. Source data are available online for this figure.

larval CNS neurons; and the CoA-related metabolites generated in the neurons contribute to effects that are similar to those detected in the whole body (Brankatschk and Eaton, 2010; Matsuo et al, 2019). In mouse tissues, effects of a LCFA-rich high-fat diet on metabolites and acetylation of histone residues have been studied (Carrer et al, 2017). In liver and perigonadal white adipose tissue, the acetyl-CoA|CoA ratio and the bulk H3K9ac were reduced, although the reduction of H3K9ac was not deemed significant in that report. Our findings substantiate these results, implying an evolutionarily conserved mechanism. LCFAs are in fact major ingredients in various recipes of high-fat diets for model organisms, including *Drosophila* and mice (e.g., Flaven-Pouchon et al, 2014; Shi et al, 2021; Nayak and Mishra, 2021; Eickelberg et al, 2022). Some of those high-fat diets elicit long-term effects, such as obesity in adulthood in mice and lipotoxic cardiomyopathy of subsequent generations in *Drosophila*; mechanistically, these effects are manifested partly by altering the DNA methylation status of the *FGF21* gene and the systemic H3K27 trimethylation level, respectively (Yuan et al, 2018; Guida et al, 2019).

Our RNA-seq and/or metabolomic analyses of whole-body and/or brain of the *nat3Δ*-history adults did not detect obvious signs of elevated inflammation or metabolic disorders. Thus, it remains to be elucidated how exactly the Gcn5 dysfunction in the larval neurons influences adult health and what directly causes the earlier death of the *nat3Δ*-history adults. Because epigenetic modifications of histones can be responsive to environmental inputs (Katan-Khaykovich and Struhl, 2002; Etchegaray and Mostoslavsky, 2016; Reid et al, 2017; Sharma and Rando, 2017; Dai et al, 2020; Oleson et al, 2021), the H3K9ac profile unique to the *nat3Δ*-fed larvae might be largely erased during the relatively much longer adult life on the laboratory standard food. Consistently, our comparison of RNA-seq datasets of the larval CNS and the adult brain indicates that the resultant mis-regulation of a set of genes in the *nat3Δ*-fed larval neurons does not persist in adults (Dataset EV7). Future works will investigate the possibility that the brain malfunction in the *nat3Δ*-history adults, or suboptimal function of other cell types and tissues, including neural stem cells or adult tissues outside the nervous system, which might be affected systemically by the *nat3Δ*-fed larval neurons, provokes premature aging.

With the help of the larval-stage specific gene KD approach, our results altogether strongly suggest a causal link between a key nutrient, diet-derived metabolites, the chromatin-modifying enzyme Gcn5, and adult lifespan. Thus, other diets might produce other epigenetic effects, and this reverse-genetic KD approach could be applied to other chromatin-modifying enzymes that are modulated by endogenous metabolites (van der Knaap and Verrijzer, 2016; Li et al, 2018; Schvartzman et al, 2018; Suganuma and Workman, 2018). Such candidates may include Ada2b, which interacts with Gcn5 to form the SAGA complex (Torres-Zelada and Weake, 2021; Dent, 2024) and plays a role of innate immune memory (Fuse et al, 2022). If weak knockdown of any of those genes in larval stages produces a significant effect on adult traits, the knowledge gained might inform predictions of potential long-term risks or benefits of nutrition histories that are either deficient or abundant in the relevant metabolites, which could then be tested with relevant experimental diets.

The long-term effects outlined in the DOHaD hypothesis derive from critical nutritional environments during pregnancy and/or the lactation period, that is, the very early phases of the mammalian life cycle (Hanson and Gluckman, 2014; Langley-Evans, 2015). On the other hand, our study has targeted the nutrition history during the entire growth phase of *Drosophila*. Despite this difference, fatty acids and BCAAs are familiar nutrients for humans (Kimura et al, 2020; Neinast et al, 2019), and Gcn5 is an evolutionarily highly conserved epigenetic regulator. Thus, high intake of those nutrients during the entire growth phase might elevate the risk of long-term effects that could impact adult human health. One particularly vulnerable organ might be the developing brain.

# Methods

## Reagents and tools table

| Reagent/resource | Reference or source | Identifier or catalog number |
|---|---|---|
| **Experimental models** | | |
| *D. melanogaster*: wDah | Grandison et al, 2009; Slack et al, 2015 | N/A |
| *D. melanogaster*: CS | EHIME-Fly *Drosophila* Stocks of Ehime University | E-10002 |
| *D. melanogaster*: DaGS | Yamashita et al, 2021 | N/A |
| *D. melanogaster*: ElavGS | Yamashita et al, 2021 | N/A |
| *D. melanogaster*: TIGS | Yamashita et al, 2021 | N/A |
| *D. melanogaster*: UroGS | Yamashita et al, 2021 | N/A |
| *D. melanogaster*: UAS-Gcn5-RNAi | Bloomington *Drosophila* Stock Center | 33981 |

| Reagent/resource | Reference or source | Identifier or catalog number |
|---|---|---|
| *D. melanogaster: UAS-mCherry-RNAi* | Bloomington *Drosophila* Stock Center | 35785 |
| *D. melanogaster: UAS-ProcR-RNAi* | Bloomington *Drosophila* Stock Center | 29414 |
| *D. melanogaster: UASt-Eip78C-isoformA* | Bloomington *Drosophila* Stock Center | 91372 |
| *D. melanogaster: UAS-mCD8::GFP* | Bloomington *Drosophila* Stock Center | 32184 |
| *D. melanogaster: UAS-Tg-RNAi* | National Institute of Genetics | 7356R-1 |
| *S. cerevisiae*: BY4741 | National BioResource Project | BY23849 |
| *S. cerevisiae*: Yeast MATa Collection | Dharmacon | YSC1053 |
| **Antibodies and other reagents for CUT&RUN** | | |
| Anti-Histone H3 (acetyl K9) antibody | Abcam | ab4441 |
| CUTANA™ IgG Negative Control Antibody for CUT&RUN and CUT&Tag | EpiCypher | 13-0042 |
| cOmplete EDTA-free protease inhibitor | Roche | 11873580001 |
| 5% Digitonin | Invitrogen | BN2006 |
| pA-MNase | Iwasaki et al, 2021 | N/A |
| CUTANA™ *E. coli* Spike-in DNA | EpiCypher | 18-1401 |
| Rnase A | Thermo Fisher Scientific | EN0531 |
| Proteinase K | Thermo Fisher Scientific | EO0491 |
| AMPure XP Beads | Beckman Coulter | A63880 |
| Ultra II DNA Library Prep Kit for Illumina | NEBNext | E7645 |
| **Oligonucleotides and RNA-seq reagents** | | |
| *Gcn5* qRT-PCR forward primer: AACCAGGAGGTGGCAAATGT | This study | N/A |
| *Gcn5* qRT-PCR reverse primer: CGCAGGCAGGTCATCCAAAT | This study | N/A |
| *rp49* qRT-PCR forward primer: CAGTCGGATCGATATGCTAAGCTG | Watanabe et al, 2019 | N/A |
| *rp49* qRT-PCR reverse primer: TAACCGATGTTGGGCATCAGATAC | Watanabe et al, 2019 | N/A |
| TRIzol | Invitrogen | 15596018 |
| RNeasy Mini Kit (50) | QIAGEN | 74104 |
| Ultra II Directional RNA Library Prep Kit for Illumina | NEBNext | E7760 |
| **Software** | | |
| R | R Core Team, 2020 | https://www.r-project.org/ |
| SGD | Cherry et al, 2012 | https://www.yeastgenome.org |
| OASIS 2 | Han et al, 2016 | https://sbi.postech.ac.kr/oasis2/ |
| MetaboAnalyst | Pang et al, 2021 | https://www.metaboanalyst.ca |
| Trim Galore! | | https://github.com/FelixKrueger/TrimGalore |
| HISAT2 | Kim et al, 2019 | http://daehwankimlab.github.io/hisat2/ |
| HTSeq | Putri et al, 2022 | https://htseq.readthedocs.io/en/latest/ |

| Reagent/resource | Reference or source | Identifier or catalog number |
|---|---|---|
| edgeR | Robinson et al, 2009; McCarthy et al, 2012 | https://bioconductor.org/packages/release/bioc/html/edgeR.html |
| DAVID | Huang et al, 2009 | https://davidbioinformatics.nih.gov |
| KEGG | Kanehisa et al, 2012 | https://www.kegg.jp |
| FlyBase | Larkin et al, 2021 | https://flybase.org |
| Biovenn | Hulsen et al, 2008 | https://www.biovenn.nl |
| Bowtie2 | Langmead and Salzberg, 2012 | https://bowtie-bio.sourceforge.net/bowtie2/index.shtml |
| SAMtools | Li et al, 2009 | http://www.htslib.org |
| Picard | | https://broadinstitute.github.io/picard/ |
| MACS2 | Zhang et al, 2008 | https://hbctraining.github.io/Intro-to-ChIPseq/lessons/05_peak_calling_macs.html |
| DiffBind | Stark and Brown, 2011 | https://bioconductor.org/packages/release/bioc/html/DiffBind.html |
| IGV | Robinson et al, 2011 | https://igv.org/doc/desktop/ |
| deepTools | Ramírez et al, 2016 | https://deeptools.readthedocs.io/en/latest/ |

## *Drosophila melanogaster* strains and laboratory standard food

Wild-type strains used in this study were *white Dahomey* (*wDah*) and *Canton-Special* (*CS*). *wDah* was used unless otherwise noted (Grandison et al, 2009; Slack et al, 2015). *wDah*, *daughterless GeneSwitch* (*DaGS*), *ElavGS*, *TIGS*, and *UroGS* were all gifts from F. Obata (Yamashita et al, 2021). *CS* was obtained from EHIME-Fly *Drosophila* Stocks of Ehime University (E-10002). The following fly strains were obtained from the Bloomington *Drosophila* Stock Center: *UAS-Gcn5-RNAi* (#33981; Zeng et al, 2015), *UAS-mCherry-RNAi* (#35785), *UAS-ProcR-RNAi* (#29414), *UASt-Eip78C-isoformA* (#91372), and *UAS-mCD8::GFP* (#32184). *UAS-Tg-RNAi* (#7356R-1) was obtained from the National Institute of Genetics. All stocks were reared at a population density of 80–120 per vial (φ21 mm × 92 mm) at 25 °C under a 12 h:12 h light:dark cycle at 60% humidity. The recipe for our laboratory standard food (LSF) was described previously (Watanabe et al, 2017; Watanabe et al, 2019); briefly, the nutrient ingredients are corn flour, corn grits, dry yeast, and glucose, whereas the anti-fungal reagents are propionic acid and butyl *p*-hydroxybenzoate.

## Live yeast–fly assay

*S. cerevisiae* strains were cultured on YPD plates [50 g/L YPD medium (Clontech, 630409) and 20 g/L agar (Nacalai Tesque, 01162-15)] or on modified synthetic complete medium (mSCM) plates [60 g/L glucose (Nacalai Tesque, 16806-25), 5 g/L ammonium sulfate (Nacalai Tesque, 02620-75), 1.71 g/L Yeast Nitrogen Base (Sunrise Science Products, 1500-100), 0.79 g/L Complete Supplement Mixture (Sunrise Science Products, 1001-010) and

20 g/L agar] that contains neither fatty acids nor cholesterol. Live yeast diets were prepared as follows: the control yeast strain, BY4741 (National BioResource Project, BY23849, *MATa his3Δ1 leu2Δ0 met15Δ0 ura3Δ0*) and *nat3Δ* (Yeast MATa Collection Dharmacon, YSC1053) were revived on YPD plates from −80 °C frozen stocks, and grown at 30 °C for 2 days, followed by culturing at 30 °C for 2 days on mSCM plates. The respective yeast cells were harvested and suspended in sterile water to an optical density ($OD_{600}$) of 60–70 (~$8 \times 10^8$ cells/ml). In total, 40 µL of yeast suspension (~$3.2 \times 10^7$ cells) was added to the surface of 5 ml hardened mSCM per tube, and the tubes were incubated at 30 °C for 2 days before adding germ-free *D. melanogaster* embryos (Fig. 1B, right), as described next.

Germ-free *D. melanogaster* embryos were prepared as previously described (Watanabe et al, 2019; Mure et al, 2023), with some alterations. *CS* or *wDah* adults eclosed within 2 days were collected and maintained on the standard food (~50 adults per vial) for 3 days. Those adults were transferred to egg collection bottles (~250 adults per bottle) and apple juice agar plates topped with yeast paste were replaced on a daily basis. After being reared for 3–6 days, the adults were placed on fresh yeast plates for 1 h between 4 PM and 5 PM for "pre-clearing" and then allowed to oviposit on fresh yeast plates for 3 h between 5 PM and 8 PM. These embryos on plates were incubated for an additional 11.5 h and collected by washing with 0.7% NaCl + 0.3% TritonX-100. After removing already-hatched larvae under a dissection microscope, the embryos (~12–15 h after egg laying) were dechorionated with 50% bleach, washed with sterile water, and further washed with 70% ethanol, followed by two washes with sterile water. These germ-free embryos were suspended in a viscous solution (2.5% Polyvinylpyrrolidone, 0.1% TritonX-100, personal communication from T. Aigaki and Y. Sato) and 20 µl of the suspension (~30 embryos) was added to each tube with the live yeast on mSCM. Each tube was plugged with an autoclaved cotton ball (Kawamoto Sangyo, #20; Fig. 1B, right) and incubated at 25 °C at 90% or higher humidity.

Because of the three findings below and an abundance of literature pertaining to *wDah* and lifespan, we chose *wDah* for our lifespan assay: First, *wDah* lays much more eggs than *CS*. Second, on average, more than 90% of *wDah* germ-free embryos hatched, whereas 70% *CS* germ-free embryos hatched. Third, a separate project in our laboratory ran into an unanticipated trait of *CS*, although it has been a widely used wild-type strain: *CS* adults, which emerge on our standard food, show disorganized or lost adipose tissue (fat body) with approximately 30% genetic penetrance (Tsuyama et al, 2023). The fat body plays essential roles in storing and mobilizing energy substrates and functions as a pivotal signaling center for inter-organ communications that regulate energetic metabolism at the organismal level. Therefore, we consider it very risky to examine adult traits including lifespan using *CS*. In contrast, such a severe phenotype of adult fat body is hardly seen in *wDah*. *wDah* larvae that had been fed on control yeast developed to pupae from 5.5 days after egg-laying (AEL) onwards. From these pupae, adults started eclosing 10 days AEL (Figs. 1D and EV1A). No larvae pupariated on mSCM only (without the live yeast). Lifespans are dependent on fly strains and the impacts of a specific diet might be different from stain to strain. Thus, we do not exclude the possibility that the remaining 27 yeast mutants out of the 29 mutants (Fig. 1C) might contain one or more "false negatives(s)".

To supplement the control yeast diet, we added oleic acid (TCI, O0180), valine (Nacalai Tesque, 36108-42), isoleucine (PEPTIDE INSTITUTE, 2712) and/or sodium acetate/acetic acid buffer (3 M; pH 5.2; Nacalai Tesque, 06893-24), to mSCM with Tween 80 (Nacalai Tesque, 11718-22). Unless otherwise noted, the final concentrations of the individual additives were 0.5% v/v for oleic acid, 1.54 g/L for valine (0.14 g/L in mSCM + 1.4 g/L) which is slightly higher than 1.2 g/L in the exome-matched FLYAA holidic medium (Piper et al, 2017) or 1.25 g/L in HolFast holidic medium optimized for larval development (Sorge et al, 2025), 0.3 g/L for isoleucine (0.05 g/L in mSCM + 0.25 g/L) which is about one-quarter concentration in FLYAA medium or HolFast medium, 25 mM for sodium acetate/acetic acid buffer, and 0.05% v/v for Tween 80 (Maruyama et al, 2024). The above amounts of valine and isoleucine were determined on the basis of the respective concentrations in mSCM and the respective fold changes in the *nat3Δ* yeast compared to the control yeast (Fig. 5B). Briefly, we first prepared a 10x branched-chain amino acid (BCAA) solution and a 10x emulsion. The 10x BCAA solution was sterilized by filtration (0.22 µm). For the 10x emulsion, water and 5% Tween 80 were sonicated with or without oleic acid for 1 s per ml volume. Next, one, two, or all three of the supplements (10x BCAA solution, sodium acetate buffer, and the 10x emulsion, respectively) were added, in this order, to the autoclaved mSCM agar, just before it had firmed up. To achieve the respective final concentrations, we reduced the amount of water in mSCM considering the total volumes of the reagent(s) added after autoclaving. Uniformly mixing oleic acid with the other ingredients of mSCM in the presence of Tween 80 was critical; otherwise, the oleic acid separated and floated on top of the media, prompting the larvae to escape from the medium and die. The control yeast BY4741 was cultured on mSCM supplemented with these reagents at 30 °C for 2 days before placing the germ-free fly embryos onto the surface. We designate the control yeast diet supplemented with all 3 added ingredients as the OVA diet.

We examined whether heat-killed yeast could be substituted for the live yeast under our assay conditions. Yeasts were killed by heating at 60 °C for 60 min with agitation at 20-min intervals. We found that no pupae developed on heat-killed yeast, even in the presence of four times the number of heat-killed cells than were in the live-yeast controls. We observed that 1st and early 2nd-instar larvae went out of the food and died. These findings suggest (an) indispensable role(s) of live yeast in larval development in our live yeast–fly assay.

## Screening the yeast single-gene knockout collection

We fed *CS* larvae on each of 5153 yeast single-gene knockout (KO) strains of the collection (Dharmacon, YSC1053) and isolated 46 yeast strains that altered the timing and/or the rate of pupariation compared to the control yeast (Fig. 1C), as described below. Each KO yeast strain of the collection is stored in a well of a 96-well microtiter plate at −80 °C, and all of the strains are preserved in a total of 57 master plates. KO strains in each master plate were revived on 2 YPD plates by using a sterile 96-pin replicator (Watson, 4820-963S), grown at 30 °C for 2 days, then subsequently replicated on 6 mSCM plates and cultured at 30 °C for 2 days. From 5 of these 6 mSCM plates, yeast cells of each strain were collected and resuspended in 60 µl sterile water in each well of a microtiter

plate by using a 96-pin replicator. In total, 40 µl of each respective yeast suspension was added to a tube that contained mSCM. After incubation at 30 °C for 2 days, ~30 germ-free *CS* embryos were added to each tube, and the number of pupae was counted on day 8 AEL.

In our primary screening, we isolated yeast strains that yielded either a low or high pupariation rate among the strains within each master plate. This was partly because the master plates did not include the control strain and partly because we intended to make our screening as high-throughput as possible. The exact procedure consisted of preparing two replicates (mSCM tubes) per strain and selecting strains that gave rise to either less than 5 pupae in both replicates (284 strains) or more than 15 pupae in both (186 strains). In the secondary screening, we made new master plates of those isolated 470 strains plus the control strain, prepared yeast cell suspensions of the individual strains, made 3 replicates per KO strain and ten for the control, then measured pupariation rates as described above. Compared to the control yeast-fed larvae, 162 yeast strains reduced the pupariation rate, whereas 45 strains increased the rate. These 207 strains were re-screened by making five replicates per strain, and 46 strains of interest were selected for further analysis (Dataset EV1).

### Larval stage-specific gene knockdown or overexpression

Virgin females of the *GeneSwitch* driver stocks and males of the *UAS* lines were collected, maintained for 2 or 3 days for maturation, and mated in vials containing the standard food. The adults were then reared in egg collection bottles. Embryos were collected overnight (3 PM-8 AM), washed with 0.7% NaCl + 0.3% TritonX-100, and suspended in the viscous solution (described above in "Live yeast–fly assay"). In all, 40 µl of the suspension (~120 embryos) was added to each vial containing the standard food with RU486 (2 µM) or ethanol (a negative control; "RU486 0 µM" in Figs. 3A and 4A) and incubated at 25 °C and 60% humidity in a chamber with automatic temperature and humidity regulation (NK System, LPH-410NS). In total, 50 mM RU486 stock solution [1 g RU486 (Tokyo Chemical Industry, M1732) in 46.6 ml ethanol] was stored at −20 °C, up to 3 months.

### Lifespan assay

Lifespan assays were conducted basically as described (Linford et al, 2013; Piper and Partridge, 2016) with some modifications. Eclosed adults were collected within 3 days (day 10-13 AEL) for yeast-history flies, and 1 day (day 10-11 AEL) for knockdown or overexpression assays. The collected adults were subsequently allowed to mate for 2 days. They were then sorted by sex under mild $CO_2$ anesthesia; 10 males (or 10 females for the assay in Figs. EV1E and EV6A–D) were put into each vial, flipped to fresh vials every 2–4 days, and deaths and censors were scored for individual vials. Throughout the lifespan assays, the standard laboratory food was used (except for the food used in Fig. 4L–N where butyl *p*-hydroxybenzoate was doubled in concentration) and the food-containing vials were incubated at 25 °C on a 12 h:12 h light:dark cycle at 60% humidity in a chamber with automatic temperature and humidity regulation (NK System, LPH-410NS) and additional USB fans (Seong et al, 2020). When flipping adults to fresh vials, positions of all racks were shifted in the chamber so

that adults experienced different positions in the chamber as equally as possible. OASIS2 was used for survival analyses. Statistics of all lifespan assays are presented in Dataset EV2.

### Climbing assay

We collected male adults and measured their climbing ability [2- or 3-week-old *wDah* adults of (Fig. 1I) and 2-week-old adults with different *Gcn5* KD histories (Fig. 4K)]. The climbing assay was performed basically as described (Ali et al, 2011). Ten individuals with the same treatment were put in one vial with the standard food, and this vial was vertically joined with an empty vial with tape, so that the vials faced each other. With the empty vial at the bottom, the adults were allowed to acclimatize to the new setting for more than 1 min. We then gently tapped the adults down to the bottom of the empty vial, and the number of adults that climbed above 6 cm within 10 s was measured. This assay for the same group was repeated three times, allowing more than 1 min for a rest period between each trial. The climbing ability of flies that died or were stuck to the food was counted as zero. Such flies amounted to just 3 *ElavGS > Gcn5* KD+ flies (Fig. 4K). The climbing abilities in each vial were plotted as the average percentage of successful climbers.

### Metabolome analysis of *Drosophila* or yeast

For sample preparations of *Drosophila* larvae, 13–15 male wandering third-instar larvae were collected to prepare one biological replicate. They were washed with PBS to remove all traces of food. For preparations of adult flies, 20 males per replicate were collected under mild $CO_2$ anesthesia. Larvae or adults were transferred to tubes, weighed on a microbalance (Sartorius, CP225D), and then snap-frozen in liquid nitrogen. For preparations of yeast samples, the yeast strains were grown at 30 °C for 2 days on YPD plates, followed by 2 days on mSCM plates, followed by another 2-day incubation on fresh mSCM plates. A 100 ± mg sample of yeast cells was scraped from the surface of the final mSCM plates and snap-frozen.

The following processing and analysis steps were performed by Shimadzu Techno-Research, which employed liquid chromatography-tandem mass spectrometry (LC-MS/MS) for Primary Metabolites Ver. 2 or LSI Medience, which employed capillary electrophoresis-mass spectrometry (CE-MS) or liquid chromatography-mass spectrometry (LC-MS) using blind tests. Briefly, methanol and chloroform were added to homogenized samples (at Shimadzu, an internal standard consisting of 2-Morpholinoethanesulfonic acid and 10 mM HCl was also added), followed by centrifugation and evaporation of the upper layers. At Shimadzu, the samples were solubilized in 10 mM HCl and injected into the LC-MS/MS system (LC: Shimadzu, LC-30A; MS: Shimadzu, LCMS-8050; column: Sigma-Aldrich, Discovery HS F5-3, 2.1 mm I.D. × 150 mm, 3 µm; solvent A: 0.1% formate/water; solvent B: 0.1% formate/acetonitrile; temperature: 40 °C; flow velocity: 0.25 ml/min). At LSI, the samples were solubilized in 10% acetonitrile and injected into the LC-MS system (LC: Agilent Technologies, Agilent 1260 Series; MS: Agilent Technologies, 6520 Q-TOF; column: Shiseido, CAPCELL PAK C18 IF, 2.0 mm I.D. × 50 mm, 2 µm; solvent A: 5 mM ammonium acetate; solvent B: acetonitrile; temperature: 40 °C) or CE-MS [CE: Agilent

Technologies, Agilent CE; MS: Agilent Technologies, 6520 Q-TOF; column: GL Science, Polymicro tube, 50 μm I.D. × 1000 mm; sheath fluid: 5 mM ammonium acetate/50% methanol; electrolyte: 1 M formate (cation) and 20 mM ammonium formate/20 mM ammonium acetate, pH 10 (anion); voltage: ± 30 kV].

The analyses of the resulting raw data (peak areas corrected by each sample weight) were performed as described (Watanabe et al, 2019) using MetaboAnalyst 5.0/6.0 with some alterations ("Log transformation" in data transformation and "Parametric" in $t$ test). The LSI Medience data was analyzed for each measurement method [CE-MS (anion), CE-MS (cation), LC-MS (negative) or LC-MS (positive)].

## Body weight measurement

Larval body weight measurement was described above, in "Metabolome analysis of *Drosophila* or yeast." To measure adult body weight, male adults were reared for 1 week or 5 weeks as described in "Lifespan assay", and each adult was weighed on an ultramicrobalance (Sartorius, MCA3.6 P) with $CO_2$ anesthesia.

## RNA sequencing

Sample preparation and analysis of RNA-seq data were performed according to our protocols (Watanabe et al, 2019; Kanaoka et al, 2023; Mure et al, 2023). To prepare one replicate for whole-body RNA-seq, 5 male or female wandering 3rd-instar larvae were washed with PBS, or 10 male adults (1 or 5 weeks old) were anesthetized with $CO_2$, before they were homogenized in TRIzol (Invitrogen, 15596018) using a Power Masher II unit (Nippi, 893002). To prepare each replicate of central nervous system (CNS) RNA-seq, larval CNS (the brain and the ventral nerve cord) was dissected from each of five male wandering 3rd-instar larvae, or adult brains were dissected from five $CO_2$-anesthetized male adults (3 weeks old). These dissected tissues were collected in ice-cold PBS before homogenization in TRIzol. Further purification of extracted RNA was carried out using RNeasy Mini Kit (QIAGEN, 74104). Prepared libraries with Ultra II Directional RNA Library Prep Kit for Illumina (NEBNext, E7760) were sequenced on an Illumina NextSeq 500 sequencer using single-end run 75 bp to achieve a depth per sample of 20 M reads except for the OVA-fed male larvae analysis where the sequence read length was 150 bp and these 3' extra 75 bp reads were excluded by the Trim Galore! v0.6.7 hardtrim5 tool. Library preparation and sequencing were performed using blind tests.

RNA-seq data were analyzed as follows: The obtained sequence data were mapped to a FASTA file (dmel-all-chromosome-r6.04.fasta) with a transcript annotation file (dmel-all-r6.04.gtf) that were downloaded from FlyBase using a mapping tool HISAT2 v2.2.1 (option: --rna-strandness R), and reads were counted using htseq-count v2.0.2 in HTSeq (option: -r pos -s reverse -a 10). Batch effects were adjusted in differential expression analyses using edgeR v3.30.3 for the dataset of 5-week-old *nat3Δ*-history male adults, all those of CNS and that of OVA-fed female larvae. The Database for Annotation, Visualization and Integrated Discovery (DAVID) Functional Annotation Chart was used to find significantly enriched Gene Ontology (GO) molecular function terms or KEGG Pathways. For comparisons with previous microarray data (Carré et al, 2008; Tsurumi et al, 2019), we updated the FlyBase ID of each gene using FlyBase ID Validator FB2023_04 and focused on genes present in our gene list. To assess correlations between the differential gene expression on the *nat3Δ* diet or the OVA diet, and by the respective gene mutations (Figs. 2C,E,G and 6D,FH; Appendix Fig. S6B), Cohen's kappa coefficient (Cohen, 1960; Landis and Koch, 1977) was calculated with each set of four boldface numbers presented in the respective figures (see legends for individual figures). Hypergeometric test (Subramanian et al, 2005; Sadeghi et al, 2011) was used to test for statistical significance of overlaps in Figs. 6A,B and EV6E,F. In Fig. 6A, for instance, the following code was input to R: phyper (527-1, 693, 11892-693, 1058, lower.tail=FALSE). All the numerical values used for these tests are shown in Dataset EV12. More information related to RNA-seq can be found in Datasets EV4, EV9 and EV11.

## qRT-PCR

qRT-PCR was performed essentially as described previously (Watanabe et al, 2019). RNA from five whole bodies of male wandering 3rd-instar larvae was extracted in 80 μl water, and that of ten whole bodies of male adults (1 week old) was extracted in 40 μl water. A 500 ng RNA sample was used for cDNA synthesis. After an initial denaturation at 95 °C for 1 min, 40 cycles of PCR were carried out: 95 °C for 15 s, 60 °C for 1 min. For the relative expression value, $2^{-\Delta\Delta Ct}$, we first calculated a $\Delta Ct$ value for each sample from the difference between the Ct of *Gcn5* and that of *rp49*, and subtracted the average $\Delta Ct$ value of the control samples from each $\Delta Ct$ value, which yielded the $\Delta\Delta Ct$ value.

## Sample preparations for CUT&RUN

Sample preparation was essentially performed as described in previous reports (Skene and Henikoff, 2017; Skene et al, 2018; Meers et al, 2019; Ahmad and Spens, 2019). To prepare one replicate, 5 male wandering 3rd larvae were washed with PBS and dissected in drops of Wash+ buffer [20 mM HEPES pH 7.5, 150 mM NaCl, 0.1% BSA, 0.5 mM spermidine, and 1 tablet of cOmplete EDTA-free protease inhibitor (Roche, 11873580001) per 50 ml] on parafilm, and all of their tissues and the remaining Wash + buffer, including hemocytes, were transferred to a microfuge tube on ice. After the collected sample was centrifuged (600 × g, 4 °C, 3 min), the supernatant was removed, the pellet was gently resuspended in DBE buffer [2 mM EDTA, 0.05% digitonin (5% stock: Invitrogen, BN2006) in Wash+ buffer], and then incubated with rotation at low speed for 10 min at 4 °C. After another centrifugation, the DBE supernatant was replaced with an antibody solution diluted 1:100 in DBE buffer and the tube was incubated at 4 °C overnight. For experimental and negative control samples, the antibodies used were anti-H3K9ac (Abcam, ab4441) and IgG (EpiCypher, 13-0042), respectively. Subsequently, the samples were washed with DBE buffer and then incubated with DBE buffer containing 700 ng/ml pA-MNase at 4 °C for 1 h. pA-MNase was a generous gift from Dr. Yoshitake (Iwasaki et al, 2021). After centrifugation and washing with Wash+ buffer, they were resuspended in Wash+ buffer and chilled to 0 °C in an ice-water bath for 3–5 min. Immediately after 2× Rxn buffer (3.8 mM $CaCl_2$ in Wash+ buffer) was added to activate pA-MNase and start the cleavage reaction, the samples were returned to the ice-water bath. After incubation at 0 °C for 30 min, 2× STOPyR buffer [2 pg/ml

E. coli Spike-in DNA (EpiCypher, 18-1401), 62.6 µg/ml RNase A (Thermo Fisher Scientific, EN0531), 200 mM NaCl, 20 mM EDTA, and 4 mM EGTA] was added and mixed to stop the cleavage reaction. The samples were then incubated at 37 °C for 30 min for fragment release, and the supernatants were collected by centrifugation. SDS and Proteinase K (Thermo Fisher Scientific, EO0491) were added to the samples, followed by incubation at 50 °C for 2 h. The DNA was recovered with AMPure XP Beads (Beckman Coulter, A63880). Finally, DNA fragments were eluted into 13 µl nuclease-free water and stored at −20 °C. We prepared two H3K9ac replicates and one IgG replicate as a negative control for each condition.

From each extracted DNA sample, a library was prepared with Ultra II DNA Library Prep Kit for Illumina (NEBNext, E7645). Amplification was performed with 13 cycles for yeast-fed samples and 15 cycles for Gcn5 KD samples. Prepared libraries were sequenced on an Illumina NextSeq 500 sequencer using paired-end run (75 bp) to achieve a depth per sample of 20 M reads (10 M fragments). Library preparation and sequencing were performed using blind tests.

## Data analysis of CUT&RUN

We analyzed CUT&RUN data according to previous reports with some alterations (Smolko et al, 2018; Uyehara and McKay, 2019; Hu et al, 2020; Montgomery et al, 2020; Zheng et al, 2020). The obtained sequencing data had adapters trimmed by FastQC with Trim Galore! v0.6.7 (option: --paired--fastqc). Sequences were mapped to D. melanogaster reference genome r6.04, which was downloaded from FlyBase and also used for our RNA-seq analysis, using Bowtie2 v2.2.5 (option: --local--very-sensitive-local--no-unal--no-mixed--no-discordant--phred33 -I 10 -X 700). Reads with mapping quality less than 10 were removed with SAMtools v1.3.1. In addition, PCR duplicates were removed with MarkDuplicates in Picard v2.18.29. H3K9ac peaks were called using MACS2 v2.2.6 (option: -f BAMPE -g dm -B--keep-dup all -q 0.05). DiffBind v3.6.5 was used to identify differentially bound sites between the control and nat3Δ-fed larvae, between the Gcn5 KD ± samples or between the control and the OVA-fed samples with options [1 kb peak interval (summits = 500) and RLE normalization on background bins]. This peak width decision was on the basis of both visualization of our data on the Integrative Genomics Viewer (IGV) and previous reports (Kharchenko et al, 2011; Yin et al, 2011). The intersectBed tool in deepTools v3.5.1 was used to find an overlap between significantly differential peaks with FDR less than 0.05 of the respective sample sets. For identification of peak-containing genes, intersectBed was also used with each down peak file and a bed file of longest gene isoforms, which was generated from a GTF file (dmel-all-r6.04.gtf) obtained from FlyBase. Hypergeometric test (Subramanian et al, 2005; Sadeghi et al, 2011) was used to test for statistical significance of overlaps in Figs. 3E,F and 7C. In Fig. 3E, for instance, the following code was input: phyper(97442-1, 1451090, 15443594-1451090, 889454, lower.tail=FALSE). All the numerical values used for these tests are shown in Dataset EV12. The bigwig files used in the images (Figs. 3B and EV2C) were produced from unnormalized H3K9ac bam files against unnormalized IgG files by bamCompare in deepTools (option: --operation subtract--binSize 10), with negative values undisplayed on IGV.

## Measurement of fatty acids in yeast

For medium-, long- and very long-chain fatty acids, we prepared frozen yeast samples in the same manner as in "Metabolome analysis of Drosophila or yeast." These samples were analyzed at Kazusa DNA Research Institute with an optimized protocols for fatty acids including both esterified and non-esterified fatty acids. Briefly, after 3:10 methanol:methyl tert-butyl ether mixture was added to the samples, they were homogenized with a beads mix ($\phi$0.5 mm and $\phi$5 mm) and sonicated, followed by addition of water and centrifugation. Methyl tert-butyl ether fractions were collected and dried by spraying nitrogen gas, followed by adding of 10% boron trifluoride methanol was added. After methyl esterification, water and hexane were added to the samples, followed by centrifugation. Fatty acid methyl ester in hexane fractions were detected by gas chromatography-mass spectrometry (GC-MS: Shimadzu, QP-2010 Ultra; auto sampler: Shimadzu, AOC-5000 Plus; column: Agilent Technologies, DB-5ms, 0.250 mm I.D. × 30 m, 0.25 µm; vaporization chamber temperature: 280 °C; column oven temperature: 40 °C, 2 min −>6 °C/min −>320 °C, 5 min) with 2 injection modes: split and splitless.

For intracellular short-chain fatty acids, we prepared yeast samples as described above. At Kazusa DNA Research Institute, after the addition of methanol (300 µl to100 mg yeast pellet), samples were homogenized (TissueLyser, 25 Hz, 2 min) with Zirconia $\phi$5 mm beads and centrifuged ($20,000 \times g$, 10 min, RT). Lipophilic substances were removed as follows: 100 µl methanol, 100 µl 75% methanol, and 100 µl supernatant were applied in a stepwise manner to MonoSpin C18 column (GL Science), and the flowthrough was collected, which was designated as solid-phase extraction (SPE) flowthrough. For derivatization, the SPE flow-through was diluted 1:1 with 75% methanol, and 50 µl of this dilution was combined stepwise with 50 µl of 5 µl/ml 2-ethylbutyric acid in 75% methanol solvent, 50 µl of 50 mM 3-nitrophenylhydrazine in 75% methanol solvent, 50 µl of 50 mM 1-ethyl-3-(3-dimethylaminopropyl)carbodiimide in 75% methanol solvent, and finally 50 µl of 7.5% pyridine in 75% methanol solvent; and the mixture was incubated at 25 °C in the dark for 30 min. Then, 250 µl 1% formic acid/75% methanol was added to this mixture and filtrated (Hydrophilic PTFE, 0.2 µm). In total, 1 µl of the filtrate was injected into the LC-MS system (Shimadzu, LCMS-8050).

For extracellular short-chain fatty acids, we cultured yeasts in the mSCM liquid medium according to Yatabe et al, 2023 and analyzed short-chain fatty acids in the medium as follows: after revival on YPD plates, the respective yeast cells were suspended in 2 ml of mSCM liquid in 18 ml glass tubes to an $OD_{600}$ of 0.1 and cultured at 30 °C for 24 h at 180 rpm shaking speed (pre-pre-culture). These cultured yeast cells were diluted in 2 ml mSCM to an $OD_{600}$ of 0.1, followed by pre-culture (30 °C, 16 h, 180 rpm). After the main culture (30 °C, 24 h, 180 rpm), 1 ml of the culture medium was collected with a 0.45 µm filter (Nacalai Tesque, 06544-94) attached to a syringe, and snap-frozen. For removal of lipophilic substances at Kazusa DNA Research Institute, 100 µl methanol, 100 µl 75% methanol, and 100 µl of 1:3 sample:methanol mixture were applied in a stepwise manner to MonoSpin C18 column, followed by the same derivatization and LC-MS analysis steps as above. In Kazusa, all these samples were analyzed using blind tests. For each GC-MS injection mode or LC-MS analysis, the

resulting raw data were analyzed using MetaboAnalyst in the same manner as in "Metabolome analysis of *Drosophila* or yeast."

## Measurement of acetyl-CoA, CoA or fatty acyl-CoAs in larvae

We prepared frozen samples of larvae as described above, in "Metabolome analysis of *Drosophila* or yeast." We discussed and optimized protocols with Shimadzu Techno-Research, where acetyl-CoA, CoA, and acyl-CoA species were quantified using blind tests. Briefly, in determining the amounts of acetyl-CoA and CoA, methanol was added to homogenized samples in 50 μM 1,4-dithioerythritol/5% 5-sulfosalicylic acid; these samples were centrifuged, and supernatants were injected into an LC-MS/MS system [column: GL Science, Inertsil ODS-3 2.1 mm I.D. × 50 mm, 3 μm; solvent A: 10 mM ammonium acetate; solvent B: methanol; temperature: 40 °C; flow velocity: 0.2 ml/min; scanning: positive ESI]. Calibration curves were generated by linear regression using the least-squares method with the peak area values of samples for calibration curves [(25-2500 ng/ml acetyl-CoA trilithium salt/5% 5-sulfosalicylic acid solution + 50 μM 1,4-dithioerythritol) or (25-2500 ng/ml CoA trilithium salt/5% 5-sulfosalicylic acid solution + 50 μM 1,4-dithioerythritol)]. In measuring fatty acyl-CoA species, samples were homogenized in 20 μM *tert*-Butylhydroquinone/2:1 methanol:chloroform mixture. Supernatants obtained after centrifugation were injected into an LC-MS/MS system (column: Imtakt, Presto FF-C18 2.0 mm I.D. × 150 mm, 2 μm; solvent A: 0.01% ammonia; solvent B: acetonitrile; solvent C: 0.1% formate/acetonitrile; temperature: 40 °C; flow velocity: 0.2 ml/min; scanning: positive ESI). Calibration curves were generated with the peak area values of samples for calibration curves (20–1000 ng/ml acyl-CoA C16:0 in 20 μM *tert*-Butylhydroquinone/50% methanol) as described above.

### Statistical analysis

We prepared biological replicates for each analysis. *P* values or adjusted *P* values (*FDR* or *Bonferroni-adjusted P* values) less than 0.05 were considered statistically significant in all analyses. For statistical analyses of lifespan, metabolome, RNA-seq, and CUT&RUN data, see each Dataset, and each respective section of "Methods". R was used for the other statistical analyses. The exact *P* values, sample sizes and statistical tests employed are listed in Dataset EV12. As for boxplots, boxes represent upper and lower quartiles, while the central lines indicate the median. Whiskers extend to the most extreme data points, which are no more than 1.5 times the interquartile range. BioVenn was used to construct Venn diagrams.

## Data availability

The source data of this paper are collected in the following database record: biostudies:S-BSST1869. All the raw sequence data obtained in this study have been deposited in the DDBJ Sequence Read Archive. Whole-body RNA-seq data of the *nat3Δ*-fed male larvae and adults: DRR514560-DRR514577 (BioProject accession number: PRJDB17027. CUT&RUN data of the *nat3Δ*-fed male larvae: DRR514578-DRR514583 (BioProject accession number: PRJDB17027. CUT&RUN data of the *Gcn5* KD male larvae: DRR514584-DRR514589 (BioProject accession number:

PRJDB17027. CNS RNA-seq data: DRR514590-DRR514613 (BioProject accession number: PRJDB17027. Whole-body RNA-seq data of the OVA-fed male larvae: DRR582639-DRR582644 (BioProject accession number: PRJDB18504. Whole-body RNA-seq data of the OVA-fed female larvae: DRR635698-DRR635703 (BioProject accession number: PRJDB20000. Whole-body RNA-seq data of the *Gcn5* KD male and female larvae: DRR635704-DRR635715 (BioProject accession number: PRJDB20000. CUT&RUN data of the OVA-fed and OV-fed male larvae: DRR635716-DRR635724 (BioProject accession number: PRJDB20000.

The source data of this paper are collected in the following database record: biostudies:S-SCDT-10_1038-S44319-025-00503-8.

## Peer review information

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

## Acknowledgements

We thank T Kondo, Y Sando, and Y Yoshitake in the NGS room in iSAL of Kyoto University; Y Yoshitake and T Kochi for technical support with CUT&RUN; F Obata for providing us with fly strains and experimental details for larval stage-specific knockdown and critical reading of the manuscript; T Aigaki, Y Sato, T Kuraishi, F Matsuda, M Tachibana, T Yamamoto, A Ito, Y Yashiroda, Y Shinkai, K Shirahige, I Hiratani, S Hirabayashi, R Niwa, T Umehara, MDW Piper, T Maruyama and K Matsumiya for technical advice and discussion; K Seong and K Siu for setting lifespan assay environments; J Hejna and S Goulas for polishing the manuscript; N Tanaka and K Ekwall for instructing statistical analysis; H Imai, Y Niitani, K Oki, and M Futamata for technical and secretarial assistance, and the other members of the Uemura laboratory for technical advice and discussion. The reagents, genomic datasets, and/or facilities were provided by the Bloomington *Drosophila* Stock Center, National Institute of Genetics, the *Drosophila* Genetic Resource Center at Kyoto Institute of Technology, EHIME-Fly *Drosophila* Stocks of Ehime University, Kyoto Encyclopedia of Genes and Genomes (KEGG; Kanehisa et al, 2012), FlyBase (Larkin et al, 2021; Gramates et al, 2022), the National BioResource Project (NBRP), Japan (JPNBRP202225), and Saccharomyces Genome Database (SGD; Cherry et al, 2012). This work was supported by the Japan Agency for Medical Research and Development (AMED-CREST; JP18gm1110001 to TU), the Japan Society for the Promotion of Science (JSPS; 21J15091 to SM, 17K15039, 21K06186, 24K09470, and 24H02321 to YH, and 17KT0018 and 23K27179 to TU), and the Japan Science and Technology Agency (JST; JPMJFR2051 to YH).

## Author contributions

**Shoko Mizutani**: Conceptualization; Resources; Data curation; Formal analysis; Validation; Investigation; Visualization; Methodology; Writing—original draft; Writing—review and editing. **Kanji Furuya**: Conceptualization; Investigation; Methodology; Writing—review and editing. **Ayumi Mure**: Methodology; Writing—review and editing. **Yuuki Takahashi**: Data curation; Formal analysis; Investigation; Methodology; Writing—review and editing. **Akihiro Mori**: Conceptualization; Methodology; Writing—review and editing. **Nozomu Sakurai**: Formal analysis; Methodology; Writing—review and editing. **Takuto Suito**: Formal analysis; Methodology; Writing—original draft; Writing—review and editing. **Kohjiro Nagao**: Conceptualization; Investigation; Methodology; Writing—review and editing. **Masato Umeda**: Conceptualization; Methodology; Writing—review and editing. **Kaori Watanabe**: Methodology; Writing—review and editing. **Yukako Hattori**: Conceptualization; Supervision; Funding acquisition; Methodology; Writing—original draft; Project administration; Writing—review and editing. **Tadashi Uemura**: Conceptualization; Supervision; Funding acquisition; Methodology; Writing—original draft; Project administration; Writing—review and editing.

Source data underlying figure panels in this paper may have individual authorship assigned. Where available, figure panel/source data authorship is listed in the following database record: biostudies:S-SCDT-10_1038-S44319-025-00503-8.

## Disclosure and competing interests statement

The authors declare no competing interests.

# Expanded View Figures

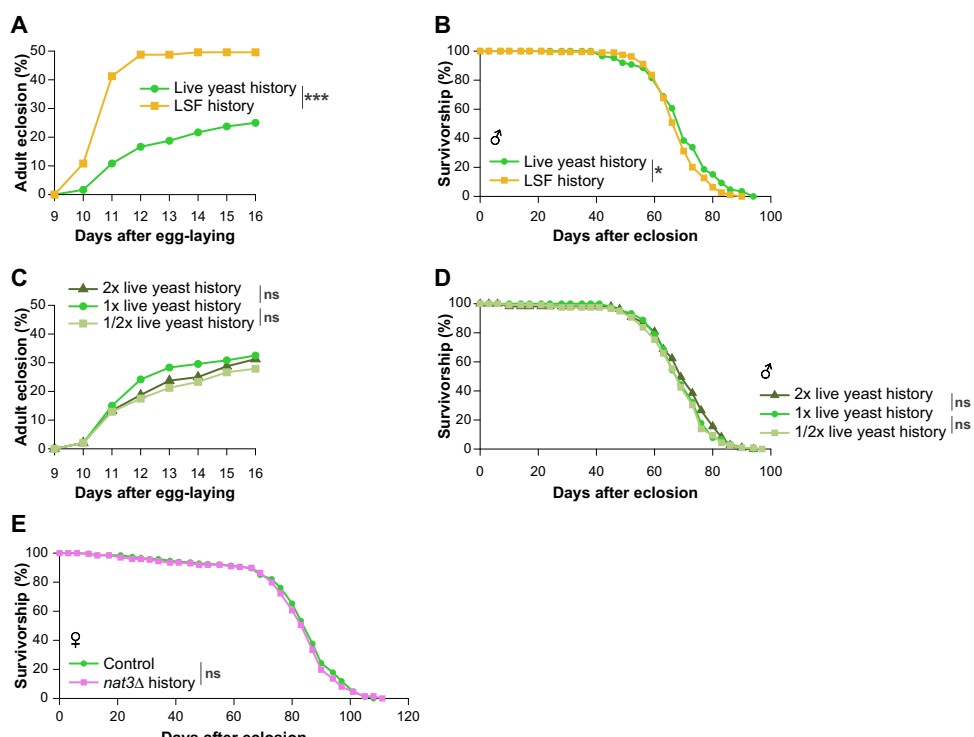

**Figure EV1. Characterization of the live yeast–fly assay.**

(A, B) Effects of the two distinct foods in larval stages on adult eclosion (A) and male lifespan (B): the modified synthetic complete medium (mSCM) plus live yeast (BY4741) in our live yeast–fly assay and LSF. (A) The numbers of emerging adults were much fewer on the live yeast diet ("Live yeast history") than on LSF ("LSF history"). In contrast to this major difference in the effect on the development, the difference in the lifespan was quite minor (B). These results indicate that our live yeast diet provides a less favorable environment for development compared to LSF, but the eclosed adults live essentially as long as the adults that developed on LSF from the very beginning of the larval stage. Adult eclosion percentage was calculated from the daily number of eclosed male and females. (C, D) Effects of amounts of live yeast (BY4741) on adult eclosion (C) and male lifespan (D). 40 µl of yeast suspensions with three different concentrations [1/2x, 1× (~8 × $10^8$ cells/ml) and 2x] were added to each mSCM tube, and these tubes were cultured at 30 °C for 2 days before addition of germ-free fly embryos. Neither half nor double the quantity of live yeast produced a significantly different effect on larval development or on adult lifespan. These results suggest that differences in the growth of yeast single-gene KO strains on mSCM, if they are within the range tested here, do not affect development or lifespan. Adult eclosion percentage was calculated from the daily number of eclosed male and females. (E) The *nat3Δ* diet in larval stages did not affect female lifespan. *$P < 0.05$, **$P < 0.01$, ***$P < 0.001$. The exact $P$ values sample sizes and statistical tests employed are listed in Dataset EV12. Source data are available online for this figure

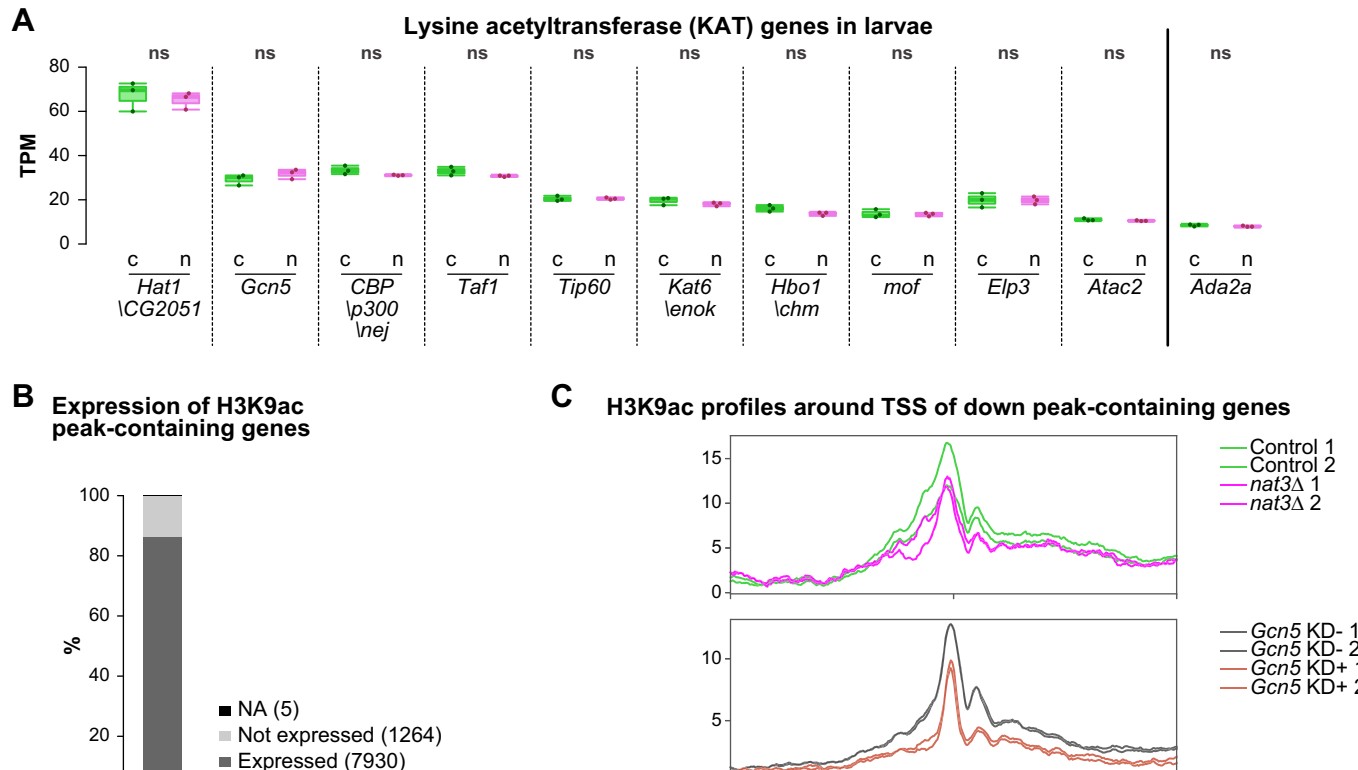

**A** Lysine acetyltransferase (KAT) genes in larvae

**B** Expression of H3K9ac peak-containing genes

**C** H3K9ac profiles around TSS of down peak-containing genes

**Figure EV2. Other characterizations of RNA-seq and CUT&RUN data related to the *nat3Δ*-fed male larvae and the *Gcn5* knockdown male larvae.**

(A) Expression values (transcripts per million; TPM) of ATAC complex genes, including *Gcn5* and *Ada2a*, and other lysine acetyltransferase (KAT) genes in our larval whole-body RNA-seq data. None of the gene expressions examined were significantly different between the control and *nat3Δ* diets. We did this analysis because we assumed that feeding the *nat3Δ* diet partially diminished Gcn5 function in larvae on the basis of the data of Fig. 2B-2E, prompting us to ask whether any subunit genes of the ATAC complex were downregulated or not. The data of *Gcn5* and *Ada2a* (rightmost) are shown in the graph and that of other subunit genes (Torres-Zelada and Weake, 2021; Dent, 2024) are not shown. The other 9 genes in the graph encode *Drosophila* KATs, whose preferred target lysine residues in histones have been studied by mass spectrometry (Feller et al, 2015). (B) Breakdown of 9199 H3K9ac peak-containing genes of yeast-fed larvae. 86% are expressed in our RNA-seq data. 5 genes in "NA", *His3:CG33866*, *His3.3 A*, *His3.3B*, *His4r* and *DIP1*, are not included in our RNA-seq annotation gene list. (C) H3K9ac profiles around TSS of the Down peak-containing genes of the *nat3Δ*-fed larvae (top) and those of *Gcn5* KD+ larvae (bottom). The profiles of the individual H3K9ac replicates are shown. Boxplots are depicted as in "Statistical analysis" in "Methods". The exact *P* values, sample sizes and statistical tests employed are listed in Dataset EV12. Source data are available online for this figure

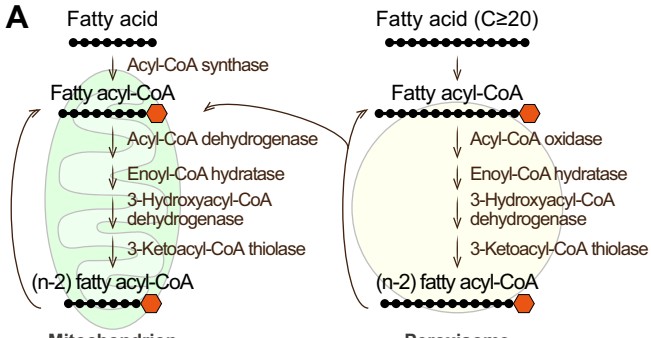

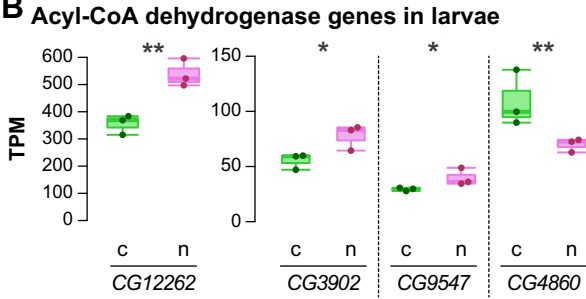

**B** Acyl-CoA dehydrogenase genes in larvae

**C** Acyl-CoA oxidase genes in larvae

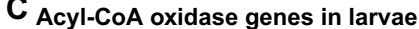

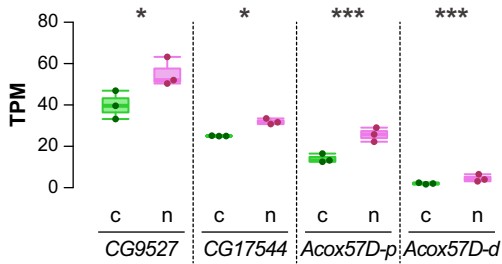

**D** Acyl-CoA binding protein genes in larvae

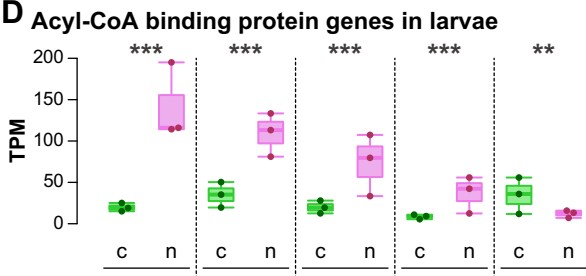

**E** Other enzyme genes of fatty acid degradation in larvae

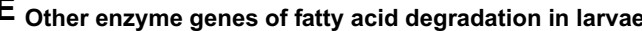

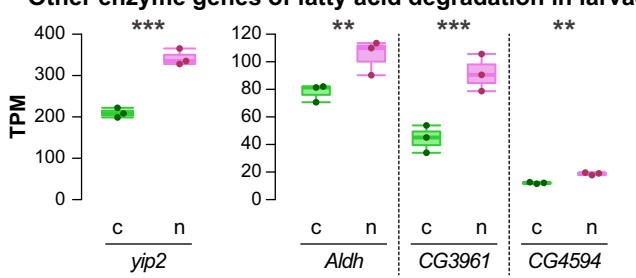

**F**

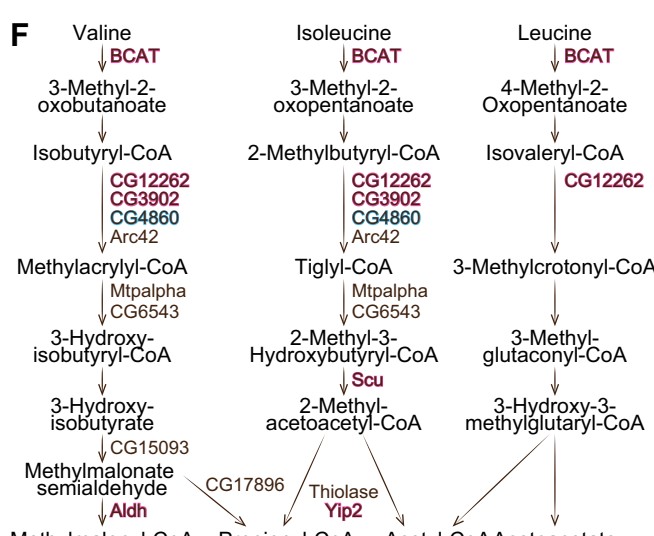

**G** Enzyme genes of in BCAA degradation in larvae

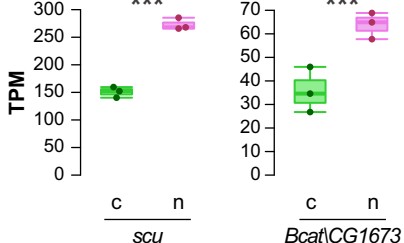

**Figure EV3.   Elevated gene expression of the fatty acid degradation pathways and branched-chain amino acid (BCAA) degradation pathways in the *nat3Δ*-fed male larvae.**

(A) β-Oxidation pathways of saturated fatty acids in mitochondria and peroxisomes, and representative enzymes that are conserved between *Drosophila* and mammals. (B–E) Gene expression values (TPM) of 4 groups of the fatty acid degradation pathways in our RNA-seq data from the control yeast-fed larvae (green, "c") and *nat3Δ*-fed larvae (pink, "n"): acyl-CoA dehydrogenase genes (B), acyl-CoA oxidase genes (C), long-chain fatty acyl-CoA binding protein genes (D), and other genes including a 3-ketoacyl-CoA thiolase (acetyl-CoA acyltransferase) gene, *yip2*, and an acyl-CoA synthetase gene, *CG3961* (E). Most of the genes shown here were upregulated in the *nat3Δ*-fed larvae, whereas *CG4860* in (B) and *CG8629* in (D) were downregulated. In each group, genes whose expression levels were not different (ns) between the control yeast-fed larvae and the *nat3Δ*-fed larvae or not detected (undetected) are the following: *CG7461* (ns) and *Arc42* (ns) in group B; *CG5009* (ns) and *CG4586* (ns) in group C; *CG8498* (ns), *CG8814* (ns), *CG14232* (ns) and *CG33713* (undetected) in group D. (F) BCAA degradation pathways in mitochondria of *D. melanogaster* that are partially shared by β-Oxidation pathways. Differentially expressed enzyme genes are shown as below: those shaded in magenta are upregulated in *nat3Δ*-fed larvae and/or in the OVA-fed larvae; CG4860 in sky blue is downregulated in both the *nat3Δ*-fed larvae and the OVA-fed larvae, and the others are upregulated only in the OVA-fed larvae (panel B; Appendix Fig. S5B,E,F). (G) Expression values (TPM) of 2 differential genes of BCAA degradation pathways in the control yeast-fed larvae (green, "c") and the *nat3Δ*-fed larvae (pink, "n"). The other differential BCAA degradation genes are common with those in fatty acid degradation pathways. See panels (B, E). Boxplots are depicted as in "Statistical analysis" in "Methods". $*P < 0.05$, $**P < 0.01$, $***P < 0.001$. The exact *P* values, sample sizes and statistical tests employed are listed in Dataset EV12. Source data are available online for this figure

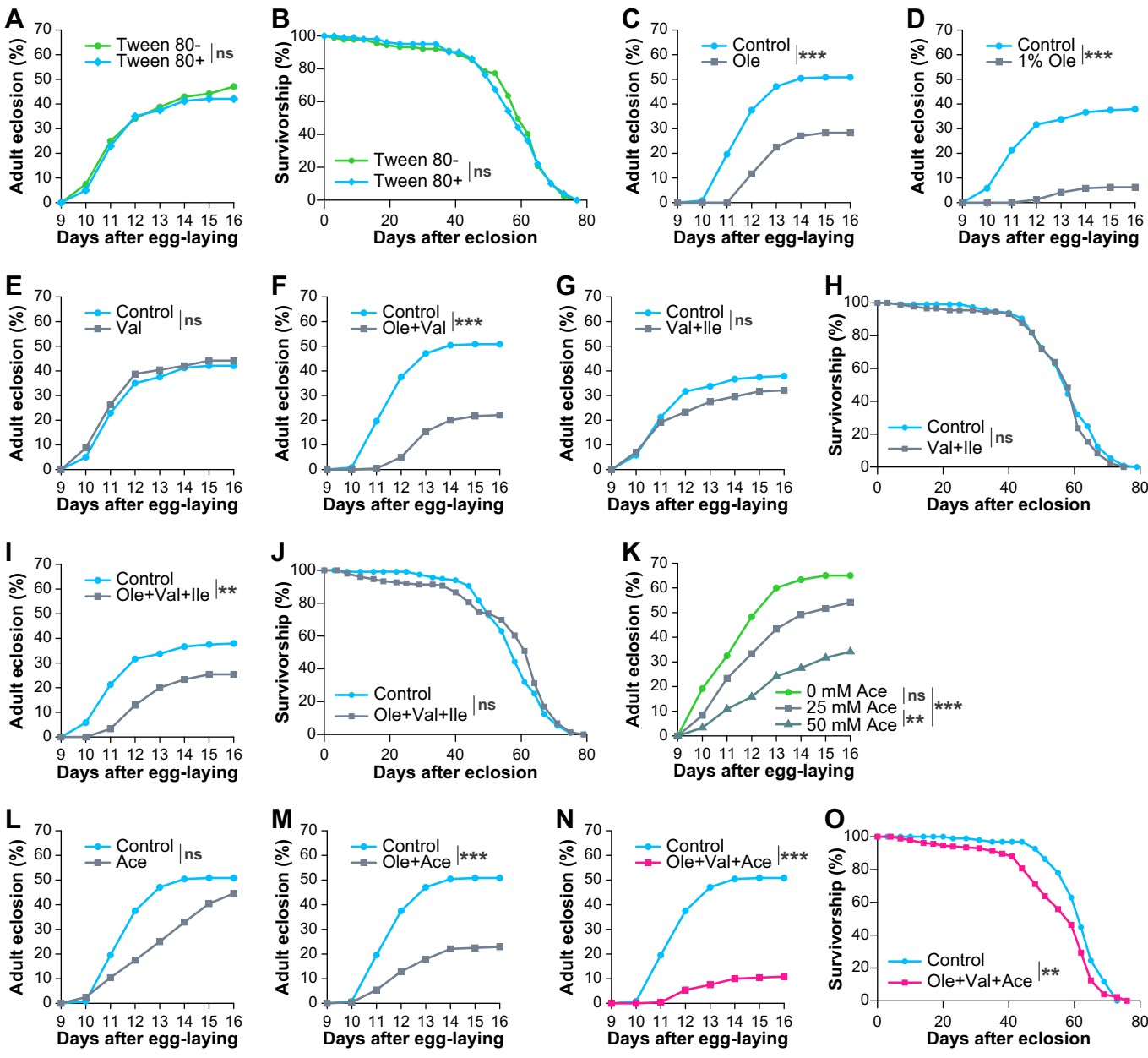

**Figure EV4. Effects of supplemented reagents on adult eclosion and male lifespan.**

(A, B) Addition of 0.05% Tween 80 to the control yeast diet for larvae did not affect adult eclosion (A) nor male lifespan (B). Adult eclosion percentage was calculated from the daily number of eclosed male and females. (C–O) Effects of supplemented reagents on adult eclosion (C–G, I, K–N) and male lifespan (H, J, O). Throughout this figure, control larvae ("Control") were fed on the live control yeast and the agar medium (mSCM) containing Tween 80 and developed to adults (sky blue). The data of the experimental groups are colored in gray. Adult eclosion percentage was calculated from the daily number of eclosed adult flies of both the sexes. Abbreviations: Ole (oleic acid), Val (valine), Ile (isoleucine), Ace (acetic acid). *$P < 0.05$, **$P < 0.01$, ***$P < 0.001$. The exact $P$ values, sample sizes and statistical tests employed are listed in Dataset EV12. Source data are available online for this figure

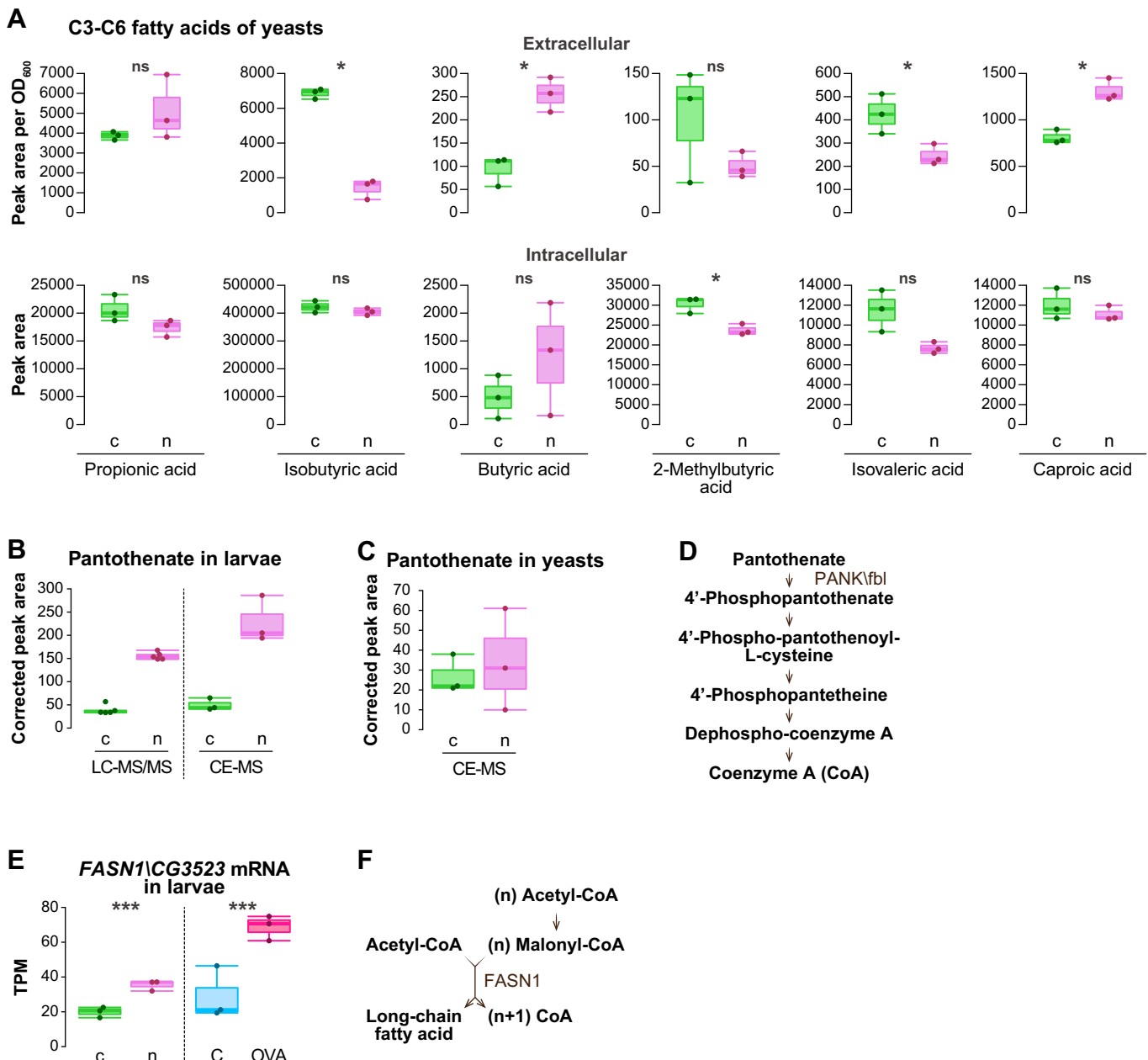

**Figure EV5. Abundance of short-chain fatty acids and pantothenate of the yeast strains and/or the yeast-fed male larvae, and expression of *Fatty acid synthase1* (*FASN1*) in the larvae.**

(A) Relative amounts of short-chain (C3-C6) fatty acids in the liquid culture media ("Extracellular") and in the yeast cells ("Intracellular") of the control or *nat3Δ* yeast strain ("c" or "n"). The vertical axes show corrected peak areas by $OD_{600}$ values or raw values from liquid chromatography-mass spectrometry (LC-MS). (B–D) Relative amounts of pantothenate in the yeast-fed larvae (B) and in the yeast strains (C). The vertical axes show peak areas that were measured in metabolome analyses (LC-MS/MS or CE-MS) and corrected by individual sample weights. Samples of either the control yeast-fed larvae or the control yeast are shown in the "c" boxplots (green), while samples of the *nat3Δ*-fed larvae or the *nat3Δ* yeast are shown in the "n" boxplots (pink). Logarithmic transformation and subsequent *t* test of MetaboAnalyst showed that pantothenate was significantly more abundant in the *nat3Δ*-fed larvae (left: $P < 0.001$, $FDR < 0.001$, right: $P < 0.01$, $FDR = 0.064922$). On the other hand, the pantothenate amount was not significantly different between the control and *nat3Δ* yeasts ($P = 0.967$). See more details in Datasets EV3 and EV5. (D) The biosynthetic pathway for CoA production. Pantothenate is first phosphorylated by pantothenate kinase (PANK). PANK is a rate-limiting enzyme in this pathway and feedback regulated by the end product, CoA, and also by acetyl-CoA and acyl-CoA species. (E) Expression values (TPM) of *FASN1\CG3523* in the *nat3Δ*-fed larvae (pink, "n"), the OVA-fed larvae (magenta, "OVA") and the respective controls (green, "c"; sky blue "C"). (F) Generalized enzymatic reaction of FASN1: a long-chain fatty acid and CoAs are produced from acetyl-CoAs in the cytoplasm. Boxplots are depicted as in "Statistical analysis" in Methods. *$P < 0.05$, **$P < 0.01$, ***$P < 0.001$. The exact $P$ values, sample sizes and statistical tests employed are listed in Dataset EV12. Source data are available online for this figure

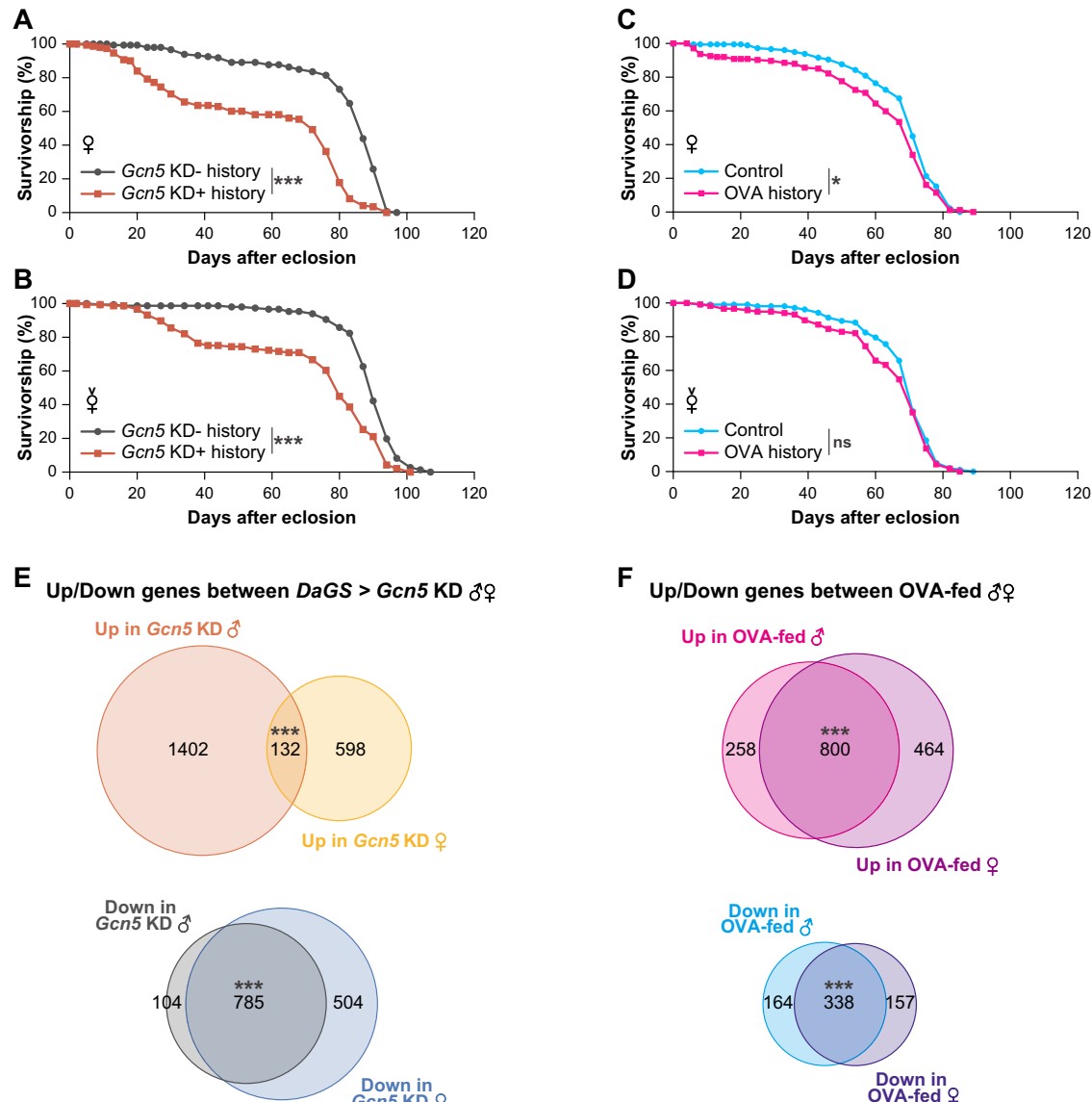

**Figure EV6. Impacts of the OVA diet or *Gcn5* knockdown in larval stages on gene expression of female larvae and female lifespan.**

(A, B) Survival curves of mated females (**A**) and virgin females (**B**) with or without the *Gcn5* KD history in larval stages. (C, D) Survival curves of mated females (**C**) and virgin females (**D**) with or without the OVA nutrition history in larval stages. (E, F) Comparisons of gene expression between male larvae and female larvae under the *Gcn5* KD condition (**E**) or the OVA diet condition (**F**). Venn diagrams showing overlaps of Up or Down genes between the sexes. The overlaps were significant in the individual comparisons and particularly large regarding the Down genes in the *Gcn5* KD condition, and both the Up and the Down genes in the OVA diet condition. All these data were obtained in a set of experiments. *$P < 0.05$, **$P < 0.01$, ***$P < 0.001$. The exact $P$ values, sample sizes and statistical tests employed are listed in Dataset EV12. Source data are available online for this figure

