## [Peer Review File · EMBO Reports]

Growth phase diets diminish histone acetyltransferase Gcn5 function and shorten lifespan of *Drosophila* males

Shoko Mizutani, Kanji Furuya, Ayumi Mure, Yuuki Takahashi, Akihiro Mori, Nozomu Sakurai, Takuto Suito, Kohjiro Nagao, Masato Umeda, Kaori Watanabe, Yukako Hattori, and Tadashi Uemura

Corresponding author(s): Tadashi Uemura (tauemura@lif.kyoto-u.ac.jp), Yukako Hattori (yhattori@lif.kyoto-u.ac.jp)

Review Timeline:

Submission Date:	5th Aug 24
Editorial Decision:	18th Sep 24
Decision Follow-up:	25th Sep 24
Revision Received:	25th Feb 25
Editorial Decision:	29th Apr 25
Revision Received:	25th May 25
Accepted:	2nd Jun 25

Editor: Deniz Senyilmaz Tiebe

Transaction Report:

Dear Tadashi,

Thank you for submitting your manuscript to EMBO Reports. My apologies for this unusual delay in getting back to you. Three referees agreed to review your manuscript. So far, we have received two referee reports that are copied below. Given that both referees are in fair agreement that you should be given a chance to revise the manuscript, I would like to ask you to begin revising your study along the lines suggested by the referees.

Please note that this is a preliminary decision made in the interest of time, and that it is subject to change should the third referee offer very strong and convincing reasons for this. As soon as we receive the final report on your manuscript, we will forward it to you as well.

Referees express interest in the proposed effect of a fatty acid- and BCAA-rich diet on fly lifespan by a mechanism involving Gcn5. However, they also raise some concerns that need to be addressed to consider publication here. In particular, sex specificity of the findings needs to be better demonstrated and discussed (both referees) and additional experiments are required to further support the proposed links. Unless the mechanism by which Gcn5 regulates gene expression is elucidated, the model in Figure 8 needs to be revised as per referee #2 (specific comment 2).

Given these positive recommendations, we would like to invite you to revise your manuscript with the understanding that the referee concerns (as in their reports) must be fully addressed and their suggestions taken on board. Please address all referee concerns in a complete point-by-point response. Acceptance of the manuscript will depend on a positive outcome of a second round of review. It is EMBO Reports policy to allow a single round of major experimental revision only and acceptance or rejection of the manuscript will therefore depend on the completeness of your responses included in the next, final version of the manuscript.

We realize that it is difficult to revise to a specific deadline. In the interest of protecting the conceptual advance provided by the work, we recommend a revision within 3 months. Please discuss the revision progress ahead of this time with me if you require more time to complete the revisions, or if you have questions or comments regarding the revision (also by video chat).

1. A data availability section providing access to data deposited in public databases is missing (where applicable).
2. Your manuscript contains statistics and error bars based on $n=2$. Please use scatter plots in these cases.

You can submit the revision either as a Scientific Report or as a Research Article. For Scientific Reports, the revised manuscript can contain up to 5 main figures and 5 Expanded View figures, and it should not exceed 27000 characters. If the revision leads to a manuscript with more than 5 main figures it will be published as a Research Article. In this case the Results and Discussion section should be separate. If a Scientific Report is submitted, these sections have to be combined. This will help to shorten the manuscript text by eliminating some redundancy that is inevitable when discussing the same experiments twice. In either case, all materials and methods should be included in the main manuscript file.

4) a .docx formatted letter INCLUDING the reviewers' reports and your detailed point-by-point responses to their comments. As part of the EMBO publication's Transparent Editorial Process, EMBO reports publishes online a Review Process File (RPF) to accompany accepted manuscripts. This File will be published in conjunction with your paper and will include the referee reports, your point-by-point response and all pertinent correspondence relating to the manuscript.

<https://www.embopress.org/page/journal/14693178/authorguide#transparentprocess>

5) a complete author checklist, which you can download from our author guidelines

<https://www.embopress.org/page/journal/14693178/authorguide>. Please insert information in the checklist that is also reflected in the manuscript. The completed author checklist will also be part of the RPF.

6) Please note that all corresponding authors are required to supply an ORCID ID for their name upon submission of a revised manuscript (<<https://orcid.org/>>). Please find instructions on how to link your ORCID ID to your account in our manuscript tracking system in our Author guidelines

<<https://www.embopress.org/page/journal/14693178/authorguide#authorshipguidelines>>

7) Before submitting your revision, primary datasets produced in this study need to be deposited in an appropriate public database (see <https://www.embopress.org/page/journal/14693178/authorguide#datadeposition>). Please remember to provide a reviewer password if the datasets are not yet public. The accession numbers and database should be listed in a formal "Data Availability" section placed after Materials & Method (see also

<https://www.embopress.org/page/journal/14693178/authorguide#datadeposition>). Please note that the Data Availability Section is restricted to new primary data that are part of this study. * Note - All links should resolve to a page where the data can be accessed. *

Additional information on source data and instruction on how to label the files are available:

<https://www.embopress.org/page/journal/14693178/authorguide#sourcedata>

9) Our journal encourages inclusion of *data citations in the reference list* to directly cite datasets that were re-used and obtained from public databases. Data citations in the article text are distinct from normal bibliographical citations and should directly link to the database records from which the data can be accessed. In the main text, data citations are formatted as follows: "Data ref: Smith et al, 2001" or "Data ref: NCBI Sequence Read Archive PRJNA342805, 2017". In the Reference list, data citations must be labeled with "[DATASET]". A data reference must provide the database name, accession number/identifiers and a resolvable link to the landing page from which the data can be accessed at the end of the reference. Further instructions are available at <http://www.embopress.org/page/journal/14693178/authorguide#referencesformat>

10) Regarding data quantification (see Figure Legends:

<https://www.embopress.org/page/journal/14693178/authorguide#figureformat>)

12) Please also note our reference format:

13) All Materials and Methods need to be described in the main text using our 'Structured Methods' format, which is required for all research articles. According to this format, the Methods section includes a Reagents and Tools Table (listing key reagents, experimental models, software and relevant equipment and including their sources and relevant identifiers) followed by a Methods and Protocols section describing the methods using a step-by-step protocol format. The aim is to facilitate adoption of the methodologies across labs. More information on how to adhere to this format as well as a downloadable template (.docx) for the Reagents and Tools Table can be found in our author guidelines:

I look forward to seeing a revised version of your manuscript when it is ready. Please let me know if you have questions or comments regarding the revision.

Kind regards,

Deniz

Deniz Senyilmaz Tiebe, PhD
Scientific Editor
EMBO Reports

Referee #1:

This manuscript presents an exceptionally massive dataset to address the role of early nutrition on lifespan, using *Drosophila*. The strategy is innovative, as it uses a screen with a collection of yeast mutants as a starting point. This is followed by extensive use of genomics, metabolomics and genetic experiments to understand the underlying causes. It is important to acknowledge that the scientific question addressed in this study is very challenging, considering the pleiotropic effects of nutrition on animal physiology, leading to difficulty to determine specific causal effects and molecular mechanisms. Despite these challenges, the authors manage to provide evidence for an epigenetic mechanism mediated through Gcn5, which is a clear merit. Despite the remaining gaps in the mechanism, which the authors properly acknowledge, the number of important biological conclusions obtained is sufficient to justify publication in EMBO Reports. However, I have two major concerns that should be thoroughly addressed prior to publication.

Concern 1: Sex specificity. The key finding of the study, reduced lifespan of flies following larval feeding on nat3delta yeast, is only observed in males but not in females (Figure 1). This is a very interesting finding. After this initial finding, however, sex specificity is not addressed in the rest of the manuscript (Figures 2-8). Moreover, the underlying reasons for the observed sex specificity are not discussed in the Discussion.

Action needed: To test the consistency of the findings and get better insight into the reasons of sex specificity, key experiments should be presented in a sex specific manner. These include Gcn5 knockdown and lifespan reduction by nutrient manipulation. Discussion should be amended to cover this topic as well.

Concern 2: Effect of nutrients. The results are mostly presented in a logical manner and key conclusions are supported by sufficient evidence. However, the final part of the Results section (esp. Figures 5-7), addressing the role of specific nutrients, does not meet similarly high standards. It was hard to follow the line of reasoning. As far as I understand, the evidence is not fully consistent with the conclusions, as addition of Oleic acid and valine (observed to be increased in nat3delta) was not sufficient to reduce the lifespan, but further addition of acetic acid (Ace) is required for reduced lifespan. The authors justify the use of Ace by presenting gene expression data with mild increase of AcCoAS expression. Despite this, the levels of Ace are

found to be reduced in the *nat3delta*, inconsistent with the model. Therefore, the manuscript falls short in explaining, which changes in nutrient availability explain the observed phenotypes.

Action needed: For the sake of clarity, I recommend radical shortening of the section on nutrient effects (Figures 5-7). In particular, reducing the weight of the gene expression data and emphasizing the metabolite measurements, which conclusively show the relevant changes in *nat3delta*, would be justifiable. The conditions used to test the nutrient effects should be based on observed changes in metabolite levels in *nat3delta*. One possibility to find the causal effects would be a broader manipulation of AA levels, as levels of many AAs are decreased in *nat3delta*. In addition to the observed increase of BCAA.

Referee #2:

In this manuscript, Mizutani et al., study how larval nutrition affects survival of adult *Drosophila melanogaster* fruit flies. To modify larval nutrition the authors used a collection of 5,153 yeast knock-out (KO) strains and then performed a screen in which fly larvae were fed with mutant yeast. From the screen the authors identified 46 yeast strains that affected fly development and two of these strains also affected subsequent survival of adult male flies. They then focused their analysis on one of the candidates the *nat3Δ* mutant strain. Noteworthy, the effect of *nat3Δ* on adult survival was specific for males and not observed in female flies. By gene expression profiling the authors next show that larvae fed with the *nat3Δ* mutant yeast strain share similarities to larvae mutant for the histone acetyltransferase *Gcn5*. H3K9ac CUT&RUN analysis showed a significant overlap in downregulated H3K9ac peaks between *nat3Δ* and *Gcn5* knock down animals but the peaks were not associated with diet-dependent gene expression changes. They next show that knock down of *Gcn5* specifically in the larval brain shortens adult male survival and they identified *Tg* and *Proc-R* as potential effector genes, whose downregulation in the larval brain also shortens adult survival. Next the authors used metabolomics to show that the *nat3Δ* mutant yeast strain contains higher levels of fatty acids and valine and a larval diet supplemented with valine, oleic acid and acetic acid recapitulated the effect of *nat3Δ* feeding on adult male survival and gene expression. Based on their findings, the authors propose that the yeast *nat3Δ* mutation increases dietary valine, oleic acid and acetic acid that then inhibit *Gcn5* function in fly larvae, which results in changes in neuronal H3K9ac and gene expression, that then via a still unknown mechanism shortens adult survival.

Overall, this is an interesting manuscript that used an innovative approach to modify larval nutrition and identified *Gcn5* as a new regulator of dietary history. However, there are still some concerns that need to be addressed before this manuscript can be accepted in EMBO Reports. One of my major concerns is the strong sex bias in the analysis. Why were the experiments only done in male and not also in female flies? Yes, females did not show the decrease in adult survival, but what about the molecular changes. Do females show the same molecular changes but are not affected by them or do they not react to the dietary change in the same way, which would be surprising. I think this is an important question and key experiments should be repeated with female flies. This should at least include addressing the effect of *Gcn5* larval knockdown and the OVA diet on adult female survival, and gene expression measurements in larvae and adult female flies either by RNA seq or selected genes by Q-RT-PCR. It is also remarkable that the sex bias is not addressed in the discussion, which needs to be changed.

Specific comments:

The connection between the dietary changes (*nat3Δ* and OVA) and *Gcn5* is only based on correlation. Why has this not been addressed more directly by activating (overexpressing) *Gcn5* in larvae fed with *nat3Δ* or the OVA diet? Even if the hypothesis is that *Gcn5* cannot be further activated under the dietary conditions due to the limitation in the relative acetyl-CoA levels, this experiment is still meaningful because it would confirm this hypothesis if no effect of the overexpression is observed. Also, it should be tested whether *Gcn5* knockdown and *nat3Δ*/OVA act additively in modulating adult survival.

Given the lack in overlap between H3K9ac reductions and differentially expressed genes in *nat3Δ*-fed larvae, how does *Gcn5* regulate gene expression? Is there any overlap between the genes changed in the adult brain compared to larval H3K9ac reductions, i.e. do the H3K9ac modifications prime later expression changes? Do reduced H3K9ac peaks include the *Tg* and *Proc-R* genes? The model in Fig. 8 should be adjusted, as currently there is no evidence that the change in H3K9ac are causative for the gene expression changes.

Fig S5D: "In the lower-expressed genes both on the *nat3Δ* diet and under the *Gcn5* KD, we selected two genes: Transglutaminase (*Tg*; Figure S5E) and Proctolin receptor (*Proc-R*; Figure S5F)."

Tg and *Proc-R* are only significantly regulated in *Gcn5* KD animals, but are not regulated in *nat3Δ* animals. Thus, *Tg* and *Proc-R* might explain the *Gcn5* effect but not of the dietary intervention. Again, to show that these genes are downstream of *Gcn5* and potentially of the *nat3Δ* effect, they would need to be activated in *Gcn5* KD larvae or *nat3Δ*/OVA animals, respectively.

Minor comment:

Fig 7: overlap should be shown between OVA, *nat3* and *Gcn5* separately for up and downregulated genes to highlight genes co-regulated among all conditions! Does the OVA diet also regulate *Tg* and *Proc-R*?

Dear Prof. Uemura,

I am following up to the editors' revision decision letter below as we have received the final report for your manuscript, from referee #3 (attached).

Please also address these comments while revising your manuscript and include your responses to these comments in your response letter.

Please note that Deniz is away this week so if you have any questions regarding these additional comments, please reach out to the editor next week.

Thank you.

Kind regards,
Bojana

Bojana Perkucin
Editorial Assistant
EMBO Press

Referee #3

This study by Mizutani et al characterized how a specific diet in developmental stages shortens the adult fly lifespan via the declined function of Gcn5. The authors performed a series of RNAseq, metabolomic and genetic experiment to demonstrate the causal link between the diet and diminished histone acetyltransferase Gcn5 function. These characterizations are very impressive and thorough, and I don't have major concerns about them.

However, I am not very excited about how a diet, or a gene pathway shortens lifespan, because there are numerous of ways to make animal live shorter. It will be super interesting to see pro-longevity effect. The authors screened 5,153 yeast KO strains, are there any strains found to increase the fly lifespan?

One potential concern is the mixed use of Canton-S and white Dahomey strains for the pupariation and lifespan assays. For example, the CS strain was primarily used for the pupariation assays, which was the first round of identifying candidates. However, all lifespans were performed in wDah. Lifespans are highly dependent on strains, and the impacts of a specific diet may be drastically different from stain to strain. It doesn't seem to be as concerning for this candidate in particular but others may have been ruled as a "false negative" as a result. It is necessary to elaborate on why this was the case. Materials and Methods section, pages 17-19.

I would explain the rationale for analyzing Gcn5 and Ada2a data and how it connects with the changes that occur when flies are fed Nat3 mutant yeast – lines 182-185, page 6.

Other specific comments and suggestions to further improve the manuscript:

- The title is a little too descriptive. Maybe something like “Alterations in juvenile nutrition cause shortened lifespan through diminished histone acetyltransferase Gcn5 function” – Page 1.

- In Abstract, the adverse effect on the function of Gcn5 is referenced frequently. This statement is pretty broad because this could mean several different things such as decrease expression (ruled out from the study), decreased ability to make the modifications, decreased ability to bind the histones, or decreased ability to assemble with other proteins to form the acetyltransferase histones. It would be better, if possible, to stipulate what you mean by diminished function – Page 2.

- I am not convinced by the data in Fig 2F and 2G where the authors compared their bulk RNAseq data with public microarray data, because there are a few major differences, such as gene detection accuracy between two technologies, fly food/diet condition, and fly genetics.

- On lines 124-127 page 4, this sentence can be rephrased to be easier to read and ensure that the oleic acid, acetic acid, and valine is not just suddenly mentioned without background. I would mention these in the previous sentence (lines 122-123) and then phrase that the supplementation of oleic acid, valine, and acetic acid during larval development will induce a shortened adult lifespan.

- I would use larger instead of fatter on line 157, page 5.

- I recommend replacing lines 167-175 (page's 5-6) with a conclusion about the section and then move the current lines to the beginning of the next section (line 179 on page 6). Because this approach has to do with the results in the next section (RNA sequencing).

- Specification needed for whether the yeast strain screening was performed in both males and females or only males. This could affect the findings from the assay.

- If females were screened, did they also show a developmental delay? What could contribute to the sex specific effects of this diet? Some discussion on the results would be good to include.

- I think it would be interesting to see how many of the DEGs in the adults overlapped with those in the larva and what processes those genes are involved in. There may be a unique opportunity to study whether an “epigenetic memory” becomes established during this feeding period and affects gene expression later in life.

- Figure 1F could be moved to supplement. I would prefer to see Gene Ontology analysis for the Nat3 mutant fed larva and/or the GO for the overlapping genes between the Nat3 and the Gcn5 mutant datasets. What if Gcn5 affects similar pathways with the Nat3 mutant fed yeast? This would establish a nice link or maybe that there are factors involved other than Gcn5. It would be good to also see the GO terms for the Gcn5 null mutant dataset to see if those overlap with the Nat3 ones.

- In Figure S4E, I would also perform GO analysis or a summary analysis on the overlapping genes.

- It's interesting that CG4860 is downregulated in the Nat3 mutant yeast fed flies even though this enzyme is involved in beta-oxidation, is it an inhibitor of the process? Figure 5B.

- Isoleucine is also enriched in the Nat3 mutant yeast and is a BCAA but was never tested, is there data as to whether this supplementation would affect pupariation and/or lifespan? Figure 5H.

Re: tracking number: EMBOR-2024-60144V1

Ms title: “A growth phase diet diminishes histone acetyltransferase Gcn5 function and shortens adult lifespan”

Ms title of the revised version: “Growth phase diets diminish histone acetyltransferase Gcn5 function and shorten adult lifespan”

Points of Revision

The reviewers' comments are written in *italics and blue*, to which our replies are provided below. Please note that we have either split a single comment into two halves or joined two comments where necessary to make our replies easier to read, and also that we have followed the nomenclature and formatting of *EMBO Reports*: Figure, Expanded View figure (Figure EV1 etc.), Appendix (Appendix Figure S1 etc.), and Dataset (Dataset EV1 etc.), unless described otherwise. Pertinent references are listed at the end of the individual responses for each reviewer.

To address the reviewers' concerns, we have performed a large set of additional experiments and carefully revised the original manuscript. As a result, we have now described various effects of multiple dietary and genetic interventions in *Drosophila* larval stages on different modes of responses in larvae and lifespan of adults of either males or females. Therefore, we first summarize our core findings and the points of our major revisions and subsequently explain how we addressed the individual concerns in detail.

We initially found that: 1) the *nat3Δ* yeast diet in larval stages diminished the function of histone acetyltransferase Gcn5, whose preferred substrates include H3K9, in larvae; and 2) the *nat3Δ* yeast diet in larval stages shortened the lifespan of male adults but not the lifespan of female adults. These observed effects of diet during early developmental stages on the adult lifespan are concordant with current notions of long-lasting epigenetic repercussions, which we refer to as a nutrition-history. These findings prompted us to identify key nutrients in the *nat3Δ* yeast diet; consequently, we have shown that “the OVA diet”, which is the control-yeast diet supplemented with three nutrients (oleic acid, valine and acetic acid), recapitulated the gene expression profile and genome-wide profiling of H3K9 acetylation in the *nat3Δ*-fed male larvae, as well as the developmental (new data: Figures 7A-7C) and lifespan phenotypes of the *nat3Δ*-history male adults. We further addressed (1) sex specificity, that is, how the OVA diet or *Gcn5* knockdown in larval stages impact gene expression in female larvae and female adult lifespan (new data: Figure EV6, and Appendix Figures S6A and S6B) and (2) the identity of the third key nutrient, that is, whether acetic acid, which was questioned as one of the key nutrients, is more abundant in the *nat3Δ* yeast than the control strain or not (new data: Figure 5L). Finally, we have carefully revised our model (Figure 7H).

To avoid confusion about sexes of the samples, we explicitly state whether the samples were males, females, or mixed sex in the individual sections of Results, the figure legends, and the datasets. Regarding the revision of the figures and the text relevant to our searches for the key nutrients, we have de-emphasized the gene expression data of the yeast-fed larvae and emphasized the metabolite measurements in the yeast diets. We have added pertinent citations to the References section in the revised version.

Response to Comments by Reviewer #1

This manuscript presents an exceptionally massive dataset to address the role of early nutrition on lifespan, using Drosophila. The strategy is innovative, as it uses a screen with a collection of yeast mutants as a starting point. This is followed by extensive use of genomics, metabolomics and genetic experiments to understand the underlying causes. It is important to acknowledge that the scientific question addressed in this study is very challenging, considering the pleiotropic effects of nutrition on animal physiology, leading to difficulty to determine specific causal effects and molecular mechanisms. Despite these challenges, the authors manage to provide evidence for an epigenetic mechanism mediated through Gcn5, which is a clear merit. Despite the remaining gaps in the mechanism, which the authors properly acknowledge, the number of important biological conclusions obtained is sufficient to justify publication in EMBO Reports.

However, I have two major concerns that should be thoroughly addressed prior to publication.

We thank reviewer #1 for appreciating our analysis and raising points for improving our manuscript.

Major concern 1: *The key finding of the study, reduced lifespan of flies following larval feeding on nat3Δ yeast, is only observed in males but not in females (Figure 1). This is a very interesting finding. After this initial finding, however, sex specificity is not addressed in the rest of the manuscript (Figures 2-8). Moreover, the underlying reasons for the observed sex specificity are not discussed in the Discussion.*

Action needed: To test the consistency of the findings and get better insight into the reasons of sex specificity, key experiments should be presented in a sex specific manner. These include Gcn5 knockdown and lifespan reduction by nutrient manipulation. Discussion should be amended to cover this topic as well.

We thank the reviewer for raising a critical point of revision (sex specificity). To address this point, we have carried out additional experiments (whole-body RNA-seq of female larvae and measurements of female adult lifespan under the OVA dietary or *Gcn5* knockdown conditions). Those results are described and discussed in the end of Results, where we have written a new section with the heading “Does the dietary or knockdown intervention impact female lifespan?” provided below for the reviewer’s convenience (revised manuscript, lines 500-535; Figure EV6; Appendix Figures S6A and S6B):

“We have so far described how the diets or the *Gcn5* knockdown in larval stages affect gene expression, metabolism, and lifespan of males. There are substantial differences in gene expression and metabolism in response to the amount of dietary sugar between male and female larvae (Millington et al., 2022). Therefore, we wondered whether any of the dietary or genetic interventions in this study impact the lifespan of female adults, and we assessed the extent to which the gene expression patterns differ between the sexes in larval stages. It is known that female lifespan of *Drosophila melanogaster* strains depends on mating status and fecundity (Austad and Fischer 2016). Therefore, we measured the lifespans of both mated and virgin females (Figure EV6A-EV6D). The ubiquitous *Gcn5* knockdown in larval stages strongly shortened the lifespans of both mated females and virgin females (Figures EV6A and EV6B), just as it shortened the male lifespan (Figure 4G); thus, the *Gcn5* knockdown in larvae effectively shortened the lifespan irrespective of the sex. On the other hand, it was difficult to draw a definite conclusion concerning the dietary effects on female lifespan, due to the following observations: first, the *nat3Δ* diet for larvae did not affect the lifespan of mated females (Figure EV1E). Second, the OVA diet marginally reduced the lifespan of mated females, whereas it did not significantly

affect that of virgin females (Figures EV6C and EV6D). It should be noted that females, whether they were mated or virgin, lived exceedingly longer than males under every condition that we tested: the *nat3Δ*-history datasets (compare Figures 1E and 1F with Figure EV1E); the OVA-history datasets (compare Figure 5K and Figure EV4O with Figure EV6C and EV6D); and the *Gcn5* KD±-history datasets (compare Figure 4G with Figure EV6A and EV6B).

RNA-seq datasets for both sexes were collected from the *Gcn5* KD± larvae and compared to each other. Downregulated genes in females largely matched those in males, implying that their mutual downregulation contributes to the common effect of the knockdown on the lifespan in both the sexes. Feeding larvae on the OVA diet also produced large overlaps of differentially expressed genes between the sexes (Figure EV6F; Dataset EV9), and those overlaps include the fatty-acid and the BCAA degradation pathway genes and *AcCoAS* (Appendix Figures S5). Furthermore, the gene expression profile of the OVA-fed female larvae showed a characteristic similarity to that of the *Gcn5* mutant, much the same as the respective profiles for OVA-fed vs. *Gcn5* mutant male larvae (compared Appendix Figures S6A and S6B with Figures 6C and 6D, respectively). These parallels suggest that *Gcn5* function is impaired in the OVA-fed larvae of both sexes. Nonetheless, the lifespan shortening of the OVA-history female adults was less conclusive compared to that of the male adults with the same nutrition history, as described above. We speculate that the OVA-history female adults, and possibly the *nat3Δ*-history female adults as well, may have better chances on the standard food to recover from the dietary effect in larval stages than the counterpart males, due to their extended longevity relative to males.”

Millington et al. A low-sugar diet enhances *Drosophila* body size in males and females via sex-specific mechanisms. *Development*. 2022 Mar 15;149(6):dev200491. doi: 10.1242/dev.200491.

Austad and Fischer. Sex Differences in Lifespan. *Cell Metab*. 2016 Jun 14;23(6):1022-1033. doi: 10.1016/j.cmet.2016.05.019.

Major Concern 2: *The results are mostly presented in a logical manner and key conclusions are supported by sufficient evidence. However, the final part of the Results section (esp. Figures 5-7), addressing the role of specific nutrients, does not meet similarly high standards. It was hard to follow the line of reasoning. As far as I understand, the evidence is not fully consistent with the conclusions, as addition of Oleic acid and valine (observed to be increased in nat3Δ) was not sufficient to reduce the lifespan, but further addition of acetic acid (Ace) is required for reduced lifespan. The authors justify the use of Ace by presenting gene expression data with mild increase of AcCoAS expression. Despite this, the levels of Ace are found to be reduced in the nat3Δ, inconsistent with the model. Therefore, the manuscript falls short in explaining, which changes in nutrient availability explain the observed phenotypes.*

Action needed (first half): For the sake of clarity, I recommend radical shortening of the section on nutrient effects (Figures 5-7). In particular, reducing the weight of the gene expression data and emphasizing the metabolite measurements, which conclusively show the relevant changes in nat3Δ, would be justifiable.

We thank the reviewer for raising another critical point of revision (the sections concerning our searches for the key nutrients). To reduce the weight of the gene expression data of the yeast-fed larvae and also emphasize the metabolite measurements in the yeast diets, we have substantially reorganized Figures 5 and 6 in the original manuscript, made a new Figure 5, which starts with comparisons of metabolite amounts between the control yeast and the *nat3Δ* yeast (long- and very long chain fatty acids in Figures 5A and amino acids in 5B), and moved the gene expression data to Figure EV3. Consequently, the first paragraph of the relevant section headed “The *nat3Δ* yeast diet is much more abundant in long-chain fatty acids (LCFAs) and branched-chain amino acids (BCAAs) than the control yeast diet” is now more succinct.

In the original version, we had working hypothesis that in addition to LCFAs and BCAAs, acetic acid might be the third key nutrient, and we speculated that it might be more abundant in the *nat3Δ* yeast than the control yeast. In addition to the intracellular amount of acetic acid in the *nat3Δ* yeast, as shown in the original version (“intracellular” of Figure 5L), we measured the amounts of short-chain fatty acids including acetic acid in the culture media of the two yeast strains (new data: “extracellular” of Figure 5L and the top row of Figure EV5A). This was because a substantial amount of acetic acid is secreted by *Saccharomyces cerevisiae*. However, the extracellular amount of acetic acid was not more abundant in the *nat3Δ* yeast than the control (“extracellular” of Figure 5L). Thus, we have concluded that the identity of a key nutrient(s) in the *nat3Δ* yeast diet, which critically influenced adult lifespan in combination with oleic acid and valine, is still unknown, and we revised our model (Figure 7H). We wrote a new paragraph in the end of the section headed “The control diet for larvae when supplemented with 3 nutrients (OVA) shortens adult lifespan as the *nat3Δ* diet does,” and described and discussed our results, provided below (revised manuscript, lines 408-423):

“Given the elevated level of *AcCoAS* transcripts in the *nat3Δ* -fed larvae, we predicted that *nat3Δ* yeast would produce more acetic acid compared to the control yeast. We measured the amounts of short-chain fatty acids including acetic acid both in the culture media and in the yeast cells, because it has been reported that a substantial amount of acetic acid is secreted by *Saccharomyces cerevisiae* (Blank et al., 2005; Kajihata et al., 2015; Zhang et al., 2022; Yatabe et al., 2023). Because the concentration of acetic acid in the culture medium is in the mM range (Zhang et al., 2022 and Yatabe et al., 2023), we suspected that the extracellular pool might be more abundant and more effective to larvae than the intracellular pool in the ingested yeasts. Contrary to our assumption, the amount of acetic acid in *nat3Δ* yeast culture was not higher in the control culture, and actually lower in the cells compared to the respective amounts for the control yeast (Figures 5L and EV5A; Datasets EV8B and EV8C). Thus, our results did not support the hypothesis of an increased amount of acetic acid in the *nat3Δ* diet, and the characterization of other key nutrients or metabolites that critically influence adult lifespan in combination with oleic acid and valine is regrettably inconclusive. Nonetheless, our results indicate that the control yeast diet supplemented with the 3 nutrients (OVA) mimicked the effects of the *nat3Δ* diet on adult lifespan, as described above, and larval responses, described next.”

Blank et al. Large-scale 13 C-flux analysis reveals mechanistic principles of metabolic network robustness to null mutations in yeast. *Genome Biol.* 2005. 6(6), 1–16. <https://doi.org/10.1186/gb-2005-6-6-r49>.

Kajihata et al. C-based metabolic flux analysis of *Saccharomyces cerevisiae* with a reduced Crabtree effect. *Journal of Bioscience and Bioengineering.* 2015. 120(2), 140–144. <https://doi.org/10.1016/j.jbiosc.2014.12.014>.

Yatabe et al. Improvement of ethanol and 2, 3-butanediol production in *Saccharomyces cerevisiae* by ATP wasting. *Microbial Cell Factories.* 2023. 22(204), 1–15. <https://doi.org/10.1186/s12934-023-02221-z>.

Zhang et al. Combined roles of exporters in acetic acid tolerance in *Saccharomyces cerevisiae*. *Biotechnol Biofuels Bioprod.* 2022 Jun 18;15(1):67. doi: 10.1186/s13068-022-02164-4.

Action needed (second half): The conditions used to test the nutrient effects should be based on observed changes in metabolite levels in nat3Δ. One possibility to find the causal effects would be a broader manipulation of AA levels, as levels of many AAs are decreased in nat3Δ, in addition to the observed increase of BCAA.

The reviewer suggested the strategy that we actually had taken before the submission of the original manuscript. We also suspected that the decreased compounds, such as amino acids in the *nat3Δ* yeast compared to the control yeast, might be causative of the *nat3Δ* diet-induced adult phenotype; accordingly, we addressed whether the addition of those to the *nat3Δ* diet would

rescue the shorter lifespan or not. However, the results of those experiments were negative and complicated, as described next:

We compared the list of the decreased metabolites in the *nat3Δ* yeast (Figure 5B and Dataset EV3) with the list of substances in the exome-matched FLYAA holidic medium (Piper et al., 2017), and selected primarily amino acids, lysine/K, histidine/H, asparagine/Q, glutamate/E and glutamine/Q, and a nucleoside, uridine, to prepare a presumptive “rescue” cocktail. Individual compounds were added to the medium, mSCM, so that the final concentrations in the medium were equivalent to those in FLYAA holidic medium (Referee Table 1). Despite our expectation, the cocktail exerted a detrimental effect, not a positive effect, on lifespan when it was added to the *nat3Δ* diet for larvae (right of Referee Figure 1). Puzzlingly, the lifespan became shorter when the cocktail was added to the control diet as well (left of Referee Figure 1). We also paid close attention to an essential amino acid, methionine/M, which was decreased in the *nat3Δ* yeast (Figure 5B), and we tested whether its addition alone to the *nat3Δ* diet might rescue the shorter lifespan of the *nat3Δ*-history adults (Referee Table 2). Unexpectedly, the M addition further shortened the lifespan (right of Referee Figure 2), although the effect of the M addition to the control yeast was not significant (left of Referee Figure 2). We have not included these results in the original or the revised manuscript.

Referee Table 1

Compound in the cocktail	Final concentration (already included + supplement)
Lysine/K	7.73 mM (0.274 + 7.46)
Histidine/H	4.22 mM (0.0954 + 4.21)
Asparagine/Q	7.78 mM (0 + 7.78)
Glutamate/E	8.12 mM (0 + 8.12)
Glutamine/Q	7.66 mM (0 + 7.66)
Uridine	0.246 mM (0 + 0.246)

Referee Figure 1

* $P < 0.05$, ** $P < 0.01$, *** $P < 0.001$ (Log-rank test).

Referee Table 2

Compound	Final concentration (already included + supplement)
Methionine/M	4.17 mM (0.134 + 4.04)

Referee Figure 2

Response to Comments by Reviewer #2

Overall, this is an interesting manuscript that used an innovative approach to modify larval nutrition and identified Gcn5 as a new regulator of dietary history. However, there are still some concerns that need to be addressed before this manuscript can be accepted in EMBO Reports.

We thank reviewer #2 for appreciating our analysis and raising points for improving our manuscript.

Major concern 1: *One of my major concerns is the strong sex bias in the analysis. Why were the experiments only done in male and not also in female flies? Yes, females did not show the decrease in adult survival, but what about the molecular changes. Do females show the same molecular changes but are not affected by them or do they not react to the dietary change in the same way, which would be surprising. I think this is an important question and key experiments should be repeated with female flies. This should at least include addressing the effect of Gcn5 larval knockdown and the OVA diet on adult female survival, and gene expression measurements in larvae and adult female flies either by RNA seq or selected genes by Q-RT-PCR. It is also remarkable that the sex bias is not addressed in the discussion, which needs to be changed.*

We thank the reviewer for raising a critical point of revision (sex specificity). To address this point, we have carried out additional experiments (whole-body RNA-seq of female larvae and measurements of female adult lifespan under the OVA dietary or *Gcn5* knockdown conditions). Those results are described and discussed in the end of Results, where we have written a new section with the heading “Does the dietary or knockdown intervention impact female lifespan?” (lines 500-535; Figure EV6, and Appendix Figures S6A and S6B):

“We have so far described how the diets or the *Gcn5* knockdown in larval stages affect gene expression, metabolism, and lifespan of males. There are substantial differences in gene expression and metabolism in response to the amount of dietary sugar between male and female

larvae (Millington et al., 2022). Therefore, we wondered whether any of the dietary or genetic interventions in this study impact the lifespan of female adults, and we assessed the extent to which the gene expression patterns differ between the sexes in larval stages. It is known that female lifespan of *Drosophila melanogaster* strains depends on mating status and fecundity (Austad and Fischer 2016). Therefore, we measured the lifespans of both mated and virgin females (Figure EV6A-EV6D). The ubiquitous *Gcn5* knockdown in larval stages strongly shortened the lifespans of both mated females and virgin females (Figures EV6A and EV6B), just as it shortened the male lifespan (Figure 4G); thus, the *Gcn5* knockdown in larvae effectively shortened the lifespan irrespective of the sex. On the other hand, it was difficult to draw a definite conclusion concerning the dietary effects on female lifespan, due to the following observations: first, the *nat3Δ* diet for larvae did not affect the lifespan of mated females (Figure EV1E). Second, the OVA diet marginally reduced the lifespan of mated females, whereas it did not significantly affect that of virgin females (Figures EV6C and EV6D). It should be noted that females, whether they were mated or virgin, lived exceedingly longer than males under every condition that we tested: the *nat3Δ*-history datasets (compare Figures 1E and 1F with Figure EV1E); the OVA-history datasets (compare Figure 5K and Figure EV4O with Figure EV6C and EV6D); and the *Gcn5* KD±-history datasets (compare Figure 4G with Figure EV6A and EV6B).

RNA-seq datasets for both sexes were collected from the *Gcn5* KD± larvae and compared to each other. Downregulated genes in females largely matched those in males, implying that their mutual downregulation contributes to the common effect of the knockdown on the lifespan in both the sexes. Feeding larvae on the OVA diet also produced large overlaps of differentially expressed genes between the sexes (Figure EV6F; Dataset EV9), and those overlaps include the fatty-acid and the BCAA degradation pathway genes and *AcCoAS* (Appendix Figures S5). Furthermore, the gene expression profile of the OVA-fed female larvae showed a characteristic similarity to that of the *Gcn5* mutant, much the same as the respective profiles for OVA-fed vs. *Gcn5* mutant male larvae (compared Appendix Figures S6A and S6B with Figures 6C and 6D, respectively). These parallels suggest that *Gcn5* function is impaired in the OVA-fed larvae of both sexes. Nonetheless, the lifespan shortening of the OVA-history female adults was less conclusive compared to that of the male adults with the same nutrition history, as described above. We speculate that the OVA-history female adults, and possibly the *nat3Δ*-history female adults as well, may have better chances on the standard food to recover from the dietary effect in larval stages than the counterpart males, due to their extended longevity relative to males.”

Millington et al. A low-sugar diet enhances *Drosophila* body size in males and females via sex-specific mechanisms. *Development*. 2022 Mar 15;149(6):dev200491. doi: 10.1242/dev.200491.

Austad and Fischer. Sex Differences in Lifespan. *Cell Metab*. 2016 Jun 14;23(6):1022-1033. doi: 10.1016/j.cmet.2016.05.019.

“key experiments with female flies should at least include (snip) gene expression measurements in larvae and adult female flies.”

To examine the sex specificity, the critical experiments were: (1) addressing the effects of *Gcn5* knockdown and the OVA diet in larval stages on female adult lifespan; and (2) gene expression measurements in larvae. We have done these experiments, and all the relevant data are explained above. The reason why we did not analyze gene expression in adult females was that the immediate cause of the early death of the male adults was unknown, as explained in Discussion (lines 624-637).

Specific comment 1: *The connection between the dietary changes (nat3Δ and OVA) and Gcn5 is only based on correlation. Why has this not been addressed more directly by activating (overexpressing) Gcn5 in larvae fed with nat3Δ or the OVA diet? Even if the hypothesis is that*

Gcn5 cannot be further activated under the dietary conditions due to the limitation in the relative acetyl-CoA levels, this experiment is still meaningful because it would confirm this hypothesis if no effect of the overexpression is observed. Also, it should be tested whether Gcn5 knockdown and nat3Δ/OVA act additively in modulating adult survival.

We agree with the reviewer that it would be important to strengthen the causal link between the response (gene expression) on the *nat3Δ* or the OVA diet, the diminished function of Gcn5 (the reduction of H3K9ac peaks), and the lifespan shortening by properly activating the Gcn5-containing complex on the *nat3Δ* or the OVA diet. Towards this goal, we attempted to conduct a “rescue” experiment to address whether overexpression of *Gcn5* in the *nat3Δ*-fed larvae normalizes the shorter lifespan of the male adults or not. To overexpress *Gcn5*, we planned to use the *GeneSwitch* (*GS*) driver stock that was used for larval stage-specific expression of short-hairpin RNA. However, we found that larvae of the *GS* stock did not grow on the live yeast diet in the presence of the chemical activator of the *GS* protein, RU486 (note that our knockdown experiments were done on the laboratory standard food containing RU486). It is known that RU486 inhibits digestion of triacylglycerol and absorption of free fatty acids in *Drosophila* adults (Ma et al., 2021) and we suspect that larvae are particularly sensitive to RU486 on our live yeast diet. We have briefly described this incompatibility between the *GS* stock and the live yeast diet in Discussion as follows (revised manuscript, lines 592-596): “Regarding the genetic rescue, we attempted to address whether *Gcn5* overexpression in the *nat3Δ*-fed or OVA-fed larvae affected lifespan. For this purpose, we performed a pilot experiment by feeding larvae harboring the *DaGS* driver construct; however, we found that the larvae could not grow on live yeast diets in the presence of the chemical activator of the *GS* protein. Thus, other approaches are necessary.” As an alternative approach, we searched for a chemical activator of Gcn5 but could not find any. Throughout the text, we have been careful to avoid any overstatements concerning the causality.

Ma et al. Mifepristone (RU486) inhibits dietary lipid digestion by antagonizing the role of glucocorticoid receptor on lipase transcription. *iScience*. 2021 May 12;24(6):102507. doi: 10.1016/j.isci.2021.102507.

Specific comment 2: *Given the lack in overlap between H3K9ac reductions and differentially expressed genes in nat3Δ-fed larvae, how does Gcn5 regulate gene expression?*

Our analysis did not provide evidence that the changes in H3K9ac are direct causes for the gene expression changes. Accordingly, we have carefully revised our model diagram (Figure 7H) and discussion concerning the connections between the H3K9ac reductions and changes of gene expression as follows (revised manuscript, lines 597-605): “Are the diet-induced Gcn5 malfunctions directly associated with the alterations in gene regulation? We examined how much the reduction in H3K9ac peaks by the *nat3Δ*-diet or the OVA-diet was correlated with the alterations of gene expression in the individual yeast diet-fed larvae. We found minor and unbiased overlaps between the genes showing diet-induced H3K9ac reductions and the Up or Down genes both in the *nat3Δ*-fed larvae and the OVA-fed larvae (Appendix Figures S6C and S6D). Similarly, overlaps between the Down H3K9ac peak-containing genes and the Up or Down genes in the *Gcn5* KD larvae were minor (Appendix Figure S6E). These results may reflect secondary and further indirect consequences of changes in transcription that were induced by the reduced H3K9ac peaks.”

Is there any overlap between the genes changed in the adult brain compared to larval H3K9ac reductions, i.e. do the H3K9ac modifications prime later expression changes? Do reduced H3K9ac peaks include the Tg and Proc-R genes? The model in Fig. 8 should be adjusted, as currently there is no evidence that the change in H3K9ac are causative for the gene expression changes.

Although the reviewer is asking some interesting questions about any overlap between the genes, whose expressions were changed in the adult brain, and larval H3K9ac reductions in larvae, we cannot answer the question because we investigated H3K9ac peaks only in whole-body larvae, not in the larval brain or adult brain.

CUT&RUN experiments to analyze alterations of H3K9ac peaks in the larval or adult brain may be challenging, due to its required sensitivity. Our speculation is based on a previous study where 10 larval brains were used for H3K27 tri-methylation (H3K27me3) analysis (Ahmad and Spens, 2019) and another study showing that the amount of H3K9ac is far less abundant than H3K27me3 by two orders of magnitude in *Drosophila* cell lines (see Table 3 in Feller et al. 2015). Considering that we have not provided concrete evidence that altered gene expression in the adult brain is due to the changes in the H3K9ac peaks, we have revised the model (Figure 7H). We did test whether we might be able to detect a difference in signal intensity between the control CNS and *Gcn5*-knockdown CNS by immunostaining with the anti-H3K9ac antibody; however, we did not observe such a difference reproducibly, due to the technical limitations.

Ahmad and Spens. Separate *Polycomb Response Elements* control chromatin state and activation of the *vestigial* gene. *PLOS Genetics*. 2019 August 19. <https://doi.org/10.1371/journal.pgen.1007877>.

Feller et al. Global and specific responses of the histone acetylome to systematic perturbation. *Molecular Cell*, 2015. 57(3), 559–571. <https://doi.org/10.1016/j.molcel.2014.12.008>

Specific comment 3: *Fig S5D (Appendix figure S3E in the revised manuscript): "In the lower-expressed genes both on the nat3D diet and under the Gcn5 KD, we selected two genes: Transglutaminase (Tg; Figure S5E [Appendix figure S3F]) and Proctolin receptor (Proc-R; Figure S5F [Appendix figure S3G])." Tg and Proc-R are only significantly regulated in Gcn5 KD animals, but are not regulated in nat3Δ animals. Thus, Tg and Proc-R might explain the Gcn5 effect but not of the dietary intervention. Again, to show that these genes are downstream of Gcn5 and potentially of the nat3Δ effect, they would need to be activated in Gcn5 KD larvae or nat3Δ/OVA animals, respectively.*

In Figures S5D-S5J in the original manuscript, we hypothesized that the *nat3Δ* diet-induced reduction in the *Gcn5* function would lead to abnormal expression patterns of a group (or groups) of genes in larval neurons, which might be an indirect cause of the lifespan shortening, and we attempted to verify this hypothesis. As the reviewer pointed out, *Tg* and *Proc-R* were significantly downregulated in *Gcn5* KD larvae, whereas their expressions tended to be decreased but not significantly in the *nat3Δ*-fed larvae. Thus, we have moved all those data and the relevant text to Appendix (Appendix Figures S3E-S3K and their legends), where we more cautiously discuss the connections between *Gcn5* function and changes in gene expression.

Minor comment 1: *Fig 7 (Figure 6A-6B in the revised version): overlap should be shown between OVA, nat3 and Gcn5 separately for up and downregulated genes to highlight genes co-regulated among all conditions! Does the OVA diet also regulate Tg and Proc-R?*

Although the reviewer suggested that we should highlight genes co-regulated among the three conditions, those Venn diagrams look exceedingly complicated, visually. Instead, we listed the overlapping upregulated and downregulated genes among the three datasets in Dataset EV9C including Gene Ontology analyses, which is stated in the Figure 6 legend. All these datasets were acquired from whole-body larvae, whereas *Tg* and *Proc-R* were DEGs in the larval CNS datasets.

Response to Comments by Reviewer #3

This study by Mizutani et al characterized how a specific diet in developmental stages shortens the adult fly lifespan via the declined function of Gcn5. The authors performed a series of RNAseq, metabolomic and genetic experiment to demonstrate the causal link between the diet and diminished histone acetyltransferase Gcn5 function. These characterizations are very impressive and thorough, and I don't have major concerns about them.

We thank reviewer #3 for appreciating our analysis.

Specific comment 1: *However, I am not very excited about how a diet, or a gene pathway shortens lifespan, because there are numerous of ways to make animal live shorter. It will be super interesting to see pro-longevity effect. The authors screened 5,153 yeast KO strains, are there any strains found to increase the fly lifespan?*

In a word, Yes! Feeding on *brp1Δ* yeast in larval stages increased the adult lifespan (Round 6 and 8 in Appendix Figure S2). The *brp1Δ* mutant and the *nat3Δ* mutant are the two yeast strains that were isolated as larval diets with the ability to affect adult stages in this study (Figure 1C in both the original manuscript and the revised version). Like this reviewer, we also consider the pro-longevity effect of the *brp1Δ* mutant interesting and have been collecting and analyzing omics data of the *brp1Δ* yeast and the *brp1Δ*-fed larvae, which hopefully we will be able to write up as a separate manuscript. On the other hand, the lifespan shortening effect of the *nat3Δ* is also interesting, because such a detrimental long-term effect has been documented in human epidemiological studies and rodent models (Eriksson et al., 1999; Ozanne & Hales, 2004; Barker & Thornburg, 2013; Kramer et al., 2023). It should be noted that out of the 29 yeast diets for larvae that cause developmental delay or increase mortality, only 2 diets (*nat3Δ* and *brp1Δ*) affected the adult lifespan (Figure 1C), indicating that those 2 nutrition histories must have unique features compared to the remaining 27. Before we found the pro-longevity effect of the *brp1Δ* mutant, we had already observed that the *nat3Δ*-induced lifespan shortening effect was reproducible; that was why we focused on *nat3Δ* in this study.

Eriksson et al. Catch-up growth in childhood and death from coronary heart disease: Longitudinal study. *British Medical Journal*, 1999. 318(7181), 427–431. <https://doi.org/10.1136/bmj.318.7181.427>.

Ozanne and Hales. Catch-up growth and obesity in male mice. *Nature*, 2004. 427(6973), 411–412. <https://doi.org/10.1038/427411b>.

Barker and Thornburg. The obstetric origins of health for a lifetime. *Clinical Obstetrics and Gynecology*, 2013. 56(3), 511–519. <https://doi.org/10.1097/GRF.0b013e31829cb9ca>.

Kramer et al. Maternal-fetal cross-talk via the placenta: influence on offspring development and metabolism. *Development (Cambridge, England)*, 2023. 150(20). <https://doi.org/10.1242/dev.202088>.

Specific comment 2: *One potential concern is the mixed use of Canton-S and white Dahomey strains for the pupariation and lifespan assays. For example, the CS strain was primarily used for the pupariation assays, which was the first round of identifying candidates. However, all lifespans were performed in wDah. Lifespans are highly dependent on strains, and the impacts of a specific diet may be drastically different from strain to strain. It doesn't seem to be as concerning for this candidate in particular but others may have been ruled as a “false negative” as a result. It is necessary to elaborate on why this was the case. Materials and Methods section, pages 17-19 (the original manuscript).*

As the reviewer pointed out, we employed a *Drosophila* strain, *wDah*, examined which of the 29 yeast mutant diets affected lifespan, and found that two yeast mutant diets did (Figure 1C and Appendix Figure S2). On the other hand, *CS* was used to examine the effects of the yeast mutants on adult emergence in Appendix Figure S1C, which is stated in its legend. We agree with the reviewer that lifespans are dependent on fly strains and the impacts of a specific diet might be different from strain to strain. Thus, we do not exclude the possibility that the remaining 27 yeast mutants might contain one or more "false negatives". We have mentioned this possibility in the "Live yeast-fly assay" section in Methods.

On the other hand, we would like to emphasize that *wDah* is a far more appropriate *Drosophila* strain for our live yeast-fly assay, compared to another strain *CS*, for the following reasons: First, *wDah* lays much more eggs than *CS*. Second, we needed to prepare germ-free embryos in the live yeast-fly assay, and the *wDah* germ-free embryos gave more than a 90% hatching rate persistently, which greatly facilitated our screening, whereas the rate of the *CS* germ-free embryos was only 70% ("Live yeast-fly assay" in Methods in both the original and revised manuscripts). Third, a separate project in our laboratory ran into an unanticipated trait of *CS*, although it has been a widely used wildtype strain: *CS* adults, which emerge on our standard food, show disorganized or lost adipose tissue (fat body) with approximately 30% genetic penetrance (see Fig. S5F-S5G and Table S2 in Tsuyama et al. *Development*, 2023: doi: 10.1242/dev.200815). The fat body plays essential roles in storing and mobilizing energy substrates and functions as a pivotal signaling center for inter-organ communications that regulate energetic metabolism at the organismal level. Therefore, we consider it very risky to examine adult traits including lifespan using *CS*. In contrast, such a severe phenotype of adult fat body is hardly seen in *wDah*. We have briefly described this concern with respect to *CS* in the "Live yeast-fly assay" section in Methods. Together with an abundance of literature pertaining to *wDah* and lifespan (Grandison et al., 2009 and Slack et al., 2015), we believe that *wDah* is the right choice in this study.

Grandison et al. Effect of a standardised dietary restriction protocol on multiple laboratory strains of *Drosophila melanogaster*. *PLoS ONE*, 2009. 4(1). <https://doi.org/10.1371/journal.pone.0004067>.

Slack et al. The Ras-Erk-ETS-Signaling Pathway Is a Drug Target for Longevity. *Cell*, 2015. 162(1), 72–83. <https://doi.org/10.1016/j.cell.2015.06.023>.

Specific comment 3: *I would explain the rationale for analyzing Gcn5 and Ada2a data and how it connects with the changes that occur when flies are fed nat3Δ mutant yeast – lines 182-185, page 6 (the original manuscript).*

We were prompted to pursue the relationship between the *nat3Δ* diet and Gcn5 function because of our and others' findings, as follows: (1) *neverland (nvd)* and 4 Halloween genes, *spookier (spok)*, *phantom (phm)*, *disembodied (dib)*, and *shadow (sad)*, were significantly upregulated in *nat3Δ*-fed larvae (Appendix Figure S3A). (2) They belong to the KEGG "insect hormone biosynthesis" pathway (Appendix Figure S4B) and encode ecdysteroidogenic enzymes that catalyze reactions from sterol to 20-hydroxyecdysone and/or other ecdysteroids. (3) It has been reported that expression of Halloween genes, including the above four, is regulated by the *Drosophila* ATAC complex that contains Gcn5 and Ada2a as subunits (Pankotai et al., 2010).

Although we explained the above rationale in a figure legend in the original manuscript, we did not explicitly describe where this rationale is in the text. Thus, we have amended the text to highlight that the rationale can be found in the Appendix Figure S3A legend, as follows (revised manuscript, lines 200-204): "upregulated or downregulated genes between the *nat3Δ*-fed larvae and the control larvae exhibited a striking similarity to those between *Gcn5* or *Ada2a* complete loss-of-function mutant larvae and their control genotypes (Figures 2B-2E; see Appendix Figure S3A and details in its legend regarding the rationale for analyzing the *Gcn5* data; Carré et al., 2008)."

Pankotai et al. Genes of the Ecdysone Biosynthesis Pathway Are Regulated by the dATAC Histone Acetyltransferase Complex in *Drosophila*. *Molecular and Cellular Biology*, 2010. 30(17), 4254–4266. <https://doi.org/10.1128/mcb.00142-10>.

Carré et al. The *Drosophila* NURF remodelling and the ATAC histone acetylase complexes functionally interact and are required for global chromosome organization. *EMBO Reports*, 2008. 9(2), 187–192. <https://doi.org/10.1038/sj.embor.7401141>.

Other specific comment 1: *The title is a little too descriptive. Maybe something like “Alterations in juvenile nutrition cause shortened lifespan through diminished histone acetyltransferase Gcn5 function” – Page 1.*

Although we thank this reviewer for the encouraging suggestion, this comment is related to, but conflicts with, Specific comment 1 of Reviewer #2, who is more cautious about the causal links between the alterations in juvenile nutrition (the *nat3Δ* diet or the OVA-supplemented diet compared to the control diet), the diminished Gcn5 function, and the lifespan shortening. As explained in our reply to Specific comment 1 of Reviewer #2, we found that a straightforward verification for the causality was technically difficult to achieve. Thus, we have employed the descriptive title and discussed the causality cautiously (revised manuscript, lines 1-2).

Other specific comment 2: *In Abstract, the adverse effect on the function of Gcn5 is referenced frequently. This statement is pretty broad because this could mean several different things such as decrease expression (ruled out from the study), decreased ability to make the modifications, decreased ability to bind the histones, or decreased ability to assemble with other proteins to form the acetyltransferase histones. It would be better, if possible, to stipulate what you mean by diminished function – Page 2.*

We state “diminished function of Gcn5” or “the adverse effect on the Gcn5 function” under the two dietary conditions (the *nat3Δ* diet and the OVA-supplemented diet) primarily on the basis of reductions in histone H3K9 acetylation in larvae (Figures 3B-3C and Figure 7A), which does not distinguish the underlying molecular mechanisms that the reviewer pointed out. Admittedly, we are limited in our ability to precisely define how the Gcn5 function is diminished, and our explanation remains more descriptive than mechanistic.

Other specific comment 3: *I am not convinced by the data in Fig 2F and 2G where the authors compared their bulk RNAseq data with public microarray data, because there are a few major differences, such as gene detection accuracy between two technologies, fly food/diet condition, and fly genetics.*

In both the original and the revised manuscripts, we show comparisons of our RNA-seq data with previously published microarray data not only in Figures 2F-2G, but also in the entire Figure 2, Figure 6, and Appendix Figures S6A and S6B. Importantly, in spite of the different technologies, foods, and fly strains, the Up and Down genes in the *nat3Δ*-fed or OVA diet-fed larvae overlap with the Up and Down genes in the *Gcn5* or *Ada2a* mutants, respectively, in a highly correlated manner (Figures 2B-2E, Figures 6C-6F, and Appendix Figures S6A and S6B), whereas such correlations are not seen with the Up and Down genes in the *Kdm4* mutant (Figures 2F, 2G, 6G and 6H). The respective degrees of agreement between the diet effect and the genotype effect were quantified by employing Cohen’s kappa (see coefficients that are associated with the individual Venn diagrams in the figures). Thus, we are confident that the comparisons are valid and meaningful.

Other specific comment 4: *On lines 124-127 page 4 (the original manuscript), this sentence can be rephrased to be easier to read and ensure that the oleic acid, acetic acid, and valine is*

not just suddenly mentioned without background. I would mention these in the previous sentence (lines 122-123 in the original manuscript) and then phrase that the supplementation of oleic acid, valine, and acetic acid during larval development will induce a shortened adult lifespan.

We thank the reviewer for raising this point for revision. We have revised the sentences in the original manuscript as follows (revised manuscript, lines 136-142): “The *nat3Δ* yeast was much more abundant in long-chain fatty acids (LCFAs) and valine, one of the branched-chain amino acids (BCAAs), than the control yeast. We explored supplementation of the control yeast diet with nutrients individually or in combination during larval development to identify which ones shortened adult lifespan; we found that the effective combination was oleic acid (a representative LCFA in yeast), valine, and acetic acid. Our data support the proposition that the high intake of these nutrients during larval development shortens adult lifespan via their adverse effect on Gcn5 function.”

Other specific comment 5: *I would use larger instead of fatter on line 157, page 5 (the original manuscript).*

We have removed the phrase “and the *nat3Δ*-fed larvae looked fatter than the control larvae (data not shown)” in the original manuscript, because we did not quantify larval body size.

Other specific comment 6: *I recommend replacing lines 167-175 (page’s 5-6 in the original manuscript) with a conclusion about the section and then move the current lines to the beginning of the next section (line 179 on page 6). Because this approach has to do with the results in the next section (RNA sequencing).*

We thank the reviewer for raising this point for revision. The lines “We assumed that the *nat3Δ* diet evoked some specific responses in larvae, which could not be restored on the standard food, even much later in adult life [...] (designated as *nat3Δ*-history adults or control adults hereafter)” have been replaced with a conclusion for that section, which is “Almost 200 confirmed or putative target proteins of yeast NatB have been reported (Caesar & Blomberg, 2004; Caesar et al., 2006; Helbig et al., 2010; Van Damme et al., 2012; Croft et al., 2018) and *nat3Δ* yeast exhibit pleiotropic phenotypes, including altered regulation of nicotinamide adenine dinucleotide metabolism (Wilson et al., 2002; Caesar et al., 2006; Croft et al., 2018; Sugaya et al., 2023)” (lines 180-184) and we moved “We assumed that the *nat3Δ* diet evoked some specific responses in larvae, [...]” to the beginning of the next section “Gene expression profiles are strikingly similar between *nat3Δ*-fed larvae and larvae with mutations in the histone acetyltransferase gene *Gcn5*” (lines 188-196).

Other specific comments 7 and 8: *Specification needed for whether the yeast strain screening was performed in both males and females or only males. This could affect the findings from the assay.*

If females were screened, did they also show a developmental delay? What could contribute to the sex specific effects of this diet? Some discussion on the results would be good to include.

We thank the reviewer for the helpful comments. The yeast strain screening for lifespan was performed in only males and we have stated “Among the 29 yeast KO larval diets, two yeast strains (*nat3Δ* and *brp1Δ*) affected the lifespan of male adults when they were aged on the standard laboratory food (Figure 1C; Appendix Figure S2; Datasets EV1 and EV2)” (revised manuscript, lines 163-165). To judge whether each diet caused a developmental delay or not, we

calculated adult eclosion percentages from the number of eclosed adults of both males and females (mixed sex) throughout this study including Appendix S1 and Figure 1D. In the revised manuscript, we explicitly state whether the samples were males, females, or mixed sex in the individual sections of Results, figure legends, and the datasets.

The comment regarding the sex specificity is related to Major concern 1 of Reviewer #1 and that of Reviewer #2. To extend our initial finding of the male-specific effect (Figures 1E and 1F versus Figure EV1E), we have carried out additional experiments (whole-body RNA-seq of female larvae and measurements of female adult lifespan under the OVA dietary or *Gcn5* knockdown conditions). Those results are described and discussed in the end of Results, where we have written a new section “Does the dietary or knockdown intervention impact female lifespan?” (revised manuscript, lines 500-535, Figure EV6, and Appendix Figures S6A and S6B): “We have so far described how the diets or the *Gcn5* knockdown in larval stages affect gene expression, metabolism, and lifespan of males. There are substantial differences in gene expression and metabolism in response to the amount of dietary sugar between male and female larvae (Millington et al., 2022). Therefore, we wondered whether any of the dietary or genetic interventions in this study impact the lifespan of female adults, and we assessed the extent to which the gene expression patterns differ between the sexes in larval stages. It is known that female lifespan of *Drosophila melanogaster* strains depends on mating status and fecundity (Austad and Fischer 2016). Therefore, we measured the lifespans of both mated and virgin females (Figure EV6A-EV6D). The ubiquitous *Gcn5* knockdown in larval stages strongly shortened the lifespans of both mated females and virgin females (Figures EV6A and EV6B), just as it shortened the male lifespan (Figure 4G); thus, the *Gcn5* knockdown in larvae effectively shortened the lifespan irrespective of the sex. On the other hand, it was difficult to draw a definite conclusion concerning the dietary effects on female lifespan, due to the following observations: first, the *nat3Δ* diet for larvae did not affect the lifespan of mated females (Figure EV1E). Second, the OVA diet marginally reduced the lifespan of mated females, whereas it did not significantly affect that of virgin females (Figures EV6C and EV6D). It should be noted that females, whether they were mated or virgin, lived exceedingly longer than males under every condition that we tested: the *nat3Δ*-history datasets (compare Figures 1E and 1F with Figure EV1E); the OVA-history datasets (compare Figure 5K and Figure EV4O with Figure EV6C and EV6D); and the *Gcn5* KD±-history datasets (compare Figure 4G with Figure EV6A and EV6B).

RNA-seq datasets for both sexes were collected from the *Gcn5* KD± larvae and compared to each other. Downregulated genes in females largely matched those in males, implying that their mutual downregulation contributes to the common effect of the knockdown on the lifespan in both the sexes. Feeding larvae on the OVA diet also produced large overlaps of differentially expressed genes between the sexes (Figure EV6F; Dataset EV9), and those overlaps include the fatty-acid and the BCAA degradation pathway genes and *AcCoAS* (Appendix Figures S5). Furthermore, the gene expression profile of the OVA-fed female larvae showed a characteristic similarity to that of the *Gcn5* mutant, much the same as the respective profiles for OVA-fed vs. *Gcn5* mutant male larvae (compared Appendix Figures S6A and S6B with Figures 6C and 6D, respectively). These parallels suggest that *Gcn5* function is impaired in the OVA-fed larvae of both sexes. Nonetheless, the lifespan shortening of the OVA-history female adults was less conclusive compared to that of the male adults with the same nutrition history, as described above. We speculate that the OVA-history female adults, and possibly the *nat3Δ*-history female adults as well, may have better chances on the standard food to recover from the dietary effect in larval stages than the counterpart males, due to their extended longevity relative to males.”

Millington et al. A low-sugar diet enhances *Drosophila* body size in males and females via sex-specific mechanisms. *Development*. 2022 Mar 15;149(6):dev200491. doi: 10.1242/dev.200491.

Austad and Fischer. Sex Differences in Lifespan. *Cell Metab*. 2016 Jun 14;23(6):1022-1033. doi: 10.1016/j.cmet.2016.05.019.

Other specific comment 9: *I think it would be interesting to see how many of the DEGs in the adults overlapped with those in the larva and what processes those genes are involved in. There may be a unique opportunity to study whether an “epigenetic memory” becomes established during this feeding period and affects gene expression later in life.*

We thank the reviewer for the insightful comment because we were also interested in how the alterations of gene expression in larvae persist in adulthood, with an expectation of visualizing “epigenetic memory.” As stated in the Figure 2A legend in the original and revised version, our whole-body larval RNA-seq analysis shows that the number of DEGs between the control yeast-fed larvae and the *nat3Δ*-fed male larvae was 1120 (693 up and 427 down in *nat3Δ*-fed), which decreased to 223 in the young male adults (121 up and 102 down in *nat3Δ*-history) and 165 in the midlife male adults (39 up and 126 down in *nat3Δ*-history). Out of the 1120 upregulated and downregulated genes in larval stages, only the expression of 6 was either kept up or down (Up: *timeless*, *Turandot A*, *CG6870* and *CG33926*; Down: *CG8560* and *GstZ2*) in the two adult stages. Whether such altered gene regulations contribute to lifespan or not awaits future investigations.

We also discuss how stable the altered H3K9ac profile unique to the *nat3Δ*-fed larvae might be, based on our datasets of the larval CNS and the adult brain (revised manuscript, lines 627-634): “Because epigenetic modifications of histones can be responsive to environmental inputs (Katan-Khaykovich & Struhl, 2002; Etchegaray & Mostoslavsky, 2016; Reid et al., 2017; Sharma & Rando, 2017; Dai et al., 2020; Oleson et al., 2021), the H3K9ac profile unique to the *nat3Δ*-fed larvae might be largely erased during the relatively much longer adult life on the laboratory standard food. Consistently, our comparison of RNA-seq datasets of the larval CNS and the adult brain indicates that the resultant mis-regulation of a set of genes in the *nat3Δ*-fed larval neurons does not persist in adults (Dataset EV7).”

Katan-Khaykovich and Struhl. Dynamics of global histone acetylation and deacetylation in vivo: Rapid restoration of normal histone acetylation status upon removal of activators and repressors. *Genes and Development*, 2002. 16(6), 743–752. <https://doi.org/10.1101/gad.967302>.

Etchegaray and Mostoslavsky. Interplay between Metabolism and Epigenetics: A Nuclear Adaptation to Environmental Changes. *Molecular Cell*, 2016. 62(5), 695–711. <https://doi.org/10.1016/j.molcel.2016.05.029>.

Reid et al. The impact of cellular metabolism on chromatin dynamics and epigenetics. *Nature Cell Biology*, 2017. 19(11), 1298–1306. <https://doi.org/10.1038/ncb3629>.

Sharma and Rando. Metabolic Inputs into the Epigenome. *Cell Metabolism*, 2017. 25(3), 544–558. <https://doi.org/10.1016/j.cmet.2017.02.003>.

Dai et al. The evolving metabolic landscape of chromatin biology and epigenetics. *Nature Reviews Genetics*, 2020. 21(12), 737–753. <https://doi.org/10.1038/s41576-020-0270-8>.

Oleson et al. Shaping longevity early in life: developmental ROS and H3K4me3 set the clock. *Cell Cycle*. 2021 Nov;20(22):2337-2347. doi: 10.1080/15384101.2021.1986317.

Other specific comment 10-1: *Figure 1F could be moved to supplement.*

We would like to show the data of Figure 1E and 1F side by side. This is because we have observed some variability in the survival curves for the *nat3Δ*-history male adults among 10 independent experiments, although they were significantly shorter-lived compared to the control adults in all ten experiments.

Other specific comment 10-2: *I would prefer to see Gene Ontology analysis for the *nat3Δ*-fed larva and/or the GO for the overlapping genes between the *nat3Δ*-fed larva and the *Gcn5* mutant datasets. What if *Gcn5* affects similar pathways with the *nat3Δ*-fed larva? This would*

establish a nice link or maybe that there are factors involved other than Gcn5. It would be good to also see the GO terms for the Gcn5 null mutant dataset to see if those overlap with the nat3 ones.

We thank the reviewer for the above comment. Because we show that both the transcriptomic data of the *nat3Δ*-fed larvae and that of the OVA-fed larvae show significant similarities to that of *Gcn5* mutant (Figure 2B-2E and 6C-6F, respectively), we listed overlapping DEGs among the three datasets (111 downregulated genes and 84 upregulated genes) in Dataset EV9C, which is described in the Figure 6 legend. GO terms (molecular function) of the trilaterally overlapping upregulated gene set include “iron-ion binding” and heme binding” (Dataset EV9C), as seen in “Up genes in *nat3Δ*-fed larvae” (Appendix figure S4A).

Other specific comment 11: *In Figure S4E (now Appendix Figure S6C in the revised manuscript), I would also perform GO analysis or a summary analysis on the overlapping genes.*

We thank the reviewer for the above comment. The overlapping genes in Appendix Figure S6C are labeled as “Down in nat3fed” or “Up in nat3fed” in the “Status of RNA-seq” column in Dataset EV6F. Because of each small gene number (110 Down genes and 83 Up genes), we have not conducted a DAVID analysis.

Other specific comment 12: *It's interesting that CG4860 is downregulated in the nat3Δ mutant yeast fed flies even though this enzyme is involved in beta-oxidation, is it an inhibitor of the process? Figure 5B (Figure EV3B in the revised manuscript).*

CG4860 is downregulated in both the *nat3Δ*-fed larvae (Figure EV3B) and in the OVA-fed male/female larvae (Appendix Figure S5B; Dataset EV9). Although it is annotated in FlyBase as “Predicted to enable short-chain fatty acyl-CoA dehydrogenase activity,” we cannot find the basis for this assertion. To our knowledge, no biochemical analysis of the protein product has been published. We do agree that it is potentially intriguing, but it should not be overemphasized at this stage.

Other specific comment 13: *- Isoleucine is also enriched in the nat3Δ mutant yeast and is a BCAA but was never tested, is there data as to whether this supplementation would affect pupariation and/or lifespan? Figure 5H (Figure 5B in the revised version).*

Yes, we supplemented the control yeast diet with oleic acid, valine, and isoleucine separately or in combination, and monitored whether any of the supplemented diets affected both the development (the timing and the rate of adult eclosion) and adult lifespan, in a similar manner to the *nat3Δ* yeast diet. These combinations include valine plus isoleucine (Figures EV4G and EV4H) and oleic acid plus valine plus isoleucine (Figures EV4I and EV4J), but neither combination shortened adult lifespan significantly.

Dear Tadashi,

Thank you for submitting your revised manuscript. It has now been seen by all of the original referees.

As you can see, referee finds that the study is significantly improved during revision and recommend publication. However, referee #2 has significant remaining outstanding concerns. Please address them by acknowledging the reported differential effects of GCN5 downregulation and the dietary interventions (OVA and nat3D diet) between male and female flies more prominently in the title, abstract, synopsis text and with more emphasis in the result section. Moreover, please add a discussion point on how GCN5 function is impaired by the nat3D diet and the connection between H3K9ac levels and gene expression changes in larvae and adult flies remains to be investigated. Lastly, please discuss how OVA diet composition was determined. Please include a point-by-point response to the remaining concerns of referee #3.

Furthermore, I need you to address the points below before I can accept the manuscript.

- Please move the Disclosure and competing interests statement section after the Acknowledgements section.
- Please remove the ORCID IDs from the title page of the manuscript.
- Please remove the Author contributions section from the manuscript.
- We note that the D81 cell of the Author Checklist remains to be responded.
- Please add page numbers to the Appendix file.
- Please remove the synopsis text from the manuscript and upload it as a separate word file.
- Please remove the Reagents and Tools table from the manuscript text and upload it as a separate word file.
- Please clarify in the source data checklist which panels share the same csv files as source data.
- Please remove the Google drive links from the Data Availability section and add links that directly resolve to the datasets in a public repository. Similarly, please add links that directly resolve to the datasets deposited in BioStudies as well (i.e. S-BSST1869, PRJDB17027, PRJDB18504, PRJDB20000 datasets).
- Please make the datasets PRJDB17027, PRJDB18504, PRJDB20000 publicly available.
- Supplementary Material section with the list of the Datasets needs to be removed from the manuscript.
- Our production/data editors have asked you to clarify several points in the figure legends - Figure Legends (main + EV):
 - o Please note that the exact p values are not provided in the legends of figures 1D-F, G-I; 4E, G, H, I, K, L, M, K; 7E-G; EV1 A, B; EV5 B, C, D, E; EV6 A, E;
 - o Please indicate what */ **/ ***/ **** represents; if this represents p value(s), please indicate the statistical test used and where appropriate and the exact p value in the legend(s) of figure(s) 4C
 - o Please indicate the statistical test used for data analysis in the legends of figures 4B, EV2 A
 - o Please note that the box plots need to be defined in terms of minima, maxima, centre, bounds of box and whiskers, and percentile in the legends of figures 4B, C, K; 5H, L; 6D-G; EV2 A, EV3 B, C, D, E, G; EV6 A, B, C, E.
 - o Please note that information related to n is missing in the legends of figures 3C, D; 4B, C; 5H, L; 6A, B; 6D-G; 7A, B; 7D-G; EV2 A, EV3 B, C, D, E, G; EV6 E.

Thank you again for giving us to consider your manuscript for EMBO Reports, I look forward to your minor revision.

Kind regards,

Deniz

--

Deniz Senyilmaz Tiebe, PhD
Senior Scientific Editor
EMBO Reports

Referee #1:

The authors have made a substantial effort to address the concerns I have raised in the original review. While some part have clearly improved, several key points of concern still remain somewhat obscure, which reflects the complexity of the research questions. It is plausible that additional experiments would not bring more clarity and therefore it is justified to publish this work as is, with its strengths and weaknesses. It is a valid addition to the literature and therefore might be suited for EMBO Reports.

Referee #2:

In their revised manuscript Mizutani et al. performed several new experiments and now provide the additional data for female

flies as requested by the reviewers. While I really appreciate the work the authors have invested, unfortunately the new data do not strengthen their conclusions nor do they support their model of how the larval nat3D diet affects adult survival. My main concern is the discrepancy observed between the effects of GCN5 downregulation and the dietary interventions (OVA and nat3D diet) between male and female flies. While larval downregulation of GCN5 shortens lifespan of both male and female flies, the dietary interventions only specifically affect male survival. Nevertheless, the molecular changes are very similar between male and female flies with larval GCN5 knock down or OVA and nat3D treatment. Thus, it is completely unclear why the same molecular changes have so different effects on male and female survival. The authors suggest without any evidence that this might be due to the longer "recovery time" of the female flies due to their longer lifespan, but this argument should also apply to the GCN5 knockdown flies. An alternative explanation could be that GCN5 is not the downstream effector of the nat3D diet effects. In this context it is very unfortunate, that their rescue attempt of activating GCN5 on the nat3D diet failed due to technical reasons. Given the new data, this experiment is really essential for publication, and other approaches than using the da-GS line should be pursued (e.g. is the elav-GS line also lethal, what about temperature sensitive Gal4/Gal80 system) to really establish that GCN5 mediates the effects of the nat3D on adult survival, which is the main conclusion of the manuscript. Furthermore, how GCN5 function is impaired by the nat3D diet and the connection between H3K9ac levels and gene expression changes in larvae and adult flies are unclear. Thus, there are too many unknowns in this model, which should prevent the publication at this time. Another concern are the data generated using the OVA diet, which in my opinion should not be included in the manuscript. The authors show that the two metabolites, which are increased (oleic acid, valine) are not sufficient to affect survival or cause similar molecular changes (e.g. H3K9ac levels) as the nat3D diet. The authors also show that acetic acid is not increased in the nat3D diet, so why test the combination of the three? This does not seem to be logical, and it does not provide evidence for a role of OV in the life-span shortening effects of the nat3D diet. I also don't agree with the way the female data were incorporated into the manuscript. The sex-specific effect should be reflected both in the title, abstract and more prominently in the result section. In summary, given the big gaps in the proposed mechanism and the above discussed discrepancy concerning the main conclusion of the study, I recommend to reject the manuscript.

Referee #3:

All my concerns have been sufficiently solved.

Re: tracking number: EMBOR-2024-60144V3

Ms title: “A growth phase diet diminishes histone acetyltransferase Gcn5 function and shortens adult lifespan”

Ms title of the re-revised version: “Growth phase diets diminish histone acetyltransferase Gcn5 function and shorten lifespan of *Drosophila* males”

Points of Revision

The reviewers' comments are written in *italics and blue*, to which our replies are provided below.

Response to the Comments by Reviewer #2

Main concern: *While I really appreciate the work the authors have invested, unfortunately the new data do not strengthen their conclusions nor do they support their model of how the larval nat3Δ diet affects adult survival. My main concern is the discrepancy observed between the effects of GCN5 downregulation and the dietary interventions (OVA and nat3Δ diet) between male and female flies. While larval downregulation of GCN5 shortens lifespan of both male and female flies, the dietary interventions only specifically affect male survival. Nevertheless, the molecular changes are very similar between male and female flies with larval GCN5 knock down or OVA and nat3Δ treatment. Thus, it is completely unclear why the same molecular changes have so different effects on male and female survival. The authors suggest without any evidence that this might be due to the longer "recovery time" of the female flies due to their longer lifespan, but this argument should also apply to the GCN5 knockdown flies. An alternative explanation could be that GCN5 is not the downstream effector of the nat3Δ diet effects.*

The above concern is related to Major concern 1 of this reviewer. As stated in our response to this concern, we now present the results of our entire additional experiments related to the sex difference in the final section in Results, under the heading “Does the dietary or knockdown intervention impact female lifespan?” and we state, cautiously, “it was difficult to draw a definite conclusion concerning the dietary effects on female lifespan.” We have now added a paragraph to the above section, starting with “our above results did not resolve the molecular basis, with respect to sex, of the discordant effects of the dietary interventions and the *Gcn5* knockdown on lifespan”, and we discuss the possible reasons why the dietary interventions significantly affected only the male lifespan in contrast to the strong effects of the *Gcn5* KD on both males and females:

“To conclude, our above results did not resolve the molecular basis, with respect to sex, of the discordant effects of the dietary interventions and the *Gcn5* knockdown on lifespan. It should be noted that females, whether they were mated or virgin, lived exceedingly longer than males under every condition tested: the *nat3Δ*-history datasets (compare Figures 1E and 1F with Figure EV1E); the OVA-history datasets (compare Figures 5K and EV4O with Figures EV6C and EV6D); and the *Gcn5* KD±-history datasets (compare Figure 4G with Figures EV6A and EV6B). We speculate that the OVA-history females, and possibly the *nat3Δ*-history females as well, may have better chances on the standard food to recover from the dietary effect in larval stages than the counterpart males, due to their extended longevity relative to males. Our results indicate that the effects of the *Gcn5* knockdown were more severe than those of the dietary interventions on lifespan irrespective of the sex (compare Figures 1E and 1F with Figure 4G, and Figures EV6A and EV6B with Figures EV6C and EV6D, respectively). The knockdown effects might be longer-lasting even in female adults compared to the dietary effects.”

In this context it is very unfortunate, that their rescue attempt of activating GCN5 on the nat3Δ diet failed due to technical reasons. Given the new data, this experiment is really essential for publication, and other approaches than using the da-GS line should be pursued (e.g. is the elav-GS line also lethal, what about temperature sensitive Gal4/Gal80 system) to really establish that GCN5 mediates the effects of the nat3Δ on adult survival, which is the main conclusion of the manuscript.

The above comment is related to Specific comment 1 of the original review of this reviewer and our response concerning the causal links between the alterations in juvenile nutrition, the diminished Gcn5 function, and the lifespan shortening. As explained in our reply in detail and in a paragraph in the Discussion, starting with “How well have we verified the hypothesis that the *nat3Δ* diet in larval stages is the cause of the shorter lifespan?”, we found that a straightforward verification of the causality was technically difficult to achieve. Thus, we have been careful to avoid any overstatements concerning the causality throughout the text and emphasized that the critical question “whether the short-lived phenotype of the *nat3Δ*-history adults would be restored if the *nat3Δ* diet-induced reduction in H3K9ac were restored either pharmacologically or genetically” needs to be verified in the future.

This reviewer suggested two alternative approaches to restore the *nat3Δ* diet-induced reduction in H3K9ac genetically. In one of the approaches, the temperature-sensitive Gal4/Gal80 system is used to overexpress *Gcn5* during larval stages by a temperature shift-up. Unfortunately, this approach requires a careful pre-examination concerning the confirmation of the *nat3Δ* diet effects on the lifespan and H3K9ac of that complex genotype, optimization of the duration of the shift-up that requires the *Gcn5* overexpression to restore the *nat3Δ* diet-induced reduction in H3K9ac, and the effect of this shift-up alone on the lifespan. Another approach uses the *ElavGS* line to overexpress *Gcn5* in larval neurons. However, before performing this experiment, it is necessary to establish a protocol to examine genome-wide profiling of H3K9ac in larval neurons. As explained in detail in our response to Specific comment 2 of this reviewer, CUT&RUN experiments to analyze alterations of H3K9ac peaks in the larval neurons may be technically challenging. A further predictable concern applicable to both approaches would be that the overexpression of the Gcn5 protein, the catalytic subunit of the ATAC complex, alone might result in hyper H3K9 acetylation, as implied in a previous study (compare Figures 6A and 6C of Carré et al. 2005).

Carré C, Szymczak D, Pidoux J, Antoniewski C (2005) The Histone H3 Acetylase dGcn5 Is a Key Player in *Drosophila melanogaster* Metamorphosis. *Molecular and Cellular Biology* 25: 8228–8238. PMID: 16135811

Furthermore, how GCN5 function is impaired by the nat3Δ diet and the connection between H3K9ac levels and gene expression changes in larvae and adult flies are unclear.

How GCN5 function is impaired by the nat3Δ diet: We have discussed this point in the second and third paragraphs of the Discussion, starting with “We explored how the intake of the *nat3Δ* diet dampened Gcn5 function and found that the acetyl-CoA|CoA ratio was decreased in the *nat3Δ*-fed larvae” and “In addition to the long-chain fatty acyl-CoA species, propionyl-CoA, one of the products of valine catabolism, would contribute to the reduction in the H3K9ac peaks in the *nat3Δ*-fed larvae, if its amount increases”, respectively, in our original and revised manuscripts (lines 545-570 in the revised manuscript). In these passages, we discuss how Gcn5 function is impaired by the *nat3Δ* diet, and our explanation is based on our data in the section “In the *nat3Δ*-fed larvae, the quantities of CoA-related metabolites were significantly changed, which could adversely affect Gcn5 function” in the Results.

The connection between H3K9ac levels and gene expression changes in larvae and adult flies:

This comment is essentially identical to Specific comment 2 of this reviewer. We discussed this point in our reply to this comment in detail and in Discussion of our revised manuscript. Please see the paragraph starting with “Are the diet-induced Gcn5 malfunctions directly associated with the alterations in gene regulation?”, which is now concluded by “The connections between the H3K9ac levels and the gene expression changes in larvae and adults remain to be investigated”, as below: “Are the diet-induced Gcn5 malfunctions directly associated with the alterations in gene regulation? We examined how much the reduction in H3K9ac peaks by the *nat3Δ*-diet or the OVA-diet was correlated with the alterations of gene expression in the individual diet-fed larvae. We found minor and unbiased overlaps between the genes showing diet-induced H3K9ac reductions and the Up or Down genes both in the *nat3Δ*-fed larvae and the OVA-fed larvae (Appendix Figures S6C and S6D). Similarly, overlaps between the Down H3K9ac peak-containing genes and the Up or Down genes in the *Gcn5* KD larvae were minor (Appendix Figure S6E). These results may reflect secondary and further indirect consequences of changes in transcription that were induced by the reduced H3K9ac peaks. The connections between the H3K9ac levels and the gene expression changes in larvae and adults remain to be investigated.”

*Another concern are the data generated using the OVA diet, which in my opinion should not be included in the manuscript. The authors show that the two metabolites, which are increased (oleic acid, valine) are not sufficient to affect survival or cause similar molecular changes (e.g. H3K9ac levels) as the *nat3Δ* diet. The authors also show that acetic acid is not increased in the *nat3Δ* diet, so why test the combination of the three? This does not seem to be logical, and it does not provide evidence for a role of OV in the lifespan shortening effects of the *nat3Δ* diet.*

The above point was raised in Major concern 2 of Reviewer #1. We have given sufficient details of how we determined the OVA diet composition, including why we became interested in acetic acid and how we devised the recipes, in our response to reviewer #1 and in two sections in Results of the revised manuscript: “The control diet for larvae when supplemented with 3 nutrients (OVA) shortens adult lifespan, just as the *nat3Δ* diet does” and “The OVA diet recapitulated the effects of the *nat3Δ* diet on larval gene expression and profiling of H3K9 acetylation”. Although the key compound that critically influences adult lifespan in combination with oleic acid and valine is inconclusive in this study, we are intrigued by a different short-chain fatty acid, butyric acid, which increased over two-fold in the *nat3Δ* extracellular medium (Figure EV5A).

I also don't agree with the way the female data were incorporated into the manuscript. The sex-specific effect should be reflected both in the title, abstract and more prominently in the result section.

We have heeded this advice and revised the title, abstract, and synopsis text to emphasize that our hypothesis in this study (“fatty acids- and branched-chain amino acid-rich diets in larval stages shorten adult lifespan via an adverse effect on Gcn5”) pertains to males. To reflect the sex difference more prominently in the Results section, we have added a paragraph to the final section in Results, under the heading “Does the dietary or knockdown intervention impact female lifespan?”, as described in the beginning of this response.

Prof. Tadashi Uemura
Kyoto University
Graduate School of Biostudies
Yoshida Konoe-cho, Sakyo-ku
Kyoto 606-8501
Japan

Dear Tadashi,

Thank you for submitting your revised manuscript. I have now looked at everything and all is fine. Therefore, I am very pleased to accept your manuscript for publication in EMBO Reports.

Congratulations on a nice work!

Kind regards,

Deniz

--

Deniz Senyilmaz Tiebe, PhD
Senior Scientific Editor
EMBO Reports

--
